# Convergences for Minimax Optimization Problems over Infinite-Dimensional Spaces Towards Stability in Adversarial Training

**Takashi Furuya**[*]                                                      *takashi.furuya0101@gmail.com*
*Shimane University*

**Satoshi Okuda**[*]                                                        *satoshi.okuda@aisin.co.jp*
*Tokyo Research Center, Aisin Corporation*

**Kazuma Suetake**                                                    *kazuma.suetake@aisin-software.com*
*AISIN SOFTWARE, Japan*

**Yoshihide Sawada**                                                    *yoshihide.sawada@aisin.co.jp*
*Tokyo Research Center, Aisin Corporation*

* These two authors contributed equally to this work

**Reviewed on OpenReview:** `https://openreview.net/forum?id=6LePXHr2f3`

## Abstract

Training neural networks that require adversarial optimization, such as generative adversarial networks (GANs) and unsupervised domain adaptations (UDAs), suffers from instability. This instability problem comes from the difficulty of the minimax optimization, and there have been various approaches in GANs and UDAs to overcome this problem. In this study, we tackle this problem theoretically through a functional analysis. Specifically, we show the convergence property of the minimax problem by the gradient descent over the infinite-dimensional spaces of continuous functions and probability measures under certain conditions. Using this setting, we can discuss GANs and UDAs comprehensively, which have been studied independently. In addition, we show that the conditions necessary for the convergence property are interpreted as stabilization techniques of adversarial training such as the spectral normalization and the gradient penalty.

**Keywords:** Minimax, Non-convex Optimization, Convergence Analysis, Adversarial Training, Functional Analysis.

## 1 Introduction

With the increased computational resources and available data, neural networks (NNs) trained by adversarial training have emerged prominently in various fields. An example is the application of generative adversarial networks (GANs) in generative tasks. GANs train the generator to capture the data distribution in an adversarial manner against the discriminator, which distinguishes between data generated by the generator and the dataset (Goodfellow et al., 2014). Another example is the utilization of adversarial training in unsupervised domain adaptations (UDAs) as generalization techniques. UDAs transfer knowledge from source domains to the target domain by extracting domain-invariant features against the domain critic that distinguish between data from source and target domains in an adversarial manner (Ganin and Lempitsky, 2015; Tzeng et al., 2017). Despite the effectiveness of GANs and UDAs, both pose challenges as nonconvex-nonconcave minimax problems, leading to inherent instability (Salimans et al., 2016). This instability, though insufficiently explored theoretically, complicates the widespread deployment of these models and hinders their

practical application. To address and pave the way for more robust applications, we analyze the instability problem from a functional analysis perspective.

As instability is related to the convergence properties of the gradient descent algorithm (Chu et al., 2020), we aim to clarify the convergence conditions for adversarial optimization problems. To facilitate the derivation of these conditions from the functional analysis perspective, we begin by considering the ideal setting. In our study, the ideal setting is derived from the dual formula of the minimization of a functional over probability distributions, leading to the *minimax* problem over *infinite-dimensional* spaces of continuous functions or probability measures. By exploring this minimax problem over infinite-dimensional spaces, we can prove the convergence to a minimax solution for a convex-concave setting (Section 5.1) and a stationary point for a nonconvex-concave setting (Section 5.2) under appropriate assumptions.

Throughout the convergence analyses, we maintain the assumption that the discrepancy measure, appearing in both GANs and UDAs, is strongly convex and *L*-smooth for the convergence. Achieving strong convexity involves confining the discriminator to a suitable subset within Lipschitz continuous function spaces. This concept aligns with the spectral normalization (Miyato et al., 2018). To ensure *L*-smoothness, we utilize the inf-convolution with a regularizer, such as the squared maximum mean discrepancy (MMD) in the reproducing kernel Hilbert spaces (RKHS) with the Gaussian kernel. This process corresponds to the gradient penalty (Gulrajani et al., 2017). Therefore, we can theoretically interpret widely-used stabilization techniques in adversarial training as the desired condition for achieving convergence properties.

**Contributions**

(A) We show the convergence to the minimax solution for a convex-concave setting and the stationary point for a nonconvex-concave setting over infinite-dimensional spaces of continuous functions or probability measures. This analysis is motivated by adversarial training in the scheme of the gradient descent (Section 5).

(B) We verify the fulfillment of sufficient conditions for the convergence properties in certain GANs and UDAs settings (Section 6), providing a theoretical interpretation of existing techniques such as the spectral normalization and gradient penalty.

## 2   Related Work

GAN training often exhibits an unstable trajectory, resulting in poor solutions (Goodfellow et al., 2014; Metz et al., 2016). To address this instability, various stabilization techniques have been proposed, including the Wasserstein GAN (Arjovsky et al., 2017), gradient penalty (Gulrajani et al., 2017), and spectral normalization (Miyato et al., 2018). The effectiveness of these techniques in stabilizing GAN training has been theoretically demonstrated (Chu et al., 2020). This theoretical result implies that the instability of GANs is due to adversarial training. Thus, UDAs with adversarial training are expected to encounter similar instability during training. Although methods to introduce various distances into UDAs to stabilize them have been investigated (Shen et al., 2018; Acuna et al., 2021; Wu et al., 2022; Chen and Marchand, 2023), a theoretical analysis encompassing UDAs and GANs with respect to instability has not yet been proposed. Notably, Chu et al. (2020) provides theoretical insight into GANs, interpreting stabilization techniques as conditions from the perspective of the minimization problem. On the other hand, our work provides similar theoretical insight from the viewpoint of the *minimax* problem rather than minimization, as the actual setting in GANs and UDAs employs the convex duality of the objective function to be minimized (Ganin and Lempitsky, 2015; Shen et al., 2018; Acuna et al., 2021). Firstly, we provide insights similar to Chu et al. (2020), which interpret stability techniques often used in GANs, such as spectral normalization and gradient penalty, via the sufficient conditions for minimax convergence, even though Chu et al. (2020) discussed the minimization problem. Secondly, we provide insight for UDAs, emphasizing that joint convexity is an important factor for stabilization, which could be achieved by strongly convex regularization for the predictor and generator. This is a new insight for UDAs, which Chu et al. (2020) never included, as they only considered minimization of divergence, not including source risk. It should be noted that while the existence of equilibrium in the GAN and UDA minimax problems has been discussed (Arora et al., 2017; Acuna et al., 2022), we discuss the convergence of appropriate algorithms to the equilibrium point.

Numerous references delve into the minimax optimization problem over finite-dimensional spaces, often treated as specific cases of Hilbert spaces. For instance, Cherukuri et al. (2017); Mokhtari et al. (2020); Du and Hu (2019) explore the convex-concave setting, while Huang et al. (2021); Thekumparampil et al. (2019); Lin et al. (2020a) focus on the nonconvex-concave setting. Although the minimax problem over Hilbert spaces has received extensive attention, with works such as Bauschke et al. (2017); Boţ et al. (2022); Bot et al. (2023), the exploration of the minimax problem over spaces of probability measures or continuous functions as opposed to Hilbert spaces remains relatively limited. On the other hand, our work delves into the minimax optimization problem for infinite-dimensional spaces of probability measures or continuous functions by generalizing results of Nedić and Ozdaglar (2009); Lin et al. (2020a). Moreover, we generalize the analysis from minimax to mininimax (minimizing with respect to the predictor and generator and maximizing with respect to the classifier).

## 3 Preliminary

This section describes the mathematical tools required in this paper.

Let $\mathbb{N}_0$ be the set of natural numbers including zero, $X \subset \mathbb{R}^d$ be a compact set, and $\overline{\mathbb{R}} = \mathbb{R} \cup \{-\infty, +\infty\}$ be the extended real number. We denote by $\mathcal{M}(X)$, $\mathcal{M}^+(X)$, and $\mathcal{P}(X)$ the set of all finite signed measures on $X$, the set of all non-negative finite measure on $X$, and the set of Borel probability measures on $X$, respectively. Let $\mathcal{C}(X)$ be the set of all continuous functions $X \to \mathbb{R}$. As shown in Aliprantis and Border (2006, Section 5.14), $\langle \mathcal{M}(X), \mathcal{C}(X) \rangle$ is a dual pair equipped with the bilinear functional

$$\langle \mu, \varphi \rangle := \int \varphi d\mu, \quad \mu \in \mathcal{M}(X), \ \varphi \in \mathcal{C}(X),$$

and the topological dual of $\mathcal{M}(X)$ with respect to weak topology is $\mathcal{C}(X)$ (Aliprantis and Border, 2006, Theorem 5.93). In the context of machine learning, we restrict $\mathcal{M}(X)$ to $\mathcal{P}(X)$. As $X$ is a compact subset of $\mathbb{R}^d$, $\mathcal{P}(X)$ is a compact subset in $\mathcal{M}(X)$ (see e.g., Aliprantis and Border (2006, Theorem 15.11)).

Let $\|\cdot\|_{\mathcal{M}(X)}$ and $\|\cdot\|_{\mathcal{C}(X)}$ be norms induced by inner products in $\mathcal{M}(X)$ and $\mathcal{C}(X)$, respectively. Then, we first define the dual norms, convex conjugation, and strong convexity as follows:

**Dual Norms** We denote dual norms $\|\cdot\|^\star_{\mathcal{M}(X)}$ and $\|\cdot\|^\star_{\mathcal{C}(X)}$ of $\|\cdot\|_{\mathcal{M}(X)}$ and $\|\cdot\|_{\mathcal{C}(X)}$ by, respectively,

$$\|\varphi\|^\star_{\mathcal{M}(X)} = \sup \left\{ \left| \int \varphi d\mu \right| : \|\mu\|_{\mathcal{M}(X)} \leq 1, \ \mu \in \mathcal{M}(X) \right\}, \quad \varphi \in \mathcal{C}(X), \tag{1}$$

$$\|\mu\|^\star_{\mathcal{C}(X)} = \sup \left\{ \left| \int \varphi d\mu \right| : \|\varphi\|_{\mathcal{C}(X)} \leq 1, \ \varphi \in \mathcal{C}(X) \right\}, \quad \mu \in \mathcal{M}(X). \tag{2}$$

**Convex Conjugation** The convex conjugates $F^\star$ and $G^\star$ of each functionals $F : \mathcal{C}(X) \to \overline{\mathbb{R}}$ and $G : \mathcal{M}(X) \to \overline{\mathbb{R}}$ are defined by, respectively,

$$F^\star(\mu) = \sup_{\varphi \in \mathcal{C}(X)} \int \varphi d\mu - F(\varphi), \quad \mu \in \mathcal{M}(X), \tag{3}$$

$$G^\star(\varphi) = \sup_{\mu \in \mathcal{M}(X)} \int \varphi d\mu - G(\mu), \quad \varphi \in \mathcal{C}(X). \tag{4}$$

**Strong Convexity** Let $S_{\mathcal{C}} \subset \mathcal{C}(X)$ and $S_{\mathcal{M}} \subset \mathcal{M}(X)$. We say that $F : \mathcal{C}(X) \to \overline{\mathbb{R}}$ and $G : \mathcal{M}(X) \to \overline{\mathbb{R}}$ are $\beta$-strongly convex ($\beta > 0$) with respect to $\|\cdot\|_{\mathcal{C}(X)}$ and $\|\cdot\|_{\mathcal{M}(X)}$ over $S_{\mathcal{C}}$ and $S_{\mathcal{M}}$, respectively, if it holds that for any $\alpha \in [0, 1]$

$$F(\alpha\psi + (1-\alpha)\varphi) \leq \alpha F(\psi) + (1-\alpha)F(\varphi) - \frac{\alpha(1-\alpha)\beta}{2} \|\psi - \varphi\|^2_{\mathcal{C}(X)}, \quad \psi, \varphi \in S_{\mathcal{C}}, \tag{5}$$

$$G(\alpha\mu + (1-\alpha)\nu) \leq \alpha G(\mu) + (1-\alpha)G(\nu) - \frac{\alpha(1-\alpha)\beta}{2} \|\mu - \nu\|^2_{\mathcal{M}(X)}, \quad \mu, \nu \in S_{\mathcal{M}}. \tag{6}$$

Next, we review Gâteaux differentials, Bregman divergences, and $L$-smoothness in order.

The Gâteaux differential is a generalization of the concept of directional derivative in finite-dimensional differential calculus. Let $F : \mathcal{C}(X) \to \overline{\mathbb{R}}$ and $G : \mathcal{M}(X) \to \overline{\mathbb{R}}$, then Gâteaux differentials are defined as follows.

**Definition 3.1.** *We define Gâteaux differentials $dF_\varphi : \mathcal{C}(X) \to \overline{\mathbb{R}}$ and $dG_\mu : \mathcal{M}(X) \to \overline{\mathbb{R}}$ of the functionals $F$ and $G$ at $\varphi \in \mathcal{C}(X)$ and $\mu \in \mathcal{M}(X)$ in the direction $\lambda \in \mathcal{C}(X)$ and $\chi \in \mathcal{M}(X)$ by, respectively,*

$$dF_\varphi(\lambda) := \lim_{\epsilon \to +0} \frac{F(\varphi + \epsilon\lambda) - F(\varphi)}{\epsilon},$$

$$dG_\mu(\chi) := \lim_{\epsilon \to +0} \frac{G(\mu + \epsilon\chi) - G(\mu)}{\epsilon}.$$

We note that if $F$ and $G$ are proper convex functionals, then for $\varphi \in \mathcal{C}(X)$ and $\mu \in \mathcal{M}(X)$ there exist Gâteaux differentials $dF_\varphi : \mathcal{C}(X) \to \overline{\mathbb{R}}$ and $dG_\mu : \mathcal{M}(X) \to \overline{\mathbb{R}}$, respectively (Aliprantis and Border, 2006, Lemma 7.14).

Then, we review the Bregman divergences. The Bregman divergences over spaces of measures and continuous functions measure between two points defined in terms of convex functions.

**Definition 3.2.** *Let $F : \mathcal{C}(X) \to \overline{\mathbb{R}}$ and $G : \mathcal{M}(X) \to \overline{\mathbb{R}}$ be proper, lower semi-continuous, and convex functionals. Then, F-Bregman divergence $D_F : \mathcal{M}(X) \times \mathcal{M}(X) \to \mathbb{R}_+$ and G-Bregman divergence $D_G : \mathcal{C}(X) \times \mathcal{C}(X) \to \mathbb{R}_+$ are defined by, respectively,*

$$D_F(\nu|\mu) := F(\nu) - F(\mu) - dF_\mu(\nu - \mu), \ \mu, \nu \in \mathcal{M}(X),$$

$$D_G(\psi|\varphi) := G(\psi) - G(\varphi) - dG_\varphi(\psi - \varphi), \ \varphi, \psi \in \mathcal{C}(X).$$

Finally, we review the $L$-smoothness. The $L$-smoothness over spaces of measures and continuous functions are defined using the Bregman divergence as follows.

**Definition 3.3.** *Let $S_\mathcal{C} \subset \mathcal{C}(X)$ and $S_\mathcal{M} \subset \mathcal{M}(X)$ be subsets, and $F : \mathcal{C}(X) \to \overline{\mathbb{R}}$ and $G : \mathcal{M}(X) \to \overline{\mathbb{R}}$ be proper, lower semi-continuous, and convex. Then, we say that $F$ and $G$ are $L$-smooth ($L > 0$) with respect to $\|\cdot\|_{\mathcal{C}(X)}$ and $\|\cdot\|_{\mathcal{M}(X)}$ over $S_\mathcal{C}$ and $S_\mathcal{M}$ if it holds that, respectively,*

$$D_F(\psi|\varphi) \leq \frac{L}{2} \|\psi - \varphi\|_{\mathcal{C}(X)}^2, \quad \varphi, \psi \in S_\mathcal{C},$$

$$D_G(\nu|\mu) \leq \frac{L}{2} \|\nu - \mu\|_{\mathcal{M}(X)}^2, \quad \mu, \nu \in S_\mathcal{M}.$$

## 4  Problem Setting

This section describes the problem setup of GAN and UDA training, building upon the reformulation introduced by Chu et al. (2019) as the foundation for our theoretical framework.

In their work, Chu et al. (2019) reformulated GAN training as a minimization problem with an objective function $J_{\nu_0}(\mu)$ over the set of probability measures, which represents a discrepancy measure between a generated distribution $\mu$ and an unknown true distribution $\nu_0$. Moreover, the adversarial loss can be obtained through the Fenchel-Moreau theorem. Consequently, they showed that various GAN models can be constructed by identifying particular discrepancy measures on an infinite dimensional space, such as the ordinal GAN (Goodfellow et al., 2014), maximal mean discrepancy (MMD) GAN (Li et al., 2015), $f$-GAN (Nowozin et al., 2016), and Wasserstein GAN (Arjovsky et al., 2017). Building upon this formulation, we extend it to unsupervised domain adaptation by adversarial training.

The UDA can be regarded as a simultaneous optimization problem for a source risk $R : \mathcal{C}(X) \times \mathcal{P}(X) \to \mathbb{R}$ and a discrepancy measure $J_{\nu_0}(\mu) : \mathcal{P}(X) \to \mathbb{R}$ between a source distribution $\mu$ and a fixed target distribution $\nu_0$. Then, the optimization problem for the UDA can be expressed as:

$$\min_{(\psi, \mu) \in \mathcal{C}(X) \times \mathcal{P}(X)} R(\psi, \mu) + J_{\nu_0}(\mu). \tag{7}$$

Here, the first variable $\psi$ in $R$ corresponds to the predictor. A typical example of $R$ is that $R(\psi, \mu) = \int |\psi(x) - \psi_0(x)|^2 d\mu(x) + V(\psi) + W(\mu)$ where $\psi_0$ is the true predictor, and $V : \mathcal{C}(X) \to \mathbb{R}$ and $W : \mathcal{P}(X) \to \mathbb{R}$ are certain regularization terms. The particular discrepancy measures lead to the well-known models of domain adversarial neural networks (DANNs) (Ganin and Lempitsky, 2015), such as DANNs with its extensions with Wasserstein-1 distance (Shen et al., 2018), $f$-divergence (Acuna et al., 2021), and MMD (Wu et al., 2022). As in the case of GANs (Chu et al., 2019), the Fenchel-Moreau theorem yields the following formulation equal to (7):

$$\min_{(\psi, \mu) \in \mathcal{C}(X) \times \mathcal{P}(X)} \max_{\varphi \in \mathcal{C}(X)} R(\psi, \mu) + \int \varphi d\mu - J_{\nu_0}^{\star}(\varphi). \tag{8}$$

This objective function is convex for $\psi$ and $\mu$, and concave for $\varphi$, where $\varphi$ corresponds to the domain classifier in the UDA. In Section 5.1, we delve into the convergence of this objective function in the general setting.

By omitting the source risk $R$, the formulation (8) reduces to that of GAN :

$$\min_{\mu \in \mathcal{P}(X)} \max_{\varphi \in \mathcal{C}(X)} \int \varphi d\mu - J_{\nu_0}^{\star}(\varphi), \tag{9}$$

where $\varphi$ corresponds to the discriminator in the GAN. This allows us to analyze the convergence properties in GANs and UDAs in a unified manner. In other words, the findings of GANs, which have been extensively studied for stability, could be used for UDAs. In fact, the assumptions used in this paper are related to the constraints of the GANs (see Section 6).

However, the formulation of (8), which extends the reformulation of Chu et al. (2019), deviates from minimax optimization in actual GANs and UDAs such as Goodfellow et al. (2014); Ganin and Lempitsky (2015), as it does not directly optimize the distribution $\mu$. To get more practical situations, we consider the source distribution $\mu$ as pushforward measure $f_\sharp \xi_0$ of fixed probability measure $\xi_0 \in \mathcal{P}(Z)$ by continuous function $f \in \mathcal{C}(Z; X)$, which corresponds to a generator in GANs, or a feature extractor in UDAs. Then, the problem (8) is reformulated as

$$\min_{\psi \in \mathcal{C}(X)} \min_{f \in \mathcal{C}(Z;X)} \max_{\varphi \in \mathcal{C}(X)} R(\psi, f_\sharp \xi_0) + \int \varphi d(f_\sharp \xi_0) - J_{\nu_0}^{\star}(\varphi). \tag{10}$$

This objective function is generally nonconvex for $\psi$ and $f$. In Section 5.2, we explore the convergence of this objective function in the general setting.

## 5 Minimax Analysis

Our goal in this section is to prove the convergence of the minimax optimization problem in the scheme of the gradient descent under appropriate assumptions. In Section 5.1, we will consider the convex-concave problem over spaces of continuous functions and probability measures, and prove that the sequence obtained by a certain gradient descent converges to the optimal minimax solution. While, in Section 5.2, we will consider the nonconvex-concave problem over spaces of continuous functions, and show that the sequence obtained by a certain gradient descent converges to a stationary point.

Note that the objective functions in Sections 5.1 and 5.2 are general forms of (8) and (10), respectively.

### 5.1 Convex-concave setting

This section considers the following minimax problem:

$$\min_{(\psi, \mu) \in S_1 \times S'} \max_{\varphi \in S_2} \mathcal{K}(\psi, \mu, \varphi), \tag{11}$$

where $S' \subset \mathcal{P}(X)$ and $S_1, S_2 \subset \mathcal{C}(X)$ are compact convex subsets and $\mathcal{K} : \mathcal{C}(X) \times \mathcal{M}(X) \times \mathcal{C}(X) \to \overline{\mathbb{R}}$ is supposed to be an objective function of GANs or UDAs. The typical example of $\mathcal{K}$ is the objective function in (8), that is,

$$\mathcal{K}(\psi, \mu, \varphi) = R(\psi, \mu) + \int \varphi d\mu - J_{\nu_0}^{\star}(\varphi).$$

We show that *the sequence obtained by the gradient descent converges to the optimal solution of* (11) *under appropriate assumptions.* To do this, we first consider the joint convexity as follows:

**Assumption 5.1.** *Assume the following:*

(i) $\mathcal{K}(\cdot, \cdot, \varphi)$ *is proper, lower semi-continuous, and convex over $S_1 \times S'$ for each $\varphi \in S_2$.*

(ii) $\mathcal{K}(\psi, \mu, \cdot)$ *is proper, upper semi-continuous, and concave over $S_2$ for each $\psi \in S_1$ and $\mu \in S'$.*

This assumption means that the problem (11) is a convex-concave problem. Under this assumption, Sion's minimax theorem (Sion, 1958) guarantees that

$$\min_{(\psi,\mu) \in S_1 \times S'} \max_{\varphi \in S_2} \mathcal{K}(\psi, \mu, \varphi) = \max_{\varphi \in S_2} \min_{(\psi,\mu) \in S_1 \times S'} \mathcal{K}(\psi, \mu, \varphi).$$

Moreover, there exists at least one minimax solution $(\psi_*, \mu_*, \varphi_*)$ in our minimax problem (11),

$$\begin{aligned} \mathcal{K}(\psi_*, \mu_*, \varphi_*) &\geq \mathcal{K}(\psi_*, \mu_*, \varphi), \quad \varphi \in S_2, \\ \mathcal{K}(\psi_*, \mu_*, \varphi_*) &\leq \mathcal{K}(\psi, \mu, \varphi_*), \quad (\psi, \mu) \in S_1 \times S'. \end{aligned} \tag{12}$$

Note that this assumption is in line with practical settings. Indeed, as shown in Section 6.1.1, the source risk $R$ of UDAs can be joint convex by adding both reproducing kernel Hilbert space (RKHS) (Alvarez et al., 2012) and maximal mean discrepancy (MMD) (Gretton et al., 2012) constraints.

Next, we put the assumptions related to the Gâteaux differentials. Let $\|\cdot\|_{\mathcal{C}(X),1}$ and $\|\cdot\|_{\mathcal{C}(X),2}$ be norms induced by inner products in $\mathcal{C}(X)$, and let $\|\cdot\|_{\mathcal{M}(X)}$ be a norm induced by an inner product in $\mathcal{M}(X)$. Note that both the first variable $\psi$ and the third variable $\varphi$ in $\mathcal{K}$ are continuous functions, but the inner product space $(\mathcal{C}(X), \|\cdot\|_{\mathcal{C}(X),1})$ for $\psi$ is different from the inner product space $(\mathcal{C}(X), \|\cdot\|_{\mathcal{C}(X),2})$ for $\varphi$.

**Assumption 5.2.** *We assume as follows:*

(i) *For each $\psi \in S_1$, $\mu \in S'$, and $\varphi \in S_2$, there exist the following arguments of the maximum:*

$$N_{\psi,\mu,\varphi} = \operatorname*{argmax}_{\nu \in \mathcal{M}(X)} \left\{ \int \psi d\nu - \mathcal{K}(\cdot, \mu, \varphi)^\star(\nu) \right\},$$

$$\Phi_{\psi,\mu,\varphi} = \operatorname*{argmax}_{\phi \in \mathcal{C}(X)} \left\{ \int \phi d\mu - \mathcal{K}(\psi, \cdot, \varphi)^\star(\phi) \right\},$$

$$\Lambda_{\psi,\mu,\varphi} = \operatorname*{argmax}_{\lambda \in \mathcal{M}(X)} \left\{ \int \varphi d\lambda - \mathcal{K}(\psi, \mu, \cdot)^\star(\lambda) \right\}.$$

(ii) $N_{\psi,\mu,\varphi}$, $\Phi_{\psi,\mu,\varphi}$, *and* $\Lambda_{\psi,\mu,\varphi}$ *are bounded with respect to dual norms* $\|\cdot\|^\star_{\mathcal{C}(X),1}$, $\|\cdot\|^\star_{\mathcal{M}(X)}$, *and* $\|\cdot\|^\star_{\mathcal{C}(X),2}$ *, that is, there exists $B > 0$ such that, for $(\psi, \mu, \varphi) \in S_1 \times S' \times S_2$,*

$$\|N_{\psi,\mu,\varphi}\|^\star_{\mathcal{C}(X),1} \leq B, \quad \|\Phi_{\psi,\mu,\varphi}\|^\star_{\mathcal{M}(X)} \leq B, \quad \|\Lambda_{\psi,\mu,\varphi}\|^\star_{\mathcal{C}(X),2} \leq B. \tag{13}$$

Here, $\mathcal{K}(\cdot, \mu, \varphi)^\star$, $\mathcal{K}(\psi, \cdot, \varphi)^\star$, and $\mathcal{K}(\psi, \mu, \cdot)^\star$ are convex conjugates of $\mathcal{K}(\cdot, \mu, \varphi)$, $\mathcal{K}(\psi, \cdot, \varphi)$, and $\mathcal{K}(\psi, \mu, \cdot)$, respectively. The above assumption guarantees that the existence of Gâteaux differentials of $\mathcal{K}$, and provide the form of their Gâteaux differentials as following lemma. The proof is given by the similar arguments in Chu et al. (2019, Theorem 2).

**Lemma 5.3.** *Let Assumption 5.2 hold. Then, for each $\psi \in S_1$, $\mu \in S'$, and $\varphi \in S_2$, there exist Gâteaux differentials $d\mathcal{K}(\cdot, \mu, \varphi)_\psi$, $d\mathcal{K}(\psi, \cdot, \varphi)_\mu$, and $d\mathcal{K}(\psi, \mu, \cdot)_\varphi$ of $\mathcal{K}(\cdot, \mu, \varphi)$, $\mathcal{K}(\psi, \cdot, \varphi)$, and $\mathcal{K}(\psi, \mu, \cdot)$ at $\psi \in S_1$, $\mu \in S'$, and $\varphi \in S_2$, and they are expressed as follows:*

$$d\mathcal{K}(\cdot, \mu, \varphi)_\psi(\eta) = \int \eta dN_{\psi,\mu,\varphi},$$

$$d\mathcal{K}(\psi, \cdot, \varphi)_\mu(\chi) = \int \Phi_{\psi,\mu,\varphi} d\chi,$$

$$d\mathcal{K}(\psi, \mu, \cdot)_\varphi(\phi) = \int \phi d\Lambda_{\psi,\mu,\varphi}.$$

In addition to these assumptions, we assume the $L$-smoothness of $\mathcal{K}$ for each variable to show the convergence to a minimax solution.

**Assumption 5.4.** *Let $L > 0$. Then, we assume as follows:*

  (i) *For each $\mu \in S'$ and $\varphi \in S_2$, $\mathcal{K}(\cdot, \mu, \varphi)$ is $L$-smooth with respect to $\|\cdot\|_{\mathcal{C}(X),1}$ over $S_1$.*

  (ii) *For each $\psi \in S_1$ and $\varphi \in S_2$, $\mathcal{K}(\psi, \cdot, \varphi)$ is $L$-smooth with respect to $\|\cdot\|_{\mathcal{M}(X)}$ over $S'$.*

  (iii) *For each $\psi \in S_1$ and $\mu \in S'$, $-\mathcal{K}(\psi, \mu, \cdot)$ is $L$-smooth with respect to $\|\cdot\|_{\mathcal{C}(X),2}$ over $S_2$.*

This assumption also aligns with practical settings, e.g., (iii) corresponds to the case where the $f$-divergence (Ali and Silvey, 1966; Csiszár, 1967) or integral probability metric (IPM) (Müller, 1997) is utilized as a discrepancy measure.

Here, we define the gradient descent for solving minimax optimization problem (11).

**Definition 5.5.** *Let $\psi_0 \in S_1$, $\mu_0 \in S'$ $\varphi_0 \in S_2$ be initial guesses. We define the gradient descent $\{(\psi_n, \mu_n, \varphi_n)\}_{n \in \mathbb{N}_0} \subset S_1 \times S' \times S_2$ by*

$$\psi_{n+1} = \operatorname*{argmin}_{\psi \in S_1} \left\{ d\mathcal{K}(\cdot, \mu_n, \varphi_n)_{\psi_n}(\psi - \psi_n) + \frac{1}{2\alpha_n} \|\psi - \psi_n\|_{\mathcal{C}(X),1}^2 \right\},$$

$$\mu_{n+1} = \operatorname*{argmin}_{\mu \in S'} \left\{ d\mathcal{K}(\psi_n, \cdot, \varphi_n)_{\mu_n}(\mu - \mu_n) + \frac{1}{2\alpha_n} \|\mu - \mu_n\|_{\mathcal{M}(X)}^2 \right\},$$

$$\varphi_{n+1} = \operatorname*{argmax}_{\varphi \in S_2} \left\{ d\mathcal{K}(\psi_n, \mu_n, \cdot)_{\varphi_n}(\varphi - \varphi_n) - \frac{1}{2\alpha_n} \|\varphi - \varphi_n\|_{\mathcal{C}(X),2}^2 \right\},$$

*where $\alpha_n > 0$ is the step size of the update rule.*

If subsets $S_1$, $S'$, and $S_2$ are subspaces, then update can be expressed as a sum of a previous step and a gradient term, a form that is commonly encountered in the gradient descent algorithm (see e.g., Chong et al. (2023)). However, in the general case of subsets $S_1$, $S'$, and $S_2$, the argmin and argmax in Definition 5.5 may not exist. Therefore, in this paper, the following assumption is established to ensure the existence of the gradient descent of Definition 5.5.

**Assumption 5.6.** *Assume that there exists a sequence $\{(\psi_n, \mu_n, \varphi_n)\}_{n \in \mathbb{N}_0} \subset S_1 \times S' \times S_2$ defined in Definition 5.5.*

Building upon the background established above, we are ready to present our main theorem of this section:

**Theorem 5.7.** *Let Assumptions 5.1, 5.2, 5.4, and 5.6 hold, and let $0 < \alpha_n \leq 1/L$.*
*Let $\{(\psi_n, \mu_n, \varphi_n)\}_{n \in \mathbb{N}_0} \subset S_1 \times S' \times S_2$ be the gradient descent defined in Definition 5.5. Let $(\psi_*, \mu_*, \varphi_*)$ be a minimax solution for (11). Then, for any $N \in \mathbb{N}$, we have*

$$\left| \mathcal{K}(\widehat{\psi}_N, \widehat{\mu}_N, \widehat{\varphi}_N) - \mathcal{K}(\psi_*, \mu_*, \varphi_*) \right| \leq \left( \sum_{n=0}^{N-1} \alpha_n \right)^{-1} \left( \frac{1}{2} C_s + 6B^2 \sum_{n=0}^{N-1} \alpha_n^2 \right), \tag{14}$$

*where*

$$C_s := \sup_{\psi \in S_1} \|\psi - \psi_0\|_{\mathcal{C}(X),1}^2 + \sup_{\mu \in S'} \|\mu - \mu_0\|_{\mathcal{M}(X)}^2 + \sup_{\varphi \in S_2} \|\varphi - \varphi_0\|_{\mathcal{C}(X),2}^2. \tag{15}$$

*Here, $\widehat{\psi}_N$, $\widehat{\mu}_N$, and $\widehat{\varphi}_N$ are weighted averages given by*

$$\widehat{\psi}_N := \frac{\sum_{n=0}^{N-1} \alpha_n \psi_n}{\sum_{n=0}^{N-1} \alpha_n}, \quad \widehat{\mu}_N := \frac{\sum_{n=0}^{N-1} \alpha_n \mu_n}{\sum_{n=0}^{N-1} \alpha_n}, \quad \widehat{\varphi}_N := \frac{\sum_{n=0}^{N-1} \alpha_n \varphi_n}{\sum_{n=0}^{N-1} \alpha_n}. \tag{16}$$

*Proof.* See Appendix A. □

We note that the the constant $C_s$ is finite due to the compactness of $S'$, $S_1$, and $S_2$. We observe that the upper bounds (14) with different choices of step sizes $\alpha_n \in (0, 1]$ are as follows:

- If the step sizes are constant, denoted by $\alpha_n = \alpha$, then the right-hand side (RHS) of (14) is expressed as

$$\text{RHS of (14)} = \frac{C_s}{2\alpha N} + 6B^2\alpha,$$

  which does not converges to zero as $N \to \infty$. Therefore, in this case, weighted averages (16) provides an approximate solution to the minimax problem. The first term converges to zero as $N \to \infty$ with an order of $\mathcal{O}(1/N)$. The second term can be reduced as $\alpha \to 0$, despite the first term diverging. This is a trade-off relationship with respect to the step size $\alpha$. A similar observation was made in Nedić and Ozdaglar (2009, Proposition 3.1), which studied the minimax problem in finite dimensional space using subgradient methods.

- If step sizes decay as $\alpha_n = \alpha/\sqrt{n}$ where $\alpha$ is a constant, then the right-hand side of (14) is expressed as

$$\text{RHS of (14)} = \frac{C_s}{2\alpha\sqrt{N}} + \frac{6B^2\alpha}{\sqrt{N}}(1 + \log N),$$

  which converges to zero as $N \to \infty$ with an order of $\mathcal{O}(\log N/\sqrt{N})$. Therefore, in this case, weighted averages (16) provide an exact solution to the minimax problem.

## 5.2 Nonconvex-concave setting

Unlike the previous section, which considered the convex-concave minimax problem expressed in (11), this section considers the nonconvex-concave minimax problem.

Let $Z \subset \mathbb{R}^{d'}$ be a compact set, and let $\mathcal{C}(Z; X)$ be the set of all continuous functions $Z \to X$, and let $S'' \subset \mathcal{C}(Z; X)$ and $S_1, S_2 \subset \mathcal{C}(X)$ be subspaces. Then, we consider the following minimax problem:

$$\min_{\psi \in S_{1,c}} \min_{f \in S''_c} \max_{\varphi \in S_{2,c}} \mathcal{G}(\psi, f, \varphi), \tag{17}$$

where $S''_c \subset S''$, $S_{1,c} \subset S_1$, and $S_{2,c} \subset S_2$ are convex subsets, and $\mathcal{G} : \mathcal{C}(X) \times \mathcal{C}(Z; X) \times \mathcal{C}(X) \to \overline{\mathbb{R}}$ is supposed to be an objective function of GANs or UDAs. The typical example of $\mathcal{G}$ is the objective function in (10), that is,

$$\mathcal{G}(\psi, f, \varphi) = R(\psi, f_\sharp\xi_0) + \int \varphi d(f_\sharp\xi_0) - J^\star_{\nu_0}(\varphi).$$

The difference with Section 5.1 is that (17) does not assume the convexity for $\mathcal{G}(\cdot, f, \varphi)$ and $\mathcal{G}(\psi, \cdot, \varphi)$. Thus, since there may not exist a Nash equilibrium point for problem (17) in general, it is difficult to prove that the sequence obtained by some gradient descent converges to the optimal minimax solution.

We show that *the sequence obtained by a certain gradient descent converges to a stationary point of (17) under appropriate assumptions.*

Throughout this section, let $\langle \cdot, \cdot \rangle_{S_1}$, $\langle \cdot, \cdot \rangle_{S''}$, and $\langle \cdot, \cdot \rangle_{S_2}$ be inner products in $S_1$, $S''$, and $S_2$, respectively. We denote $\| \cdot \|_{S_1}$, $\| \cdot \|_{S''}$, and $\| \cdot \|_{S_2}$ by norms induced by thier inner products.

First, we put the following assumption for $S_1$, $S''$, and $S_2$.

**Assumption 5.8.** *Assume that $S_1$, $S''$, and $S_2$ are closed subspace with respect to norms $\| \cdot \|_{S_1}$, $\| \cdot \|_{S''}$, and $\| \cdot \|_{S_2}$ in $\mathcal{C}(X)$, $\mathcal{C}(Z; X)$, and $\mathcal{C}(X)$, respectively.*

Under Assumption 5.8, $S_1$, $S''$, and $S_2$ are Hilbert spaces equipped with inner products $\langle \cdot, \cdot \rangle_{S_1}$, $\langle \cdot, \cdot \rangle_{S''}$, and $\langle \cdot, \cdot \rangle_{S_2}$, respectively. This assumption implies that convex subsets $S''_c$, $S_{1,c}$, and $S_{2,c}$ in continuous function spaces (where gradient descent updates are actually performed) must be contained in Hilbert spaces $S''$, $S_1$, and $S_2$.

Next, we put the assumption about the $\beta$-strongly concavity.

**Assumption 5.9.** *Let $\beta > 0$. Assume that for each $\psi \in S_{1,c}$ and $f \in S_c''$, $\mathcal{G}(\psi, f, \cdot)$ is $\beta$-strongly concave with respect to $\| \cdot \|_{S_2}$ over $S_{2,c}$.*

Note that this assumption is related to the gradient penalties (Gulrajani et al., 2017), widely-used as the stabilization techniques in adversarial training, as detailed in Section 6.2.1.

Under this assumption, we can define for $\psi \in S_{1,c}$ and $f \in S_c''$,

$$
\begin{aligned}
\Phi(\psi, f) &:= \underset{\varphi \in S_{2,c}}{\operatorname{argmax}} \, \mathcal{G}(\psi, f, \varphi), \\
G(\psi, f) &:= \max_{\varphi \in S_{2,c}} \mathcal{G}(\psi, f, \varphi) = \mathcal{G}(\psi, f, \Phi(f, \psi)).
\end{aligned}
\tag{18}
$$

Hereby, the minimax problem (17) is equivalent to minimization of (18) under Assumption 5.9.

Then, we put the following assumption related to the Gâteaux differentials, which is associated with the spectral normalization (Miyato et al., 2018) widely-used as stabilization techniques for GANs.

**Assumption 5.10.** *Assume that, for each $\psi \in S_{1,c}$, $f \in S_c''$, and $\varphi \in S_{2,c}$, there exist Gâteaux differentials $d\mathcal{G}(\cdot, f, \varphi)_\psi$, $d\mathcal{G}(\psi, \cdot, \varphi)_f$, and $d\mathcal{G}(\psi, f, \cdot)_\varphi$ of $\mathcal{G}(\cdot, f, \varphi)$, $\mathcal{G}(\psi, \cdot, \varphi)$, and $\mathcal{G}(\psi, f, \cdot)$ at $\psi \in S_{1,c}$, $f \in S_c''$, and $\varphi \in S_{2,c}$, respectively.*

Under Assumptions 5.8 and 5.10, Gâteaux differentials $d\mathcal{G}(\cdot, f, \varphi)_\psi : S_1 \to \overline{\mathbb{R}}$, $d\mathcal{G}(\psi, \cdot, \varphi)_f : S' \to \overline{\mathbb{R}}$, and $d\mathcal{G}(\psi, f, \cdot)_\varphi : S_2 \to \overline{\mathbb{R}}$ are identified with some elements in Hilbert spaces $S_1$, $S''$, and $S_2$, referred as $\nabla\mathcal{G}(\cdot, f, \varphi)_\psi \in S_1$, $\nabla\mathcal{G}(\psi, \cdot, \varphi)_f \in S''$, and $\nabla\mathcal{G}(\psi, f, \cdot)_\varphi \in S_2$, respectively. Furthermore, by Riesz representation theorem, we have the following:

$$
\begin{aligned}
d\mathcal{G}(\cdot, f, \varphi)_\psi &= \langle \nabla\mathcal{G}(\cdot, f, \varphi)_\psi, \cdot \rangle_{S_1}, & \|d\mathcal{G}(\cdot, f, \varphi)_\psi\|_{S_1}^\star &= \|\nabla\mathcal{G}(\cdot, f, \varphi)_\psi\|_{S_1}, \\
d\mathcal{G}(\psi, \cdot, \varphi)_f &= \langle \nabla\mathcal{G}(\psi, \cdot, \varphi)_f, \cdot \rangle_{S''}, & \|d\mathcal{G}(\psi, \cdot, \varphi)_f\|_{S''}^\star &= \|\nabla\mathcal{G}(\psi, \cdot, \varphi)_f\|_{S''}, \\
d\mathcal{G}(\psi, f, \cdot)_\varphi &= \langle \nabla\mathcal{G}(\psi, f, \cdot)_\varphi, \cdot \rangle_{S_2}, & \|d\mathcal{G}(\psi, f, \cdot)_\varphi\|_{S_2}^\star &= \|\nabla\mathcal{G}(\psi, f, \cdot)_\varphi\|_{S_2}.
\end{aligned}
$$

In addition to these assumptions, we assume the $L$-smoothness of $\mathcal{G}$ for each variable $\psi$, $f$, and $\varphi$ to show the convergence to a stationary point.

**Assumption 5.11.** *Let $L > 0$. Then, we assume the following: for $\psi, \psi_1, \psi_2 \in S_1$, $f, f_1, f_2 \in S''$, and $\varphi, \varphi_1, \varphi_2 \in S_2$,*

$$
\begin{aligned}
(a) &: \quad \|\nabla\mathcal{G}(\cdot, f, \varphi)_{\psi_1} - \nabla\mathcal{G}(\cdot, f, \varphi)_{\psi_2}\|_{S_1} \le L\|\psi_1 - \psi_2\|_{S_1}, \\
(b) &: \quad \|\nabla\mathcal{G}(\cdot, f_1, \varphi)_\psi - \nabla\mathcal{G}(\cdot, f_2, \varphi)_\psi\|_{S_1} \le L\|f_1 - f_2\|_{S''}, \\
(c) &: \quad \|\nabla\mathcal{G}(\cdot, f, \varphi_1)_\psi - \nabla\mathcal{G}(\cdot, f, \varphi_1)_\psi\|_{S_1} \le L\|\varphi_1 - \varphi_2\|_{S_2}, \\
(d) &: \quad \|\nabla\mathcal{G}(\psi, \cdot, \varphi)_{f_1} - \nabla\mathcal{G}(\psi, \cdot, \varphi)_{f_2}\|_{S''} \le L\|f_1 - f_2\|_{S''}, \\
(e) &: \quad \|\nabla\mathcal{G}(\psi_1, \cdot, \varphi)_f - \nabla\mathcal{G}(\psi_2, \cdot, \varphi)_f\|_{S''} \le L\|\psi_1 - \psi_2\|_{S_1}, \\
(f) &: \quad \|\nabla\mathcal{G}(\psi, \cdot, \varphi_1)_f - \nabla\mathcal{G}(\psi, \cdot, \varphi_2)_f\|_{S''} \le L\|\varphi_1 - \varphi_2\|_{S_2}, \\
(g) &: \quad \|\nabla\mathcal{G}(\psi, f, \cdot)_{\varphi_1} - \nabla\mathcal{G}(\psi, f, \cdot)_{\varphi_2}\|_{S_2} \le L\|\varphi_1 - \varphi_2\|_{S_2}, \\
(h) &: \quad \|\nabla\mathcal{G}(\psi_1, f, \cdot)_\varphi - \nabla\mathcal{G}(\psi_2, f, \cdot)_\varphi\|_{S_2} \le L\|\psi_1 - \psi_2\|_{S_1}, \\
(i) &: \quad \|\nabla\mathcal{G}(\psi, f_1, \cdot)_\varphi - \nabla\mathcal{G}(\psi, f_2, \cdot)_\varphi\|_{S_2} \le L\|f_1 - f_2\|_{S''}.
\end{aligned}
\tag{19}
$$

Here, we define the projected gradient descent for solving minimax optimization problem (17).

**Definition 5.12.** *Let $\psi_0 \in S_{1,c}$, $f_0 \in S_c''$ $\varphi_0 \in S_{2,c}$ be initial guesses. Then, we define the projected gradient descent $\{(\psi_n, f_n, \varphi_n)\}_{n \in \mathbb{N}_0} \subset S_{1,c} \times S_c'' \times S_{2,c}$ by*

$$\widetilde{\psi}_{n+1} = \underset{\psi \in S_1}{\operatorname{argmin}} \left\{ d\mathcal{G}(\cdot, f_n, \varphi_n)_{\psi_n}(\psi - \psi_n) + \frac{1}{2\alpha_{\psi,n}} \|\psi - \psi_n\|_{S_1}^2 \right\},$$

$$\psi_{n+1} = \mathcal{P}_{S_{1,c}}(\widetilde{\psi}_{n+1}),$$

$$\widetilde{f}_{n+1} = \underset{f \in S''}{\operatorname{argmin}} \left\{ d\mathcal{G}(\psi_n, \cdot, \varphi_n)_{f_n}(f - f_n) + \frac{1}{2\alpha_{f,n}} \|\mu - \mu_n\|_{S''}^2 \right\},$$

$$f_{n+1} = \mathcal{P}_{S_c''}(\widetilde{f}_{n+1}),$$

$$\widetilde{\varphi}_{n+1} = \underset{\varphi \in S_2}{\operatorname{argmax}} \left\{ d\mathcal{G}(\psi_n, f_n, \cdot)_{\varphi_n}(\varphi - \varphi_n) - \frac{1}{2\alpha_{\varphi,n}} \|\varphi - \varphi_n\|_{S_2}^2 \right\},$$

$$\varphi_{n+1} = \mathcal{P}_{S_{2,c}}(\widetilde{\varphi}_{n+1})$$

*where $\mathcal{P}_{S_{1,c}}$, $\mathcal{P}_{S_c''}$, and $\mathcal{P}_{S_{2,c}}$ are projection operators on $S_{1,c}$, $S_c''$, and $S_{2,c}$, and $\alpha_{\psi,n} > 0$, $\alpha_{f,n} > 0$, and $\alpha_{\varphi,n} > 0$ are step sizes.*

Remark that, using Riesz representation theorem under Assumption 5.8, the above update rule is equivalent to the following:

$$\psi_{n+1} = \mathcal{P}_{S_{1,c}}\left(\psi_n - \alpha_{\psi,n} \nabla\mathcal{G}(\cdot, f_n, \varphi_n)_{\psi_n}\right),$$
$$f_{n+1} = \mathcal{P}_{S_c''}\left(f_n - \alpha_{f,n} \nabla\mathcal{G}(\psi_n, \cdot, \varphi_n)_{f_n}\right),$$
$$\varphi_{n+1} = \mathcal{P}_{S_{2,c}}\left(\varphi_n + \alpha_{\varphi,n} \nabla\mathcal{G}(\psi_n, f_n, \cdot)_{\varphi_n}\right).$$

We also assume small step sizes to show the convergence of the gradient descent algorithm as follows:

**Assumption 5.13.** *Assume that there exists $C_0, C > 0$ and $C_\psi, C_f, \gamma \in (0,1)$ such that for all $n \in \mathbb{N}_0$,*

*(i)* $C_0 < \alpha_{\varphi,n} < \min\left(\frac{1}{L}, \frac{1}{\beta}\right),$

*(ii)* $L + L^2 L_\beta \alpha_{\psi,n}^2 + L^3(1+L)(1+L_\beta)\alpha_{\psi,n}^2 + L^2 L_\beta \alpha_{f,n}^2 + \frac{L^2(1+L_\beta)\alpha_{f,n}^2}{2} \le C,$

*(iii)* $(1 - \beta^2 \alpha_{\varphi,n-1}^2) + \frac{2L^2}{\beta^2}\left(1 + \frac{1}{\beta C_0}\right)(\alpha_{\psi,n-1}^2 + \alpha_{f,n-1}^2) \le \gamma,$

*(iv)* $C_\psi \le 1 - \frac{L\alpha_{\psi,n}}{2} - L(1+L)(1+L_\beta)\alpha_{\psi,n} - L_\beta\alpha_{\psi,n} - \frac{2L^2 C\alpha_{\psi,n}}{\beta^2(1-\gamma)}\left(1 + \frac{1}{\beta C_0}\right),$

*(v)* $C_f \le 1 - \frac{L\alpha_{f,n}}{2} - \frac{L(1+L_\beta)\alpha_{f,n}}{2} - L_\beta\alpha_{f,n} - \frac{2L^2 C\alpha_{f,n}}{\beta^2(1-\gamma)}\left(1 + \frac{1}{\beta C_0}\right),$

*where we denote by $L_\beta := L\left(\frac{L}{\beta} + 1\right)$.*

These assumptions imposes small step sizes $\alpha_{\psi,n}, \alpha_{f,n}, \alpha_{\varphi,n}$, depending on constants $L$ and $\beta$. Roughly speaking, Assumption 5.13 requires that step sizes be chosen small enough depending on $\beta$ and $L$. The followings are the details : First, we need to choose small $\alpha_{\varphi,n}$ satisfying (i). However, $\alpha_{\varphi,n}$ should not converge to zero as $n \to \infty$. Therefore, its lower bound is set to $C_0 > 0$. Next, we need to choose small $\alpha_{\psi,n}$ and $\alpha_{f,n}$ satisfying (ii)–(v). Note that the smallness of $\alpha_{\psi,n}$ and $\alpha_{f,n}$ depends on lower bound $C_0 > 0$ (see (iii)–(v)). In other words, $\alpha_{\psi,n}$ and $\alpha_{f,n}$ should be chosen to be smaller than $\alpha_{\varphi,n}$. Similar assumptions are often made in the context of nonconvex-concave minimax problems, as observed in works such as Huang et al. (2021); Lin et al. (2020a).

Building upon the background established above, we are ready to present our main theorem of this section:

**Theorem 5.14.** *Let Assumptions 5.8, 5.9, 5.10, 5.11, and 5.13 hold. Let $\{(\psi_n, f_n, \varphi_n)\}_{n \in \mathbb{N}_0} \subset S_{1,c} \times S_c'' \times S_{2,c}$ be the projected gradient descent defined in Definition 5.12. Then, for $N \in \mathbb{N}$, we have*

$$\left\| \widehat{\nabla G}_{\psi,N} \right\|_{S_1} \leq \widehat{C} \left( \sum_{n=0}^{N-1} \alpha_{\psi,n} \right)^{-1/2}, \tag{20}$$

$$\left\| \widehat{\nabla G}_{f,N} \right\|_{S''} \leq \widehat{C} \left( \sum_{n=0}^{N-1} \alpha_{f,n} \right)^{-1/2}, \tag{21}$$

*where*

$$\widehat{C} = \left( G(\psi_0, f_0) - \inf_{(\psi,f) \in S_{1,c} \times S_c''} G(\psi, f) + \frac{C \| \Phi(\psi_0, f_0) - \varphi_0 \|_{S_1}^2}{1 - \gamma} \right)^{1/2}.$$

*Here, $\widehat{\nabla G}_{\psi,N}$ and $\widehat{\nabla G}_{f,N}$ are weighted averages given by*

$$\widehat{\nabla G}_{\psi,N} := \frac{\sum_{n=0}^{N-1} \alpha_{\psi,n} \nabla G(\cdot, f_n)_{\psi_n}}{\sum_{n=0}^{N-1} \alpha_{\psi,n}}, \quad \widehat{\nabla G}_{f,N} := \frac{\sum_{n=0}^{N-1} \alpha_{f,n} \nabla G(\psi_n, \cdot)_{f_n}}{\sum_{n=0}^{N-1} \alpha_{f,n}}. \tag{22}$$

*Proof.* See Appendix B. □

The idea of the proof is to generalize Lin et al. (2020a, Theorem 4.4), which studied the convergence of the nonconvex-concave minimax problem in the finite dimensional setting, to the infinite dimensional function spaces, and to generalize two variables to three variables. Note that if step sizes are chosen as constants satisfying Assumption 5.13, then the right-hand sides of (20) and (21) are expressed as

$$\text{RHS of (20) and (21)} = O(1/\sqrt{N})$$

which converges to zero as $N \to \infty$. In other words, we have proved that the gradient decent defined by Definition 5.12 converges to a stationary point. Finally, we note that the order $\mathcal{O}(1/\sqrt{N})$ agrees with the result obtain by Lin et al. (2020a, Theorem 4.4), though we have adopted the infinite dimensional setting.

# 6 Examples of Relationship Between Objective Functions for GANs and UDAs and Assumptions for Convergence

In this section, we confirm that certain objective functions of GANs and UDAs fulfill the conditions for guaranteed convergence described in Section 5. In Section 6.1, we will verify Assumptions 5.1 (i) and 5.4 (i) & (iii) for the problem (8), and in Section 6.2 we will verify Assumptions 5.9 and 5.10 for the problem (10) because we can immediately confirm that the remaining assumptions hold for our objective function.

## 6.1 Convex-concave setting

In this section, we will verify Assumptions 5.1 (i) and 5.4 (i) and (iii) for the convex-concave setting (8).

### 6.1.1 Assumption 5.1 (i) (Joint convexity of $(\psi, \mu) \mapsto R(\psi, \mu)$)

Assumption 5.1 requires the joint convexity of a source risk $R(\psi, \mu) = \int \ell(\psi, \psi_0) d\mu$ with respect to $(\psi, \mu) \in \mathcal{C}(X) \times \mathcal{P}(X)$ for the existence of a minimax solution. Here, $\mu$ is a marginal distribution of a source domain, $\psi$ is a predictor to be optimized for a task (which may be implemented as a neural network), and $\psi_0$ is the true predictor for a task. Also, $\ell(\psi, \psi_0)$ denotes a loss function for a task in the source domain. In general, it is obvious that the source risk does not possess the joint convexity. Therefore, we need to introduce some regularization terms to the source risk such as

$$R(\psi, \mu) = \int \ell(\psi, \psi_0) d\mu + V(\psi) + W(\mu), \tag{23}$$

where $V : \mathcal{C}(X) \to \mathbb{R}$ and $W : \mathcal{P}(X) \to \mathbb{R}$ are regularization terms. The next proposition gives the sufficient conditions of the joint convexity for the source risk:

**Proposition 6.1.** *Let* $\|\cdot\|_{\mathcal{C}(X),1}$ *and* $\|\cdot\|_{\mathcal{C}(X),2}$ *be norms in* $\mathcal{C}(X)$. *Let* $\rho, \gamma > 0$ *with* $\gamma \geq \rho$. *Let be a loss function* $\ell : \mathcal{C}(X) \times \mathcal{C}(X) \to \mathbb{R}$. *Assume that*

(i) $\psi \mapsto \ell(\psi, \psi_0)$ *is convex for each* $\psi_0 \in \mathcal{C}(X)$.

(ii) $\psi \mapsto \ell(\psi, \psi_0)$ *is* $\rho$-*Lipschitz with respect to* $\|\cdot\|_{\mathcal{C}(X),1}$ *and* $\|\cdot\|_{\mathcal{C}(X),2}$ *for any* $\psi_0 \in \mathcal{C}(X)$, *that is,*

$$\|\ell(\psi_1, \psi_0) - \ell(\psi_2, \psi_0)\|_{\mathcal{C}(X),1} \leq \rho \|\psi_1 - \psi_2\|_{\mathcal{C}(X),2}, \ \psi_1, \psi_2 \in \mathcal{C}(X).$$

(iii) $V$ *and* $W$ *are* $\gamma$-*strongly convex with respect to* $\|\cdot\|_{\mathcal{C}(X),2}$ *and* $\|\cdot\|_{\mathcal{C}(X),1}^{\star}$, *respectively.*

*Then, the source risk* $R(\psi, \mu)$ *is joint convex with respect to* $(\psi, \mu)$.

*Proof.* See Appendix C.1 for the proof. $\qquad\square$

Assumption (i) and (ii) are the convexity and Lipschitz continuity for the loss function. For example, the squared error loss satisfies these assumptions. The example for (iii) is that $V = \frac{1}{2} \|\cdot\|_{\mathcal{H}_2}^2$ and $W = \frac{1}{2} \|\cdot\|_{\mathcal{H}_1}^{\star 2}$ where $(\mathcal{H}_i, \|\cdot\|_{\mathcal{H}_i})$ is a reproducing kernel Hilbert space (RKHS) with a positive definite kernel $K_i : X \times X \to \mathbb{R}$. Note that the dual norm $\|\cdot\|_{\mathcal{H}_1}^{\star}$ of the RKHS norm $\|\cdot\|_{\mathcal{H}_1}$ corresponds to a maximal mean discrepancy (MMD). As both $\|\cdot\|_{\mathcal{H}_2}$ and $\|\cdot\|_{\mathcal{H}_1}^{\star}$ are norms induced by inner products, $V$ and $W$ are 1-strongly convex with respect to $\|\cdot\|_{\mathcal{H}_2}$ and $\|\cdot\|_{\mathcal{H}_1}^{\star}$, respectively.

### 6.1.2 Assumption 5.4 (i) (Smoothness of $\psi \mapsto \int \ell(\psi, \psi_0) d\mu$)

Assumption 5.4(i) demands that the source risk $R(\psi, \mu) = \int \ell(\psi, \psi_0) d\mu$ is $L$-smooth for $\psi \in \mathcal{C}(X)$. Let us consider the following general functional $I_{h,\mu} : \mathcal{C}(X) \to \overline{\mathbb{R}}$ for the later convenience:

$$I_{h,\mu}(\Psi) := \int h(\Psi(x)) d\mu(x), \ \Psi \in \mathcal{C}(X),$$

where $h : \mathbb{R} \to \mathbb{R}$ and $\mu \in \mathcal{P}(X)$. Then, the next lemma guarantees $L$-smoothness of $I_{h,\mu}(\Psi)$.

**Lemma 6.2.** *Let be* $a, b \in [-\infty, \infty]$ *and* $h \in \mathcal{C}^1(a, b)$. *Then, we denote*

$$S_{\mathcal{C},a,b} := \{\psi \in \mathcal{C}(X) \ : \ a \leq \psi(x) \leq b, \ x \in X\}.$$

*Assume that* $h : (a, b) \to \mathbb{R}$ *is* $L$-*smooth, that is,*

$$D_h(s|t) \leq \frac{L}{2}|s - t|^2, \ s, t \in (a, b),$$

*where* $D_h(s|t) = h(s) - h(t) - h'(t)(s - t)$. *Then,* $I_{h,\mu}$ *is* $L$-*smooth with respect to* $\|\cdot\|_{L^2(X,\mu)}$ *over* $S_{\mathcal{C},a,b}$.

*Proof.* See Appendix C.2 for the proof. $\qquad\square$

If loss function $\ell(\cdot, \psi_0)$ is $L$-smooth (e.g., squared error loss), then by Lemma 6.2, $\psi \mapsto R(\psi, \mu) = \int \ell(\psi, \psi_0) d\mu$ is $L$-smooth with respect to $\|\cdot\|_{L^2(X,\mu)}$.

### 6.1.3   Assumption 5.4 (iii) (Smoothness of $\varphi \mapsto J_{\nu_0}^\star(\varphi)$)

Assumption 5.4 (iii) imposes the $L$-smoothness condition on the convex conjugate $J_{\nu_0}^\star(\varphi)$ of the discrepancy measure $J_{\nu_0}(\mu)$ to ensure convergence for both GAN and UDA. The representative discrepancy measures are $f$-divergence (Ali and Silvey, 1966; Csiszár, 1967) and integral probability metric (IPM) (Müller, 1997), which respectively unify different divergences between probability measures with various applications such as GAN and UDA. The $f$-divergence includes Kullback-Liebler divergence, Jensen-Shannon divergence, and Pearson $\chi^2$ divergence, while the IPM includes Wasserstein-1 distance, Dudley metric, and maximum mean discrepancy.

In the subsequent, we provide several examples of $J_{\nu_0}^\star$ that satisfy the $L$-smoothness for (A) $f$-divergence and (B) IPM.

**(A) $f$-divergence**   Let $f : \mathrm{dom}_f \subset \mathbb{R}_+ \to \mathbb{R}$ be a proper, lower semi-continuous and convex function. Then, the $f$-divergence $D_f(\mu|\nu)$ between $\mu \in \mathcal{P}(X)$ and $\nu \in \mathcal{P}(X)$ is defined as

$$D_f(\mu|\nu) := \begin{cases} \int f\left(\frac{d\mu}{d\nu}\right) d\nu & \text{if } \mu \ll \nu \\ +\infty & \text{otherwise} \end{cases}, \tag{24}$$

where $\mu \ll \nu$ denotes that $\mu$ is absolutely continuous with respect to $\nu$.

The $f$-divergence is joint convex with respect to $(\mu,\nu)$ (as the mapping $(p,q) \mapsto qf(p/q)$ is joint convex) and non-negative for all $\mu$ and $\nu$ but not symmetric with respect to $\mu$ and $\nu$ in general. In our case, we set $J_{f,\nu_0}(\mu) = D_f(\mu|\nu_0)$ with a fixed measure $\nu_0$ which implies a true distribution for GAN and a target distribution for UDA. The convergence theorem demands the $L$-smoothness for the convex conjugate $J_{f,\nu_0}^\star(\mu)$ of $J_{f,\nu_0}(\mu)$.

The following lemma provide the representation of the convex conjugate $J_{f,\nu_0}^\star(\varphi)$ of $J_{f,\nu_0}(\mu)$:

**Lemma 6.3.** *Assume that $f \in C^1(\mathrm{dom}_f)$, and there exists the inverse $(f')^{-1}$ of $f'$. Then, the convex conjugate $J_{f,\nu_0}^\star$ of $J_{f,\nu_0}$ is given by $J_{f,\nu_0}^\star(\varphi) = \int f^\star \circ \varphi \, d\nu_0$ for $\varphi \in S_{\mathcal{C},f} := \{\varphi \in \mathcal{C}(X) \; : \; \varphi(x) \in \mathrm{dom}_{(f')^{-1}}\}$, where*

$$f^\star(s) := \sup_t \{st - f(t)\} = s \cdot (f')^{-1}(s) - f \circ (f')^{-1}(s), \;\; s \in \mathrm{dom}_{(f')^{-1}}.$$

*Proof.* See Appendix C.3 for the proof. □

In the context of $f$-divergence, it is sufficient to confirm the smoothness of the convex conjugate $f^\star$ in the sense of the real function. We can take Jensen-Shannon divergence and Pearson $\chi^2$ divergence as examples and confirm that $J_{f,\nu_0}^\star(\mu)$ satisfies the $L$-smoothness.

**Example 6.4** (Jensen–Shannon divergence)**.** *The Jensen-Shannon divergence Lin (1991) is defined as*

$$D_{\mathrm{JS}}(\mu|\nu) := \frac{1}{2}D_{\mathrm{KL}}(\mu|\rho) + \frac{1}{2}D_{\mathrm{KL}}(\nu|\rho), \tag{25}$$

*where $\rho = (\mu+\nu)/2$, and $D_{\mathrm{KL}}(\mu|\nu)$ is the Kullback–Leibler divergence between $\mu$ and $\nu$ defined by*

$$D_{\mathrm{KL}}(\mu|\nu) = \int \frac{d\mu}{d\nu} \log \frac{d\mu}{d\nu} d\nu.$$

*Here, $f(t)$ is represented as*

$$f_{\mathrm{JS}}(t) := -\frac{1}{2}(t+1)\log\left(\frac{1+t}{2}\right) + \frac{1}{2}t\log t, \;\; t \in (0,\infty). \tag{26}$$

*The convex conjugate $f_{\mathrm{JS}}^\star(s)$ is*

$$f_{\mathrm{JS}}^\star(s) = -\frac{1}{2}\log(1 - \frac{1}{2}e^{2s}) - \frac{1}{2}\log 2, \quad s \in (-\infty, \frac{1}{2}\log 2). \tag{27}$$

*The convex conjugate $f_{\mathrm{JS}}^\star$ is $L$-smooth over $(a,b)$ with some $L > 0$, and $a, b \in (-\infty, \frac{1}{2}\log 2)$. Therefore, by using Lemma 6.2, $J_{f_{\mathrm{JS}},\nu_0}^\star$ is $L$-smooth with respect to $\|\cdot\|_{L^2(X,\mu)}$.*

**Example 6.5** (Pearson $\chi^2$ divergence). *The Pearson $\chi^2$ divergence is defined as*

$$D_{\chi^2}(\mu|\nu) := \int (\frac{d\mu}{d\nu} - 1)^2 d\nu.$$

*$f(t)$ is represented as*

$$f_{\mathrm{P}}(t) := (t-1)^2, \ t \in \mathbb{R}.$$

*The convex conjugate $f_{\mathrm{P}}^\star(s)$ is*

$$f_{\mathrm{P}}^\star(s) = \frac{1}{4}s^2 + s, \ s \in \mathbb{R}.$$

*The convex conjugate $f_P^\star$ is L-smooth over $(a,b)$ with some $L > 0$, and $a, b \in \mathbb{R}$. Therefore, by using Lemma 6.2, $J_{f_{\mathrm{P}},\nu_0}^\star$ is L-smooth with respect to $\|\cdot\|_{L^2(X,\mu)}$.*

**(B) Integral Probability Metric (IPM)** Let $\mathcal{F}$ be a class of real-valued bounded measurable functions on $X$. The IPM associated with $\mathcal{F}$ is defined as

$$d_{\mathcal{F}}(\mu,\nu) := \sup_{g \in \mathcal{F}} \left\{ \left| \int g d\mu - \int g d\nu \right| \right\} \tag{28}$$

for all pairs of measures $(\mu,\nu) \in \mathcal{P}(X) \times \mathcal{P}(X)$ such that all functions in $\mathcal{F}$ are absolutely $\mu$- and $\nu$-integrable. The typical examples are Wasserstein-1 distance for $\mathcal{F} = \{g \in \mathrm{Lip}(X) : \|g\|_{\mathrm{Lip}} \leq 1\}$ where $\mathrm{Lip}(X)$ is the class of the real-valued Lipschitz functions on $X$ and $\|\cdot\|_{\mathrm{Lip}}$ is the Lipschitz norm, and the MMD for $\mathcal{F} = \{g \in \mathcal{H} : \|g\|_{\mathcal{H}} \leq 1\}$ where $(\mathcal{H}, \|\cdot\|_{\mathcal{H}})$ is an RKHS with a positive definite kernel $K : X \times X \to \mathbb{R}$. In our case, we set $J_{\mathrm{IPM},\nu_0}(\mu) = d_{\mathcal{F}}(\mu,\nu_0)$ with a fixed measure $\nu_0$. Then, we can obtain the following Lemma.

**Lemma 6.6.** *Assume that $\mathcal{F}$ include the zero function. Then, the convex conjugate $J_{\mathrm{IPM},\nu_0}^\star$ of $J_{\mathrm{IPM},\nu_0}$ is given by*

$$J_{\mathrm{IPM},\nu_0}^\star(\varphi) = \int \varphi d\nu_0 + \chi\{\varphi \in \mathcal{F}\},$$

*where the indicator function is give by*

$$\chi\{A\} := \begin{cases} 0 & \text{if } A \text{ is true} \\ \infty & \text{if } A \text{ is false} \end{cases}. \tag{29}$$

*Proof.* See Appendix C.4 for the proof. $\qquad\square$

From Lemma 6.6, the differential $d(J_{\mathrm{IPM},\nu_0}^\star)_\varphi$ of $J_{\mathrm{IPM},\nu_0}^\star$ at $\varphi \in \mathcal{F}$ is given by

$$d(J_{\mathrm{IPM},\nu_0}^\star)_\varphi(\lambda) = J_{\mathrm{IPM},\nu_0}^\star(\lambda), \quad \lambda \in \mathcal{C}(X),$$

which implies that, by Definition 3.2

$$D_{J_{\mathrm{IPM},\nu_0}^\star}(\psi|\varphi) = 0, \quad \psi, \varphi \in \mathcal{F}.$$

Therefore, we obtain the following proposition.

**Proposition 6.7.** *For any $L > 0$, and any norm $\|\cdot\|_{\mathcal{C}(X)}$ induced by inner products, $J_{\mathrm{IPM},\nu_0}^\star$ is L-smooth with respect to $\|\cdot\|_{\mathcal{C}(X)}$ over $\mathcal{F}$.*

### 6.1.4 Examples of $\mathcal{K}(\psi, \mu, \varphi)$ simultaneously satisfying all assumptions in Section 5.1

In this section, we provide an example of our objective function (8) that simultaneously satisfies all the assumptions (Assumptions 5.1 and 5.4) for the convex-concave structure and smoothness.

Let $\psi_0 \in \mathcal{C}(X)$ be a true predictor, and let $\nu_0 \in \mathcal{P}(X)$ be a true distribution. We consider an objective function $\mathcal{K}_1 : S_1 \times S' \times S_2 \to \overline{\mathbb{R}}$ defined as

$$\mathcal{K}_1(\psi, \mu, \varphi) := \frac{1}{2} \int (\psi - \psi_0)^2 d\mu + \frac{\gamma}{2} \|\psi\|_{\mathcal{H}_{2\sqrt{2}\sigma}}^2 + \frac{\gamma}{2} \|\mu\|_{\mathcal{H}_\sigma}^{\star 2} + \int \varphi d\mu - \int k(\varphi) d\nu_0,$$

which corresponds to the problem (8) with $R(\psi, \mu) = \frac{1}{2}\int(\psi - \psi_0)^2 d\mu + \frac{\gamma}{2}\|\psi\|^2_{\mathcal{H}_{2\sqrt{2}\sigma}} + \frac{\gamma}{2}\|\mu\|^{\star 2}_{\mathcal{H}_\sigma}$ and $J_{\nu_0}$ is either IPMs or $f$-divergences. Here, $\gamma > 0$ is a regularization parameter, and $k : (a, b) \to \mathbb{R}$ is a convex and $C^1$-function with some $a, b \in \overline{\mathbb{R}}$, introduced to encompass more general situations including both IPMs and $f$-divergences. If the function $k$ takes the form $k(s) = s$, then the discrepancy measure $J_{\nu_0}$ corresponds to IPMs. If the function $k$ takes the form $k(s) = f^\star(s)$, then the discrepancy measure $J_{\nu_0}$ corresponds to $f$-divergences. Also, $(\mathcal{H}_\sigma, \|\cdot\|_{\mathcal{H}_\sigma})$ is a RKHS with Gaussian kernel $K_\sigma(x, y) = (2\pi\sigma^2)^{-d/2}e^{-|x-y|^2/2\sigma^2}$ with variance $\sigma^2$,

We choose convex subsets $S_1$, $S'$, and $S_2$ as

$$S_1 := \{\psi \in \mathcal{C}_0^\infty(X) : \|\partial_x^\alpha \psi\|_{L^\infty(\mathbb{R}^d)} \leq C_b \text{ for all } \alpha \in \mathbb{N}_0^d\},$$
$$S' := \{\mu \in \mathcal{P}(X) : \mu(A) \leq \mu_u(A) \text{ for all measurable sets } A \text{ in } X\},$$
$$S_2 := \{\varphi \in \mathcal{F} : a \leq \varphi(x) \leq b, \ x \in X\},$$

where $\mathcal{C}_0^\infty(X)$ is the space of $\mathcal{C}^\infty$ functions with compact support in $X$, $\mathcal{F}$ is a subset in $\mathcal{C}(X)$, and $\mu_u \in \mathcal{M}^+(X)$ is a non-negative measure. Then, Proposition 6.9 is obtained if the following assumption is satisfied.

**Assumption 6.8.** *We assume the following:*

- $\sigma < \frac{1}{2}$.

- $\gamma \geq 4C_b^2 C_\sigma^d$ where $C_\sigma := \sum_{j \in \mathbb{N}_0}(4\sigma^2)^j$.

- $\psi_0 \in S_1$ and $\nu_0 \in S'$.

- $k : (a, b) \to \mathbb{R}$ is $L_k$-smooth in the sense of the real function.

**Proposition 6.9.** *Let Assumption 6.8 hold. Then, the following statements hold:*

(1) *[Assumption 5.1 (i)]* $(\psi, \mu) \mapsto \mathcal{K}_1(\psi, \mu, \varphi)$ is convex.

(2) *[Assumption 5.1 (ii)]* $\varphi \mapsto \mathcal{K}_1(\psi, \mu, \varphi)$ is concave.

(3) *[Assumption 5.4 (i)]* $\psi \mapsto \mathcal{K}_1(\psi, \mu, \varphi)$ is 1-smooth with respect to

$$\left(\frac{1}{2}\|\cdot\|^2_{L^2(X, \mu_u)} + \frac{\gamma}{2}\|\cdot\|^2_{\mathcal{H}_{2\sqrt{2}\sigma}}\right)^{1/2}.$$

(4) *[Assumption 5.4 (ii)]* $\mu \mapsto \mathcal{K}_1(\psi, \mu, \varphi)$ is 1-smooth with respect to $\left(\frac{\gamma}{2}\right)^{1/2}\|\cdot\|^\star_{\mathcal{H}_\sigma}$.

(5) *[Assumption 5.4 (iii)]* $\varphi \mapsto -\mathcal{K}_1(\psi, \mu, \varphi)$ is $L_k$-smooth with respect to $\|\cdot\|_{L^2(X, \mu_u)}$.

*Proof.* See Appendix C.5. □

Note that we can immediately confirm that the remaining assumptions hold for our objective function. That is, Theorem 5.7 is satisfied by the setting of this section.

## 6.2 Nonconvex-concave setting

In this section, we will verify Assumptions 5.9 and 5.10 for the nonconvex-concave setting (10).

### 6.2.1 Assumption 5.9 (Strong convexity of $\varphi \mapsto J^{\star}_{\nu_0}(\varphi)$)

Assumption 5.9 requires the strong concavity for the problem (10) with respect to $\varphi$. This requirement is equivalent to the strong convexity of discrepancy measure $J_{\nu_0}$. We realize this by adding some regularization to the discrepancy measure.

Assume that the discrepancy measure $J_{\nu_0}$ is proper, lower semi-continuous, and convex. We define inf-convolution $J_{\nu_0} \oplus \mathcal{I}(\mu)$ as

$$J_{\nu_0} \oplus \mathcal{I}(\mu) := \inf_{\xi \in \mathcal{M}(X)} J_{\nu_0}(\xi) + \mathcal{I}(\mu - \xi)$$

where $\mathcal{I} : \mathcal{M}(X) \to \overline{\mathbb{R}}$ is a regularizer function, which is proper, lower semi-continuous, and convex. Then, the following Lemma holds.

**Lemma 6.10.** *Let $\beta > 0$ and $\| \cdot \|_{\mathcal{M}(X)}$ be a norm induced by an inner product in $\mathcal{M}(X)$. Then, if $\mathcal{I} : \mathcal{M}(X) \to \mathbb{R}$ is $(1/\beta)$-smooth with respect to $\| \cdot \|_{\mathcal{M}(X)}$, then $(J_{\nu_0} \oplus \mathcal{I})^{\star} : \mathcal{C}(X) \to \mathbb{R}$ is $\beta$-strongly convex with respect to $\| \cdot \|^{\star}_{\mathcal{M}(X)}$.*

*Proof.* See Appendix D.1 for the proof. $\qquad\square$

Thanks to this Lemma, we can attain the strong convexity by introducing the regularizer, such as the squared MMD in the RKHS with the Gaussian kernel, which corresponds to the gradient penalty (Gulrajani et al., 2017).

### 6.2.2 Assumption 5.10 (Gâteaux differentiability with respect to $f$)

Here, we consider the case when the source risk has the form $R(\psi, \mu) = \int \ell(\psi, \psi_0) d\mu$ where the loss function $\ell(\psi, \psi_0)$ is convex with respect to $\psi$. Then, the minimax problem (10) is translated into

$$\min_{\psi \in \mathcal{C}(X)} \min_{f \in \mathcal{C}(Z;X)} \max_{\varphi \in \mathcal{C}(X)} \int \ell(\psi \circ f, \psi_0 \circ f) d\mu_0 + \int \varphi \circ f d\mu_0 - J^{\star}_{\nu_0}(\varphi). \tag{30}$$

It is obvious that the above objective function is Gâteaux differentiable with respect to $\psi$ and $\varphi$ because the above objective function is convex and concave for $\psi$ and $\varphi$, respectively. As functions $\psi$ and $\varphi$ are composed with $f$, some regularity for $\psi$ and $\varphi$ is required to hold Gâteaux differentiable with respect to $f$.

Let us consider the following general functional $\mathcal{J}_{h,\xi} : \mathcal{C}(Z;X) \to \mathbb{R}$ :

$$\mathcal{J}_{h,\xi}(f) := \int h \circ f d\xi, \quad f \in \mathcal{C}(Z;X),$$

where $h \in \mathcal{C}(X)$ and $\xi \in \mathcal{P}(Z)$ are fixed. Then, the following lemma guarantees the Gâteaux differentiability of $\mathcal{J}_{h,\xi}$.

**Lemma 6.11.** *Assume that $h \in \mathrm{Lip}(X)$, $f \in \mathcal{C}(Z;X)$, and $\mu \ll m$ where $m$ is the Lebesgue measure. Then, $\mathcal{J}_{h,\xi}$ is Gâteaux differentiable at $f$. Furthermore, its Gâteaux differential is given by $d(\mathcal{J}_{h,\xi})_f$*

$$d(\mathcal{J}_{h,\xi})_f(g) = \int (\nabla h \circ f) \cdot g d\xi.$$

*Proof.* See Appendix D.2 for the proof. $\qquad\square$

Thanks to this Lemma, we can attain the Lipschitzness interpreted as applying the spectral normalization (Miyato et al., 2018), a widely-used stabilization technique for GANs.

### 6.2.3 Examples of $\mathcal{G}(\psi, f, \varphi)$ simultaneously satisfying all assumptions in Section 5.2

In this section, we provide an example of our objective function (10) to simultaneously satisfy all the assumptions (Assumptions 5.9, 5.10 and 5.11).

Let $\psi_0 \in \mathcal{C}(X)$ be a true predictor, $\nu_0 \in \mathcal{P}(X)$ be a true distribution, and $\beta > 0$. We then consider an objective function $\mathcal{G}_1 : S_1 \times S'' \times S_2 \to \overline{\mathbb{R}}$ defined as

$$\mathcal{G}_1(\psi, f, \varphi) := \frac{1}{2} \int (\psi \circ f - \psi_0 \circ f)^2 d\xi_0 + \int \varphi \circ f d\xi_0 - \int k(\varphi) d\nu_0 - \frac{\beta}{2} \|\varphi\|_{\mathcal{H}_\sigma}^2,$$

which corresponds to the problem (10) with $R(\psi, \mu) = \int (\psi - \psi_0)^2 d\mu$, and discrepancy measure $J_{\nu_0}$ is replaced with inf-convolution $J_{\nu_0} \oplus (\frac{1}{2\beta} \|\cdot\|_{\mathcal{H}_{\sigma^2}})$ where $J_{\nu_0}$ is either IPMs or $f$-divergences. Here, $(\mathcal{H}_\sigma, \|\cdot\|_{\mathcal{H}_\sigma})$ is a RKHS with Gaussian kernel $K_\sigma(x, y) = (2\pi\sigma^2)^{-d/2} e^{-|x-y|^2/2\sigma^2}$ with variance $\sigma^2$. In the same way of Section 6.1.4, we introduce $k : (a, b) \to \mathbb{R}$, which is a convex and $C^1$-function with some $a, b \in \overline{\mathbb{R}}$, to encompass more general situations including both IPMs and $f$-divergences.

We choose norms $\|\cdot\|_{S_1}$, $\|\cdot\|_{S''}$, and $\|\cdot\|_{S_2}$ as

$$\|\cdot\|_{S_1} := \|\cdot\|_{H^1(X)}, \quad \|\cdot\|_{S''} := \|\cdot\|_{L^2(Z;X,\xi_0)}, \quad \|\cdot\|_{S_2} := \|\cdot\|_{\mathcal{H}_\sigma}, \tag{31}$$

and subset $S_1$, $S_{1,c}$, $S''$, $S''_c$, $S_2$, and $S_{2,c}$ as

$$S_1 := \overline{\{\psi \in H^1(X) : \psi \text{ and } \nabla\psi \text{ are Lipschitz continuous}\}}^{\|\cdot\|_{S_1}},$$

$$S_{1,c} := \left\{ \psi \in S_1 : \mathrm{Lip}(\psi), \mathrm{Lip}(\nabla\psi) \leq C_1, \quad \sup_{x \in X} |\psi(x)|, \sup_{x \in X} |\nabla\psi(x)| \leq C_2 \right\},$$

$$S'' := \overline{\left\{ f \in \mathcal{C}(Z;X) : \|f\|_{L^2(Z;X,\xi_0)} < \infty, \ f_\sharp \xi_0 \ll m, \ \sup_x \left| \frac{d(f_\sharp \xi_0)}{dm} \right| < \infty \right\}}^{\|\cdot\|_{S''}},$$

$$S''_c := \left\{ f \in S'' : \sup_x \left| \frac{d(f_\sharp \xi_0)}{dm} \right| \leq C_3 \right\},$$

$$S_2 := \overline{\{\varphi \in \mathcal{C}_0^\infty(X) \cap \mathcal{H}_\sigma \cap \mathcal{F} : \varphi \text{ and } \nabla\varphi \text{ are Lipschitz continuous}\}}^{\|\cdot\|_{S_2}},$$

$$S_{2,c} := \left\{ \varphi \in S_2 : \mathrm{Lip}(\nabla\varphi) \leq C_4, \ a \leq \varphi(x) \leq b, \ x \in X \right\},$$

with some constants $C_1, C_2, C_3, C_4 > 0$, where $\mathrm{Lip}(\psi)$ is the Lipschitz constant for function $\psi$, and $\mathcal{F}$ is a subset in $\mathcal{C}(X)$. Then, Proposition 6.13 is obtained if the following assumption is satisfied.

**Assumption 6.12.** *We assume the following:*

- *$\psi_0 \in S_{1,c}$.*

- *The derivative $k'$ of $k$ is $L_k$-Lipschitz continuous.*

- *$\nu \ll m$, and $\sup_{x \in X} \left| \frac{d\nu_0}{dm}(x) \right| < \infty$.*

**Proposition 6.13.** *Let Assumption 6.12 hold. Then, the following statements hold:*

*(1) [Assumption 5.9] $\varphi \mapsto \mathcal{G}_1(\psi, f, \varphi)$ is $\beta$-strongly concave with respect to $\|\cdot\|_{S_2}$.*

*(2) [Assumption 5.10] $\mathcal{G}_1(\psi, f, \varphi)$ is Gâteaux differentiable for each variable.*

*(3) [Assumption 5.11] $\mathcal{G}_1(\psi, f, \varphi)$ satisfies the condition (19).*

*Proof.* See Appendix D.3 for the proof. $\square$

As well as Section 6.1.4, the remaining assumptions hold for our objective function. Therefore, Theorem 5.14 is satisfied by the setting of this section.

### 6.3 Interpretations of our analysis

Throughout Sections 6.1 and 6.2, we have verified that certain objective functions for ideal settings of GANs and UDAs satisfy the sufficient conditions for the convergences discussed in Section 5. Both objective functions for GANs and UDAs involve the discrepancy measure, and its convex conjugate need to be strongly convex and $L$-smooth.

An example for achieving strong convexity is through the inf-convolution with a discrepancy measure $J_{\nu_0}$ and a regularizer such as the squared MMD $\|\cdot\|_{\mathcal{H}_\sigma}^{\star 2}$ in the RKHS $\mathcal{H}_\sigma$ with Gaussian kernel $K_\sigma(x, y) = (2\pi\sigma^2)^{-d/2}e^{-|x-y|^2/2\sigma^2}$ having variance $\sigma^2$ (Lemma 6.10). The convex conjugate of this inf-convolution can be expressed as

$$(J_{\nu_0} \oplus \|\cdot\|_{\mathcal{H}_\sigma}^{\star 2})^\star(\varphi) = J_{\nu_0}^\star(\varphi) + \|\varphi\|_{\mathcal{H}_\sigma}^2.$$

Also, the RKHS norm $\|\varphi\|_{\mathcal{H}_\sigma}$ in this equation is represented as (Chu et al. (2020, Proposition 14))

$$\|\varphi\|_{\mathcal{H}_\sigma}^2 = \sum_{k=0}^\infty (\frac{1}{2}\sigma^2)^k \sum_{|\alpha|=k} \frac{1}{\alpha!} \|\partial_x^\alpha \varphi\|_{L^2}^2,$$

and minimizing this RKHS norm involves constraining the gradient of discriminator to be small. This can be interpreted as applying gradient penalties (Gulrajani et al., 2017), common stabilization techniques in adversarial training, to penalize gradients with large norm values. Note that the gradient penalty (Gulrajani et al., 2017) is a regularization technique to add the gradient norm $\mathbb{E}_{x\sim\mathbb{P}}[|\nabla\varphi(x) - 1|^2]$ to the discriminator's loss function.

On the other hand, when considering the discrepancy measure as IPMs, the convex conjugate of IPMs is given by

$$J_{\text{IPM},\nu_0}^\star(\varphi) = \int \varphi d\nu_0 + \chi\{\varphi \in \mathcal{F}\},$$

which is $L$-smoothness for $\varphi \in \mathcal{F}$ (Lemma 6.6). The function class $\mathcal{F}$ should be the subset of Lipschitz continuous function spaces $\text{Lip}(X)$ due to the Gâteaux differentiability of objective functions in the nonconvex-concave problem (17) (Lemma 6.11). The restriction of $\mathcal{F} \subset \text{Lip}(X)$ can be interpreted as applying the spectral normalization (Miyato et al., 2018), widely-used stabilization technique, to enforce the discriminator to be Lipschitz continuous. The spectral normalization (Miyato et al., 2018) is a normalization technique for weights of neural networks so that the Lipschitz norm $\|\varphi\|_{\text{Lip}}$ of the discriminator is bounded above by 1.

In addition, our sufficient conditions involve the joint convexity of the objective functions for UDAs. An example of achieving joint convexity is adding the strongly convex regularization to source risk, that is, considering $R$ as

$$R(\psi, \mu) = \int \ell(\psi, \psi_0) d\mu + \frac{1}{2} \|\psi\|_{\mathcal{H}_2}^2 + \frac{1}{2} \|\mu\|_{\mathcal{H}_1}^{\star 2}$$

which is joint convex with respect to $(\psi, \mu)$ (Proposition 6.1). Here, $(\mathcal{H}_i, \|\cdot\|_{\mathcal{H}_i})$ is a RKHS with a positive definite kernel $K_i : X \times X \to \mathbb{R}$. The RKHS norm is used as regularization for NNs, such as in Bietti et al. (2019).

## 7 Conclusion and future work

We provided the rigorous framework for the convergence analysis of the minimax problem in the infinite-dimensional spaces of continuous functions and probability measures. We discussed GANs and UDAs comprehensively and interpreted the assumptions for the convergences as stabilization techniques.

The following is the list of future work:

- It would be interesting to verify whether the $H$-divergence satisfies the assumption. In our current formulation, we utilized the convex conjugates of the divergence, implying that the classifier $\varphi : X \to \mathbb{R}$ belongs to continuous function space $\mathcal{C}(X)$. However, the formulation of the $H$-divergence (e.g.,

Ben-David et al. (2010; 2006)) employs the hypothesis spaces, where the corresponding classifier $\varphi : X \to \{0, 1\}$ is a labeling function. Therefore, integrating the $H$-divergence would need a different formulation.

- We employed gradient descent (GD), the simplest algorithm that suffices to achieve our aim. However, formulating the stochastic gradient descent (SGD) within the framework of our work would be an interesting future direction. Notably, Lin et al. (2020a) studied the convergence in finite-dimensional minimax problems using both GD and SGD. Our work in Section 5.2 extended the results from GD presented in Lin et al. (2020a) to infinite dimensional spaces. Similarly, we could generalize the results for SGD as well. In addition, sophisticated minimax algorithms to obtain better convergence rates have been proposed (see, e.g., Lin et al. (2020b)). It would be also interesting to formulate such algorithms in our framework.

- We optimized the distribution $\mu$ or the transport map $f$ as variables and fixed $\nu_0$ as the true distribution, which deviates from the actual setting of UDAs. In a more realistic scenario for UDAs, $\mu$ and $\nu_0$ would be formulated as $\mu = f_\sharp \xi_s$ and $\nu_0 = g_\sharp \xi_t$, where $f$ and $g$ are, for instance, neural networks optimizing their weight parameters, and $\xi_s$ and $\xi_t$ are fixed source and target distributions. However, this scenario is highly nonconvex-nonconcave, and this setting is more challenging to show convergence.

- It would be highly interesting to experimentally confirm how convergence properties in this study relate to the stability of GANs and UDAs. However, due to the fact that the optimization steps outlined in Definitions 5.5 and 5.12 entail proximal steps over infinite-dimensional spaces, practical implementations pose significant challenges.

## Acknowledgments

The research was jointly funded by AISIN and AISIN SOFTWARE. TF was partially supported by JSPS KAKENHI Grant Number JP24K16949.

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

# Appendix

| Notation | Meaning |
|---|---|
| $\mathbb{N}_0$ | the set of natural numbers including zero |
| $\overline{\mathbb{R}}$ | extended real number |
| $X$ | compact set in $\mathbb{R}^d$ |
| $Z$ | compact set in $\mathbb{R}^{d'}$ |
| $\mathcal{M}(X)$ | the set of all finite signed measures on $X$ |
| $\mathcal{M}^+(X)$ | the set of all non-negative finite measure on $X$ |
| $\mathcal{P}(X)$ | the set of Borel probability measures on $X$ |
| $\mathcal{C}(X)$ | the set of all continuous functions $X \to \mathbb{R}$ |
| $\mathcal{C}(Z;X)$ | the set of all continuous functions $Z \to X$ |
| $\lVert \cdot \rVert_{\mathcal{M}(X)}$ | norms induced by inner products in $\mathcal{M}(X)$ |
| $\lVert \cdot \rVert_{\mathcal{C}(X)}$ | norms induced by inner products in $\mathcal{C}(X)$ |
| $\lVert \cdot \rVert_{\mathcal{M}(X)}^{\star}$ | dual norm of $\lVert \cdot \rVert_{\mathcal{M}(X)}$ |
| $\lVert \cdot \rVert_{\mathcal{C}(X)}^{\star}$ | dual norm of $\lVert \cdot \rVert_{\mathcal{C}(X)}$ |
| $dF_\varphi$ | Gâteaux differentials of $F : \mathcal{C}(X) \to \overline{\mathbb{R}}$ at $\varphi \in \mathcal{C}(X)$ |
| $D_F$ | Bregman divergence associated with $F : \mathcal{C}(X) \to \overline{\mathbb{R}}$ |
| $R$ | source risk |
| $\nu_0$ | fixed target distribution |
| $\psi_0$ | true predictor |
| $J_{\nu_0}$ | discrepancy measure |
| $J_{\nu_0}^{\star}$ | convex conjugate of $J_{\nu_0}$ |
| $\mathcal{K}$ | convex-concave objective function in Section 5.1 |
| $S'$ | subset in $\mathcal{P}(X)$ (for source distribution $\mu$ in Section 5.1) |
| $S_1$ | subset in $\mathcal{C}(X)$ (for predictor $\psi$ in Section 5.1) |
| $S_2$ | subset in $\mathcal{C}(X)$ (for classifier $\varphi$ in Section 5.1) |
| $\alpha_n$ | step size in Definition 5.5 |
| $\mathcal{G}$ | nonconvex-concave objective function in Section 5.2 |
| $S_c''$ | convex subset in $\mathcal{C}(Z;X)$ (for generator $f$ in Section 5.2) |
| $S_{1,c}$ | convex subset in $\mathcal{C}(X)$ (for predictor $\psi$ in Section 5.2) |
| $S_{2,c}$ | convex subset in $\mathcal{C}(X)$ (for classifier $\varphi$ in Section 5.2) |
| $\alpha_{f,n}$ | step size (for generator $f$ in Section 5.2) in Definition 5.12 |
| $\alpha_{\psi,n}$ | step size (for predictor $\psi$ in Section 5.2) in Definition 5.12 |
| $\alpha_{\varphi,n}$ | step size (for classifier $\varphi$ in Section 5.2) in Definition 5.12. |
| $D_f$ | $f$-divergence |
| $\mathcal{F}$ | the class of real-valued bounded measurable functions on $X$ |
| $d_{\mathcal{F}}$ | integral probability metric associated with $\mathcal{F}$ |
| $\mathrm{Lip}(X)$ | the class of the real-valued Lipschitz functions on $X$ |
| $(\mathcal{H}, \lVert \cdot \rVert_{\mathcal{H}})$ | reproducing kernel Hilbert space |

Table 1: Table of Notations

## A  Proof of Theorem 5.7

Before the proof of Theorem 5.7, we review the three-point inequality.

**Three-point inequality**   The three-point inequality is a key ingredient for the proof of Theorem 5.7, which was first introduced by Chen and Teboulle (1993). We introduce the three-point inequality in the space of measures and in continuous function spaces without the proof. See Aubin-Frankowski et al. (2022) for the proof.

**Lemma A.1** (Three-point inequality for the space of measures). *Let $S_\mathcal{M} \subset \mathcal{M}(X)$, and let $G : \mathcal{M}(X) \to \overline{\mathbb{R}}$ be a proper, lower semi-continuous, and convex function. Let $\alpha_\mathcal{M} > 0$. For a given $\mu \in \mathcal{M}(X)$, let*

$$\overline{\nu} := \underset{\nu \in S_\mathcal{M}}{\operatorname{argmin}} \left\{ G(\nu) + \frac{1}{2\alpha_\mathcal{M}} \|\nu - \mu\|_{\mathcal{M}(X)}^2 \right\},$$

*where $\|\cdot\|_{\mathcal{M}(X)}$ is a norm induced by inner products in $\mathcal{M}(X)$. Then,*

$$G(\nu) + \frac{1}{2\alpha_\mathcal{M}} \|\nu - \mu\|_{\mathcal{M}(X)}^2 \geq G(\overline{\nu}) + \frac{1}{2\alpha_\mathcal{M}} \|\overline{\nu} - \mu\|_{\mathcal{M}(X)}^2 + \frac{1}{2\alpha_\mathcal{M}} \|\nu - \overline{\nu}\|_{\mathcal{M}(X)}^2 \quad \text{for all } \nu \in S_\mathcal{M}.$$

**Lemma A.2** (Three-point inequality for continuous function space). *Let $S_\mathcal{C} \subset \mathcal{C}(X)$, and let $F : \mathcal{C}(X) \to \overline{\mathbb{R}}$ be a proper, lower semi-continuous, and convex function. Let $\alpha_\mathcal{C} > 0$. For a given $f \in \mathcal{C}(X)$, let*

$$\overline{g} := \underset{g \in S_\mathcal{C}}{\operatorname{argmin}} \left\{ F(g) + \frac{1}{2\alpha_\mathcal{C}} \|g - f\|_{\mathcal{C}(X)}^2 \right\}.$$

*where $\|\cdot\|_{\mathcal{C}(X)}$ is a norm induced by inner products in $\mathcal{C}(X)$. Then,*

$$F(g) + \frac{1}{2\alpha_\mathcal{C}} \|g - f\|_{\mathcal{C}(X)}^2 \geq F(\overline{g}) + \frac{1}{2\alpha_\mathcal{C}} \|\overline{g} - f\|_{\mathcal{C}(X)}^2 + \frac{1}{2\alpha_\mathcal{C}} \|g - \overline{g}\|_{\mathcal{C}(X)}^2 \quad \text{for all } g \in S_\mathcal{C}.$$

The proof of Theorem 5.7 is essentially based on the $L$-smoothness of $\mathcal{K}(\psi, \mu, \varphi)$ for each variables and the three-point inequality in Lemma A.1 and A.2 associated with the update rules of the gradient descent in Definition 5.5.

*Proof of Theorem 5.7.* First, we evaluate the lower bound of $\mathcal{K}(\psi_n, \mu_n, \varphi_{n+1})$. The following holds for any $\varphi \in S_2$:

$$\mathcal{K}(\psi_n, \mu_n, \varphi_{n+1})$$
$$\geq \mathcal{K}(\psi_n, \mu_n, \varphi_n) + d\mathcal{K}(\psi_n, \mu_n, \cdot)_{\varphi_n}(\varphi_{n+1} - \varphi_n) - \frac{L}{2} \|\varphi_{n+1} - \varphi_n\|_{\mathcal{C}(X),2}^2$$
$$\geq \mathcal{K}(\psi_n, \mu_n, \varphi_n) + d\mathcal{K}(\psi_n, \mu_n, \cdot)_{\varphi_n}(\varphi_{n+1} - \varphi_n) - \frac{1}{2\alpha_n} \|\varphi_{n+1} - \varphi_n\|_{\mathcal{C}(X),2}^2$$
$$\geq \mathcal{K}(\psi_n, \mu_n, \varphi_n) + d\mathcal{K}(\psi_n, \mu_n, \cdot)_{\varphi_n}(\varphi - \varphi_n) - \frac{1}{2\alpha_n} (\|\varphi - \varphi_n\|_{\mathcal{C}(X),2}^2 - \|\varphi - \varphi_{n+1}\|_{\mathcal{C}(X),2}^2)$$
$$\geq \mathcal{K}(\psi_n, \mu_n, \varphi) - \frac{1}{2\alpha_n} (\|\varphi - \varphi_n\|_{\mathcal{C}(X),2}^2 - \|\varphi - \varphi_{n+1}\|_{\mathcal{C}(X),2}^2),$$

where the first inequality follows from the $L$-smoothness of $\varphi \mapsto \mathcal{K}(\psi, \mu, \varphi)$ , and the second inequality follows from $0 < \alpha_n \leq 1/L$, and the last inequality results from the concavity of $\varphi \to \mathcal{K}(\psi, \mu, \varphi)$ for each $\psi$ and $\mu$. Also, the third inequality follows from the three-point inequality in Lemma A.2 with $F(\varphi) = -d\mathcal{K}(\psi_n, \mu_n, \cdot)_{\varphi_n}(\varphi - \varphi_n)$ and $f = \varphi_n$:

$$d\mathcal{K}(\psi_n, \mu_n, \cdot)_{\varphi_n}(\varphi_{n+1} - \varphi_n) - \frac{1}{2\alpha_n} \|\varphi_{n+1} - \varphi_n\|_{\mathcal{C}(X),2}^2$$
$$\leq d\mathcal{K}(\psi_n, \mu_n, \cdot)_{\varphi_n}(\varphi - \varphi_n) - \frac{1}{2\alpha_n} (\|\varphi - \varphi_n\|_{\mathcal{C}(X),2}^2 - \|\varphi - \varphi_{n+1}\|_{\mathcal{C}(X),2}^2), \quad \varphi \in S_2.$$

Furthermore, by using the the concavity of $\varphi \mapsto \mathcal{K}(\psi_n, \mu_n, \varphi)$ for $\mathcal{K}(\psi_n, \mu_n, \varphi_{n+1})$ at the first line, we have

$$- \alpha_n \mathcal{K}(\psi_n, \mu_n, \varphi_n) + \alpha_n \mathcal{K}(\psi_n, \mu_n, \varphi)$$
$$\leq \frac{1}{2} (\|\varphi - \varphi_n\|_{\mathcal{C}(X),2}^2 - \|\varphi - \varphi_{n+1}\|_{\mathcal{C}(X),2}^2) + \alpha_n d\mathcal{K}(\psi_n, \mu_n, \cdot)_{\varphi_n}(\varphi_{n+1} - \varphi_n). \tag{32}$$

By taking the summation of (32) from $n = 0$ to $N - 1$, we obtain

$$-\sum_{n=0}^{N-1} \alpha_n \mathcal{K}(\psi_n, \mu_n, \varphi_n) + \left(\sum_{n=0}^{N-1} \alpha_n\right) \mathcal{K}(\widehat{\psi}_N, \widehat{\mu}_N, \varphi) \leq \frac{1}{2}\|\varphi - \varphi_0\|_{\mathcal{C}(X),2}^2 + C_{\mathcal{K}}^N. \tag{33}$$

Here, we used $\sum_{n=0}^{N-1} \alpha_n \mathcal{K}(\psi_n, \mu_n, \varphi) \geq \sum_{n=0}^{N-1} \alpha_n \mathcal{K}(\widehat{\psi}_N, \widehat{\mu}_N, \varphi)$ by Jensen's inequality , where the weighted sums $(\widehat{\psi}_N, \widehat{\mu}_N, \widehat{\varphi}_N)$ are defined by (16). Here, we introduced $C_{\mathcal{K}}^N$ defined as

$$C_{\mathcal{K}}^N := \sum_{n=0}^{N-1} \alpha_n d\mathcal{K}(\psi_n, \mu_n, \cdot)_{\varphi_n}(\varphi_{n+1} - \varphi_n) \geq 0. \tag{34}$$

The non-negativity follows from the update rule defined by Definition 5.5. When $\varphi = \widehat{\varphi}_N$ in (33), we have

$$-\sum_{n=0}^{N-1} \alpha_n \mathcal{K}(\psi_n, \mu_n, \varphi_n) + \left(\sum_{n=0}^{N-1} \alpha_n\right) \mathcal{K}(\widehat{\psi}_N, \widehat{\mu}_N, \widehat{\varphi}_N) \leq \frac{1}{2}\|\widehat{\varphi}_N - \varphi_0\|_{\mathcal{C}(X),2}^2 + C_{\mathcal{K}}^N. \tag{35}$$

When $\varphi = \varphi_*$ in (33), we have

$$-\sum_{n=0}^{N-1} \alpha_n \mathcal{K}(\psi_n, \mu_n, \varphi_n) + \left(\sum_{n=0}^{N-1} \alpha_n\right) \mathcal{K}(\psi_*, \mu_*, \varphi_*) \leq \frac{1}{2}\|\varphi_* - \varphi_0\|_{\mathcal{C}(X),2}^2 + C_{\mathcal{K}}^N, \tag{36}$$

where $(\psi_*, \mu_*, \varphi_*)$ is a saddle point defined at (12) satisfying $\mathcal{K}(\psi_*, \mu_*, \varphi_*) \leq \mathcal{K}(\widehat{\psi}_N, \widehat{\mu}_N, \varphi_*)$.

Second, we evaluate the upper bound of $\mathcal{K}(\psi_n, \mu_{n+1}, \varphi_n)$. The following holds for any $(\psi, \mu) \in S_{\mathcal{M}} \times S_{\mathcal{C}}$:

$$\mathcal{K}(\psi_n, \mu_{n+1}, \varphi_n)$$
$$\leq \mathcal{K}(\psi_n, \mu_n, \varphi_n) + d\mathcal{K}(\psi_n, \cdot, \varphi_n)_{\mu_n}(\mu_{n+1} - \mu_n) + \frac{L}{2}\|\mu_{n+1} - \mu_n\|_{\mathcal{M}(X)}^2$$
$$\leq \mathcal{K}(\psi_n, \mu_n, \varphi_n) + d\mathcal{K}(\psi_n, \cdot, \varphi_n)_{\mu_n}(\mu_{n+1} - \mu_n) + \frac{1}{2\alpha_n}\|\mu_{n+1} - \mu_n\|_{\mathcal{M}(X)}^2$$
$$\quad - d\mathcal{K}(\cdot, \mu_n, \varphi_n)_{\psi_n}(\psi_{n+1} - \psi_n)$$
$$\quad + d\mathcal{K}(\cdot, \mu_n, \varphi_n)_{\psi_n}(\psi_{n+1} - \psi_n) + \frac{1}{2\alpha_n}\|\psi_{n+1} - \psi_n\|_{\mathcal{C}(X),1}^2$$
$$\leq \mathcal{K}(\psi_n, \mu_n, \varphi_n) + d\mathcal{K}(\cdot, \mu_n, \varphi_n)_{\psi_n}(\psi - \psi_n) + d\mathcal{K}(\psi_n, \cdot, \varphi_n)_{\mu_n}(\mu - \mu_n)$$
$$\quad - d\mathcal{K}(\cdot, \mu_n, \varphi_n)_{\psi_n}(\psi_{n+1} - \psi_n)$$
$$\quad + \frac{1}{2\alpha_n}\left(\|\psi - \psi_n\|_{\mathcal{C}(X),1}^2 - \|\psi - \psi_{n+1}\|_{\mathcal{C}(X),1}^2 + \|\mu - \mu_n\|_{\mathcal{M}(X)}^2 - \|\mu - \mu_{n+1}\|_{\mathcal{M}(X)}^2\right)$$
$$\leq \mathcal{K}(\psi, \mu, \varphi_n) - d\mathcal{K}(\cdot, \mu_n, \varphi_n)_{\psi_n}(\psi_{n+1} - \psi_n)$$
$$\quad + \frac{1}{2\alpha_n}\left(\|\psi - \psi_n\|_{\mathcal{C}(X),1}^2 - \|\psi - \psi_{n+1}\|_{\mathcal{C}(X),1}^2 + \|\mu - \mu_n\|_{\mathcal{M}(X)}^2 - \|\mu - \mu_{n+1}\|_{\mathcal{M}(X)}^2\right),$$

where the first inequality follows from the $L$-smoothness of $\mu \mapsto \mathcal{K}(\psi, \mu, \varphi)$ for each $\psi$ and $\varphi$, the second inequality follows from $0 < \alpha_n \leq 1/L$, and the last inequality is the result of the joint convexity of $(\psi, \mu) \mapsto \mathcal{K}(\psi, \mu, \varphi)$ for any $\varphi$. Also, the third inequality results from the three-point inequality in Lemma A.1 with $G(\nu) = d\mathcal{K}(\psi_n, \cdot, \varphi_n)_{\mu_n}(\nu - \mu_n)$ and $\mu = \mu_n$,

$$d\mathcal{K}(\psi_n, \cdot, \varphi_n)_{\mu_n}(\mu_{n+1} - \mu_n) + \frac{1}{2\alpha_n}\|\mu_{n+1} - \mu_n\|_{\mathcal{M}(X)}^2$$
$$\leq d\mathcal{K}(\psi_n, \cdot, \varphi_n)_{\mu_n}(\mu - \mu_n) + \frac{1}{2\alpha_n}(\|\mu - \mu_n\|_{\mathcal{M}(X)}^2 - \|\mu - \mu_{n+1}\|_{\mathcal{M}(X)}^2), \quad \mu \in S',$$

and Lemma A.2 with $F(\psi) = d\mathcal{K}(\cdot, \mu_n, \varphi_n)_{\psi_n}(\psi - \psi_n)$ and $f = \psi_n$

$$dK(\cdot, \mu_n, \varphi_n)_{\psi_n}(\psi_{n+1} - \psi_n) + \frac{1}{2\alpha_n}\|\psi_{n+1} - \psi_n\|^2_{\mathcal{C}(X),1}$$
$$\leq dK(\cdot, \mu_n, \varphi_n)_{\psi_n}(\psi - \psi_n) + \frac{1}{2\alpha_n}\left(\|\psi - \psi_n\|^2_{\mathcal{C}(X),1} - \|\psi - \psi_{n+1}\|^2_{\mathcal{C}(X),1}\right), \quad \mu \in S_1.$$

Furthermore, by using the the convexity of $\mu \mapsto \mathcal{K}(\psi_n, \mu, \varphi_n)$ for $\mathcal{K}(\psi_n, \mu_{n+1}, \varphi_n)$ at the first line, we have

$$\alpha_n \mathcal{K}(\psi_n, \mu_n, \varphi_n) - \alpha_n \mathcal{K}(\psi, \mu, \varphi_n)$$
$$\leq \frac{1}{2}\left(\|\psi - \psi_n\|^2_{\mathcal{C}(X),1} - \|\psi - \psi_{n+1}\|^2_{\mathcal{C}(X),1} + \|\mu - \mu_n\|^2_{\mathcal{M}(X)} - \|\mu - \mu_{n+1}\|^2_{\mathcal{M}(X)}\right) \quad (37)$$
$$- \alpha_n dK(\cdot, \mu_n, \varphi_n)_{\psi_n}(\psi_{n+1} - \psi_n) - \alpha_n dK(\psi_n, \cdot, \varphi_n)_{\mu_n}(\mu_{n+1} - \mu_n).$$

We perform the summation of (37) over the interval $n = 0$ to $N - 1$:

$$\sum_{n=0}^{N-1} \alpha_n \mathcal{K}(\psi_n, \mu_n, \varphi_n) - \left(\sum_{n=0}^{N-1} \alpha_n\right)\mathcal{K}(\psi, \mu, \widehat{\varphi}_N) \leq \frac{1}{2}\left(\|\psi - \psi_0\|^2_{\mathcal{C}(X),1} + \|\mu - \mu_0\|^2_{\mathcal{M}(X)}\right) + \widetilde{C}^N_\mathcal{K}, \quad (38)$$

where we used $\sum_{n=0}^{N-1} \alpha_n \mathcal{K}(\psi, \mu, \varphi_n) \leq \left(\sum_{n=0}^{N-1} \alpha_n\right)\mathcal{K}(\psi, \mu, \widehat{\varphi}_N)$ following from the Jensen's inequality and we introduced $\widetilde{C}^N_\mathcal{K}$ defined as

$$\widetilde{C}^N_\mathcal{K} := -\sum_{n=0}^{N-1} \alpha_n \left(dK(\cdot, \mu_n, \varphi_n)_{\psi_n}(\psi_{n+1} - \psi_n) + dK(\psi_n, \cdot, \varphi_n)_{\mu_n}(\mu_{n+1} - \mu_n)\right) \geq 0. \quad (39)$$

The non-negativity follows from the update rule defined by Definition 5.5. By substituting $\psi = \widehat{\psi}_N$ and $\mu = \widehat{\mu}_N$ into (38), we have

$$\sum_{n=0}^{N-1} \alpha_n \mathcal{K}(\psi_n, \mu_n, \varphi_{n+1}) - \left(\sum_{n=0}^{N-1} \alpha_n\right)\mathcal{K}(\widehat{\psi}_N, \widehat{\mu}_N, \widehat{\varphi}_N)$$
$$\leq \frac{1}{2}\left(\|\widehat{\psi}_N - \psi_0\|^2_{\mathcal{C}(X),1} + \|\widehat{\mu}_N - \mu_0\|^2_{\mathcal{M}(X)}\right) + \widetilde{C}^N_\mathcal{K}. \quad (40)$$

Also, by taking $(\psi, \mu) = (\psi_*, \mu_*)$ in (38) which is a saddle point for the minimax solution for minimax problem $\mathcal{K}(\psi, \mu, \varphi)$ on $(S_1 \times S') \times S_2$ such that $(\psi_*, \mu_*, \varphi_*)$ satisfies the $\mathcal{K}(\psi_*, \mu_*, \widehat{\varphi}_N) \leq \mathcal{K}(\psi_*, \mu_*, \varphi_*)$, we have

$$\sum_{n=0}^{N-1} \alpha_n \mathcal{K}(\psi_n, \mu_n, \varphi_{n+1}) - \left(\sum_{n=0}^{N-1} \alpha_n\right)\mathcal{K}(\psi_*, \mu_*, \varphi_*)$$
$$\leq \frac{1}{2}\left(\|\psi_* - \psi_0\|^2_{\mathcal{C}(X),1} + \|\mu_* - \mu_0\|^2_{\mathcal{M}(X)}\right) + \widetilde{C}^N_\mathcal{K}. \quad (41)$$

Third, let combine all the results we have obtained. Summing up (36) and (40) yields

$$\left(\sum_{n=0}^{N-1} \alpha_n\right)(\mathcal{K}(\psi_*, \mu_*, \varphi_*) - \mathcal{K}(\widehat{\psi}_N, \widehat{\mu}_N, \widehat{\varphi}_N))$$
$$\leq \frac{1}{2}\left(\|\widehat{\psi}_N - \psi_0\|^2_{\mathcal{C}(X),1} + \|\widehat{\mu}_N - \mu_0\|^2_{\mathcal{M}(X)} + \|\varphi_* - \varphi_0\|^2_{\mathcal{C}(X),2}\right) + C^N_\mathcal{K} + \widetilde{C}^N_\mathcal{K}.$$

Also, summing up (35) and (41) gives

$$\left(\sum_{n=0}^{N-1} \alpha_n\right)(\mathcal{K}(\widehat{\psi}_N, \widehat{\mu}_N, \widehat{\varphi}_N) - \mathcal{K}(\psi_*, \mu_*, \varphi_*))$$
$$\leq \frac{1}{2}\left(\|\psi_* - \psi_0\|^2_{\mathcal{C}(X),1} + \|\mu_* - \mu_0\|_{\mathcal{M}(X)} + \|\widehat{\varphi}_N - \varphi_0\|^2_{\mathcal{C}(X),2}\right) + C^N_\mathcal{K} + \widetilde{C}^N_\mathcal{K}.$$

Therefore, we obtain

$$\left|\mathcal{K}(\widehat{\psi}_N, \widehat{\mu}_N, \widehat{\varphi}_N) - \mathcal{K}(\psi_*, \mu_*, \varphi_*)\right| \leq \left(\sum_{n=0}^{N-1} \alpha_n\right)^{-1} \left(\frac{1}{2} C_s + C_{\mathcal{K}}^N + \widetilde{C}_{\mathcal{K}}^N\right), \tag{42}$$

where $C_s > 0$ is a finite constant defined in (15). This means that the value of the object function at $(\widehat{\psi}_N, \widehat{\mu}_N, \widehat{\varphi}_N)$ approximately converges to a saddle point under the gradient descent update rule, if the sums $C_{\mathcal{K}}^N + \widetilde{C}_{\mathcal{K}}^N$ of the Gâteaux differentials in (34) and (39) are finite.

Finally, we prove that the term $C_{\mathcal{K}}^N + \widetilde{C}_{\mathcal{K}}^N$ defined in (34) and (39) are bounded from above by the norms $\|\mu_{n+1} - \mu_n\|_{\mathcal{M}(X)}$, $\|\psi_{n+1} - \psi_n\|_{\mathcal{C}(X),1}$ and $\|\varphi_{n+1} - \varphi_n\|_{\mathcal{C}(X),2}$ under Assumptions 5.2. Actually, each Gâteaux differential is bounded from above as follows:

$$-d\mathcal{K}(\cdot, \mu_n, \varphi_n)_{\psi_n}(\psi_{n+1} - \psi_n) = -\int (\psi_{n+1} - \psi_n) dN_{\psi_n, \mu_n, \varphi_n}$$
$$\leq \|\psi_{n+1} - \psi_n\|_{\mathcal{C}(X),1} \underbrace{\|N_{\psi_n, \mu_n, \varphi_n}\|_{\mathcal{C}(X),1}^{\star}}_{\leq B}, \tag{43}$$

and

$$-d\mathcal{K}(\psi_n, \cdot, \varphi_n)_{\mu_n}(\mu_{n+1} - \mu_n) = -\int \Phi_{\psi_n, \mu_n, \varphi_n} d(\mu_{n+1} - \mu_n)$$
$$\leq \underbrace{\|\Phi_{\psi_n, \mu_n, \varphi_n}\|_{\mathcal{M}(X)}^{\star}}_{\leq B} \|\mu_{n+1} - \mu_n\|_{\mathcal{M}(X)}, \tag{44}$$

and

$$d\mathcal{K}(\psi_n, \mu_n, \cdot)_{\varphi_n}(\varphi_{n+1} - \varphi_n) = \int \varphi_{n+1} - \varphi_n d\Lambda_{\psi_n, \mu_n, \varphi_n}$$
$$\leq \|\varphi_{n+1} - \varphi_n\|_{\mathcal{C}(X),2} \underbrace{\|\Lambda_{\psi_n, \mu_n, \varphi_n}\|_{\mathcal{C}(X),2}^{\star}}_{\leq B}. \tag{45}$$

Moreover, by taking into account that the gradient decent scheme in Definition 5.5 with $\psi = \psi_n$ implies $d\mathcal{K}(\cdot, \mu_n, \varphi_n)_{\psi_n}(\psi_{n+1} - \psi_n) + \frac{1}{2\alpha_n} \|\psi_{n+1} - \psi_n\|_{\mathcal{C}(X),1}^2 \leq 0$, we can obtain that

$$\frac{1}{2\alpha_n} \|\psi_{n+1} - \psi_n\|_{\mathcal{C}(X)}^2 \leq -d\mathcal{K}(\cdot, \mu_n, \varphi_n)_{\psi_n}(\psi_{n+1} - \psi_n)$$
$$\leq B \|\psi_{n+1} - \psi_n\|_{\mathcal{C}(X),1},$$

which is equivalent to

$$\|\psi_{n+1} - \psi_n\|_{\mathcal{C}(X),1} \leq 2B\alpha_n. \tag{46}$$

By the similar argument, we obtain

$$\|\mu_{n+1} - \mu_n\|_{\mathcal{M}(X)} \leq 2B\alpha_n, \quad \|\varphi_{n+1} - \varphi_n\|_{\mathcal{C}(X),2} \leq 2B\alpha_n. \tag{47}$$

By combining (42) with (43) - (47), we conclude (14):

$$\left|\mathcal{K}(\widehat{\psi}_N, \widehat{\mu}_N, \widehat{\varphi}_N) - \mathcal{K}(\psi_*, \mu_*, \varphi_*)\right| \leq \left(\sum_{n=0}^{N-1} \alpha_n\right)^{-1} \left(\frac{1}{2} C_s + 6B^2 \sum_{n=0}^{N-1} \alpha_n^2\right).$$

$$\square$$

# B   Proof of Theorem 5.14

Before the proof of the main result, we will show two lemmas used in the proof of Theorem 5.14.

**Lemma B.1.** *Let Assumptions 5.8, 5.9, 5.10, and 5.11 hold. Then, we have the following:*

(i) $\|\Phi(\psi_1, f) - \Phi(\psi_2, f)\|_{S_2} \leq \frac{L}{\beta}\|\psi_1 - \psi_2\|_{S_1}$ *for* $\psi_1, \psi_2 \in S_{1,c}$, $f \in S_c''$.

(ii) $\|\Phi(\psi, f_1) - \Phi(\psi, f_2)\|_{S_2} \leq \frac{L}{\beta}\|f_1 - f_2\|_{S''}$ *for* $\psi \in S_{1,c}$, $f_1, f_2 \in S_c''$.

(iii) $\psi \mapsto G(\psi, f)$ *is* $L\left(\frac{L}{\beta} + 1\right)$*-smooth with respect to* $\|\cdot\|_{S_1}$ *over* $S_{1,c}$ *for each* $f \in S_c''$.

(iv) $f \mapsto G(\psi, f)$ *is* $L\left(\frac{L}{\beta} + 1\right)$*-smooth with respect to* $\|\cdot\|_{S''}$ *over* $S_c''$ *for each* $\psi \in S_{1,c}$.

*Proof.* The proof is a generalization of Lin et al. (2020a, Lemma 4.3) to infinite dimensional function spaces with two variable.

By the optimality, we have

$$d\mathcal{G}(\psi_1, f, \cdot)_{\Phi(\psi_1, f)}(\Phi(\psi_2, f) - \Phi(\psi_1, f)) \leq 0,$$
$$d\mathcal{G}(\psi_2, f, \cdot)_{\Phi(\psi_2, f)}(\Phi(\psi_1, f) - \Phi(\psi_2, f)) \leq 0,$$

which implies that

$$[d\mathcal{G}(\psi_1, f, \cdot)_{\Phi(\psi_1, f)} - d\mathcal{G}(\psi_2, f, \cdot)_{\Phi(\psi_2, f)}](\Phi(\psi_2, f) - \Phi(\psi_1, f)) \leq 0. \tag{48}$$

With Assumption 5.9 and (48), we estimate that

$$\begin{aligned}
&\beta\|\Phi(\psi_1, f) - \Phi(\psi_2, f)\|_{S_2} \\
&\leq [d\mathcal{G}(\psi_1, f, \cdot)_{\Phi(\psi_1, f)} - d\mathcal{G}(\psi_1, f, \cdot)_{\Phi(\psi_2, f)}](\Phi(\psi_2, f) - \Phi(\psi_1, f)) \\
&\leq [d\mathcal{G}(\psi_2, f, \cdot)_{\Phi(\psi_2, f)} - d\mathcal{G}(\psi_1, f, \cdot)_{\Phi(\psi_2, f)}](\Phi(\psi_2, f) - \Phi(\psi_1, f)) \\
&\leq \|\nabla\mathcal{G}(\psi_2, f, \cdot)_{\Phi(\psi_2, f)} - \nabla\mathcal{G}(\psi_1, f, \cdot)_{\Phi(\psi_2, f)}\|_{S_2}\|\Phi(\psi_1, f) - \Phi(\psi_2, f)\|_{S_2} \\
&\leq L\|\psi_1 - \psi_2\|_{S_2}\|\Phi(\psi_1, f) - \Phi(\psi_2, f)\|_{S_2},
\end{aligned}$$

where last inequality results from Assumption 5.11. Hence, we obtain (i). (ii) is given by the same arguments of (i).

By the envelop theorem (Milgrom and Segal, 2002), the Gâteaux differential $dG(\cdot, f)_\psi$ of $G(\cdot, f)$ at $\psi \in S_1$ is represented as $dG(\cdot, f)_\psi = d\mathcal{G}(\cdot, f, \Phi(\psi, f))_\psi$. Using this, we estimate that

$$\begin{aligned}
&\|\nabla G(\cdot, f)_{\psi_1} - \nabla G(\cdot, f)_{\psi_2}\| \\
&= \|\nabla\mathcal{G}(\cdot, f, \Phi(\psi_1, f))_{\psi_1} - \nabla\mathcal{G}(\cdot, f, \Phi(\psi_2, f))_{\psi_2}\| \\
&\leq \|\nabla\mathcal{G}(\cdot, f, \Phi(\psi_1, f))_{\psi_1} - \nabla\mathcal{G}(\cdot, f, \Phi(\psi_2, f))_{\psi_1}\| \\
&\quad + \|\nabla\mathcal{G}(\cdot, f, \Phi(\psi_2, f))_{\psi_1} - \nabla\mathcal{G}(\cdot, f, \Phi(\psi_2, f))_{\psi_2}\| \\
&\leq L\left(\|\Phi(\psi_1, f) - \Phi(\psi_2, f)\|_{S_2} + \|\psi_1 - \psi_2\|_{S_1}\right) \\
&\leq L\left(\frac{L}{\beta} + 1\right)\|\psi_1 - \psi_2\|_{S_1},
\end{aligned}$$

where last inequality follows from (i). Using the above estimate, we further estimate that

$$
\begin{aligned}
& G(\psi_1, f) - G(\psi_2, f) - dG(\cdot, f)_{\psi_2}(\psi_1 - \psi_2) \\
& \leq \int_0^1 \frac{d}{d\epsilon} G(\psi_2 + \epsilon(\psi_1 - \psi_2), f) - dG(\cdot, f)_{\psi_2}(\psi_1 - \psi_2) d\epsilon \\
& \leq \int_0^1 dG(\cdot, f)_{\psi_2 + \epsilon(\psi_1 - \psi_2)}(\psi_1 - \psi_2) - dG(\cdot, f)_{\psi_2}(\psi_1 - \psi_2) d\epsilon \\
& \leq \int_0^1 L\left(\frac{L}{\beta} + 1\right)\epsilon \|\psi_1 - \psi_2\|_{S_1}^2 d\epsilon \\
& \leq \frac{1}{2} L\left(\frac{L}{\beta} + 1\right)\|\psi_1 - \psi_2\|_{S_1}^2.
\end{aligned}
\tag{49}
$$

Hence, we obtain (iii). (iv) is given by the same arguments. $\qquad\square$

**Lemma B.2.** *Let Assumptions 5.8, 5.9, 5.10, and 5.11 hold. Let $\eta > 0$ and $\varphi \in S_{2,c}$, and we denote by*

$$
\varphi_+ := \mathcal{P}_{S_{2,c}}\left(\varphi + \eta \nabla \mathcal{G}(\psi, f, \cdot)_\varphi\right)
$$

*Then, it holds that for $\psi \in S_{1,c}$, $f \in S_c''$, and $\phi \in S_{2,c}$,*

$$
-\mathcal{G}(\psi, f, \varphi_+) + \mathcal{G}(\psi, f, \phi) \leq \frac{1}{\eta}\langle \varphi_+ - \varphi, \phi - \varphi\rangle_{S_2} + \left(\frac{L}{2} - \frac{1}{\eta}\right)\|\varphi_+ - \varphi\|_{S_2}^2 - \frac{\beta}{2}\|\varphi - \phi\|_{S_2}^2.
$$

*Proof.* The proof is generalized from the finite dimensional case (Bubeck et al., 2015, Lemma 3.6).

By a property of the projection $\mathcal{P}_{S_{2,c}}$ (see Nesterov (2003, Lemma 3.1.4)), we have

$$
\langle \varphi_+ - (\varphi + \eta \nabla \mathcal{G}(\psi, f, \cdot)_\varphi), \varphi_+ - \phi\rangle_{S_2} \leq 0.
\tag{50}
$$

By Assumption 5.11 and same argument in (49), we can show that

$$
-\mathcal{G}(\psi, f, \varphi_+) \leq -\mathcal{G}(\psi, f, \varphi) + d\mathcal{G}(\psi, f, \cdot)_\varphi(\varphi_+ - \varphi) + \frac{L}{2}\|\varphi_+ - \varphi\|_{S_2}^2,
$$

which implies that with Assumption 5.9 and (50)

$$
\begin{aligned}
& -\mathcal{G}(\psi, f, \varphi_+) + \mathcal{G}(\psi, f, \phi) \\
& \leq -\mathcal{G}(\psi, f, \varphi_+) + \mathcal{G}(\psi, f, \varphi) - \mathcal{G}(\psi, f, \varphi) + \mathcal{G}(\psi, f, \phi) \\
& \leq -d\mathcal{G}(\psi, f, \cdot)_\varphi(\varphi_+ - \varphi) + \frac{L}{2}\|\varphi_+ - \varphi\|_{S_2}^2 + d\mathcal{G}(\psi, f, \cdot)_\varphi(\varphi - \phi) - \frac{\beta}{2}\|\varphi - \phi\|_{S_2}^2, \\
& \leq -\langle \nabla \mathcal{G}(\psi, f, \cdot)_\varphi, \varphi_+ - \phi\rangle_{S_2} + \frac{L}{2}\|\varphi_+ - \varphi\|_{S_2}^2 - \frac{\beta}{2}\|\varphi - \phi\|_{S_2}^2, \\
& \leq -\frac{1}{\eta}\langle \varphi_+ - \varphi, \varphi_+ - \phi\rangle_{S_2} + \frac{L}{2}\|\varphi_+ - \varphi\|_{S_2}^2 - \frac{\beta}{2}\|\varphi - \phi\|_{S_2}^2, \\
& \leq \frac{1}{\eta}\langle \varphi_+ - \varphi, \phi - \varphi\rangle_{S_2} + \left(\frac{L}{2} - \frac{1}{\eta}\right)\|\varphi_+ - \varphi\|_{S_2}^2 - \frac{\beta}{2}\|\varphi - \phi\|_{S_2}^2.
\end{aligned}
$$

$\qquad\square$

*Proof of Theorem 5.14.* The proof is a generalization of Lin et al. (2020a, Theorem 4.4) to infinite dimensional function spaces with three variables.

We denote by $L_\beta = L\left(\frac{L}{\beta} + 1\right)$. First, we estimate the upper bound of $G(\psi_{n+1}, f_{n+1})$. By Lemma B.1, we have

$$
G(\psi_{n+1}, f_{n+1}) \leq G(\psi_{n+1}, f_n) + dG(\psi_{n+1}, \cdot)_{f_n}(f_{n+1} - f_n) + \frac{L_\beta}{2}\|f_{n+1} - f_n\|_{S''}^2.
\tag{51}
$$

By a property of the projection $\mathcal{P}_{S''}$ (see Nesterov (2003, Lemma 3.1.5) and Young's inequality, we have

$$
\begin{aligned}
&\|f_{n+1} - f_n\|_{S''}^2 \\
&\leq \alpha_{f,n}^2 \|\nabla \mathcal{G}(\psi_n, \cdot, \varphi_n)_{f_n}\|_{S''}^2 \\
&\leq 2\alpha_{f,n}^2 \|\nabla \mathcal{G}(\psi_n, \cdot, \Phi(\psi_n, f_n))_{f_n} - \nabla \mathcal{G}(\psi_n, \cdot, \varphi_n)_{f_n}\|_{S''}^2 + 2\alpha_{f,n}^2 \|\nabla \mathcal{G}(\psi_n, \cdot, \Phi(\psi_n, f_n))_{f_n}\|_{S''}^2 \\
&\leq 2L^2 \alpha_{f,n}^2 \|\Phi(\psi_n, f_n) - \varphi_n\|_{S_2}^2 + 2\alpha_{f,n}^2 \|\nabla G(\psi_n, \cdot)_{f_n}\|_{S''}^2.
\end{aligned}
\tag{52}
$$

We estimate that

$$
\begin{aligned}
&dG(\psi_{n+1}, \cdot)_{f_n}(f_{n+1} - f_n) \\
&= \langle \nabla \mathcal{G}(\psi_{n+1}, \cdot, \Phi(\psi_{n+1}, f_n))_{f_n}, f_{n+1} - f_n \rangle_{S_2} \\
&= -\langle \nabla \mathcal{G}(\psi_{n+1}, \cdot, \Phi(\psi_{n+1}, f_n))_{f_n}, \alpha_{f,n} \nabla \mathcal{G}(\psi_n, \cdot, \Phi(\psi_n, f_n))_{f_n} \rangle_{S_2} \\
&\quad + \langle \nabla \mathcal{G}(\psi_{n+1}, \cdot, \Phi(\psi_{n+1}, f_n))_{f_n}, f_{n+1} - f_n + \alpha_{f,n} \nabla \mathcal{G}(\psi_n, \cdot, \Phi(\psi_n, f_n))_{f_n} \rangle_{S_2},
\end{aligned}
\tag{53}
$$

and by Lemma B.1 and Assumption 5.11,

$$
\begin{aligned}
&\|\nabla \mathcal{G}(\psi_{n+1}, \cdot, \Phi(\psi_{n+1}, f_n))_{f_n}\|_{S''} \\
&\leq \|\nabla \mathcal{G}(\psi_{n+1}, \cdot, \Phi(\psi_{n+1}, f_n))_{f_n} - \nabla \mathcal{G}(\psi_n, \cdot, \Phi(\psi_{n+1}, f_n))_{f_n}\|_{S''} \\
&\quad + \|\nabla \mathcal{G}(\psi_n, \cdot, \Phi(\psi_{n+1}, f_n))_{f_n} - \nabla \mathcal{G}(\psi_n, \cdot, \Phi(\psi_n, f_n))_{f_n}\|_{S''} \\
&\quad + \|\nabla \mathcal{G}(\psi_n, \cdot, \Phi(\psi_n, f_n))_{f_n}\|_{S''}, \\
&\leq L(1 + L_\beta)\|\psi_{n+1} - \psi_n\|_{S''} + \|\nabla G(\psi_n, \cdot)_{f_n}\|_{S''},
\end{aligned}
\tag{54}
$$

and by a property of the projection $\mathcal{P}_{S_{2,c}}$ (see Nesterov (2003, Lemma 3.1.5))

$$
\begin{aligned}
&\|f_{n+1} - f_n + \alpha_{f,n} \nabla \mathcal{G}(\psi_n, \cdot, \Phi(\psi_n, f_n))_{f_n}\|_{S''} \\
&\leq \alpha_{f,n} \|\nabla \mathcal{G}(\psi_n, \cdot, \varphi_n)_{f_n} - \nabla \mathcal{G}(\psi_n, \cdot, \Phi(\psi_n, f_n))_{f_n}\|_{S''} \\
&\leq L\alpha_{f,n} \|\varphi_n - \Phi(\psi_n, f_n)\|_{S_2}.
\end{aligned}
\tag{55}
$$

Combining (53) with (54) and (55), we futhre estimate that

$$
\begin{aligned}
&dG(\psi_{n+1}, \cdot)_{f_n}(f_{n+1} - f_n) \\
&\leq L(1 + L_\beta)\alpha_{f,n} \|\psi_{n+1} - \psi_n\|_{S_1} \|\nabla G(\psi_n, \cdot)_{f_n}\|_{S''} \\
&\quad - \alpha_{f,n} \|\nabla G(\psi_n, \cdot)_{f_n}\|_{S''}^2 \\
&\quad + L^2(1 + L_\beta)\alpha_{f,n} \|\psi_{n+1} - \psi_n\|_{S_1} \|\varphi_n - \Phi(\psi_n, f_n)\|_{S_2}, \\
&\quad + L\alpha_{f,n} \|\nabla G(\psi_n, \cdot)_{f_n}\|_{S''} \|\varphi_n - \Phi(\psi_n, f_n)\|_{S_2} \\
&\leq \frac{L(1 + L_\beta)}{2} \|\psi_{n+1} - \psi_n\|_{S_1}^2 + \frac{L(1 + L_\beta)\alpha_{f,n}^2}{2} \|\nabla G(\psi_n, \cdot)_{f_n}\|_{S''}^2 \\
&\quad - \alpha_{f,n} \|\nabla G(\psi_n, \cdot)_{f_n}\|_{S''}^2 \\
&\quad + \frac{L^2(1 + L_\beta)}{2} \|\psi_{n+1} - \psi_n\|_{S_1}^2 + \frac{L^2(1 + L_\beta)\alpha_{f,n}^2}{2} \|\varphi_n - \Phi(\psi_n, f_n)\|_{S_2}^2, \\
&\quad + \frac{L\alpha_{f,n}^2}{2} \|\nabla G(\psi_n, \cdot)_{f_n}\|_{S''}^2 + \frac{L}{2} \|\varphi_n - \Phi(\psi_n, f_n)\|_{S_2}^2,
\end{aligned}
\tag{56}
$$

where we have employed Young's inequality for last inequality. By the same way with (52), we have

$$
\begin{aligned}
&\|\psi_{n+1} - \psi_n\|_{S_1}^2 \\
&\leq \alpha_{\psi,n}^2 \|\nabla \mathcal{G}(\cdot, f_n, \varphi_n)_{\psi_n}\|_{S_1}^2 \\
&\leq 2\alpha_{\psi,n}^2 \|\nabla \mathcal{G}(\cdot, f_n, \Phi(\psi_n, f_n))_{\psi_n} - \nabla \mathcal{G}(\cdot, f_n, \varphi_n)_{\psi_n}\|_{S_1}^2 + 2\alpha_{\psi,n}^2 \|\nabla \mathcal{G}(\cdot, f_n, \Phi(\psi_n, f_n))_{\psi_n}\|_{S_1}^2 \\
&\leq 2L^2 \alpha_{\psi,n}^2 \|\Phi(\psi_n, f_n) - \varphi_n\|_{S_2}^2 + 2\alpha_{\psi,n}^2 \|\nabla G(\cdot, f_n)_{\psi_n}\|_{S_1}^2.
\end{aligned}
\tag{57}
$$

With (51), (52), (56), and (57), we obtain that

$$
\begin{aligned}
G(\psi_{n+1}, & f_{n+1}) \\
\leq\; & G(\psi_{n+1}, f_n) \\
& + \alpha_{f,n} \left\{ -1 + \frac{L(1+L_\beta)\alpha_{f,n}}{2} + \frac{L\alpha_{f,n}}{2} + L_\beta \alpha_{f,n} \right\} \|\nabla G(\psi_n, \cdot)_{f_n}\|_{S''}^2 \\
& + \frac{L(1+L)(1+L_\beta)}{2} \|\psi_{n+1} - \psi_n\|_{S_1}^2 \\
& + \left\{ \frac{L}{2} + \frac{L^2(1+L_\beta)\alpha_{f,n}^2}{2} + L^2 L_\beta \alpha_{f,n}^2 \right\} \|\Phi(\psi_n, f_n) - \varphi_n\|_{S_2}^2 \\
\leq\; & G(\psi_{n+1}, f_n) \\
& + \alpha_{f,n} \left\{ -1 + \frac{L(1+L_\beta)\alpha_{f,n}}{2} + \frac{L\alpha_{f,n}}{2} + L_\beta \alpha_{f,n} \right\} \|\nabla G(\psi_n, \cdot)_{f_n}\|_{S''}^2 \\
& + L(1+L)(1+L_\beta)\alpha_{\psi,n}^2 \|\nabla G(\cdot, f_n)_{\psi_n}\|_{S_1}^2 \\
& + \left\{ \frac{L}{2} + \frac{L^2(1+L_\beta)\alpha_{f,n}^2}{2} + L^2 L_\beta \alpha_{f,n}^2 + L^3(1+L)(1+L_\beta)\alpha_{\psi,n}^2 \right\} \|\Phi(\psi_n, f_n) - \varphi_n\|_{S_2}^2.
\end{aligned}
\tag{58}
$$

Second, we estimate the upper bound of $G(\psi_{n+1}, f_n)$. By Lemma B.1, we have

$$
G(\psi_{n+1}, f_n) \leq G(\psi_n, f_n) + dG(\cdot, f_n)_{\psi_n}(\psi_{n+1} - \psi_n) + \frac{L_\beta}{2} \|\psi_{n+1} - \psi_n\|_{S_1}^2.
\tag{59}
$$

By the same way with (58), we estimate that

$$
\begin{aligned}
dG(\cdot, & f_n)_{\psi_n}(\psi_{n+1} - \psi_n) \\
=\; & \langle \nabla \mathcal{G}(\cdot, f_n, \Phi(\psi_n, f_n))_{\psi_n}, \psi_{n+1} - \psi_n \rangle_{S_1} \\
=\; & -\alpha_{\psi,n} \|\nabla \mathcal{G}(\cdot, f_n, \Phi(\psi_n, f_n))_{\psi_n}\|_{S_1}^2 \\
& + \langle \nabla \mathcal{G}(\cdot, f_n, \Phi(\psi_n, f_n))_{\psi_n}, \psi_{n+1} - \psi_n + \alpha_{\psi,n}\nabla \mathcal{G}(\cdot, f_n, \Phi(\psi_n, f_n))_{\psi_n} \rangle_{S_2} \\
\leq\; & -\alpha_{\psi,n} \|\nabla G(\cdot, f_n)_{\psi_n}\|_{S_1}^2 \\
& + L\|\nabla G(\cdot, f_n)_{\psi_n}\|_{S_1} \|\varphi_n - \Phi(\psi_n, f_n)\|_{S_2} \\
\leq\; & \left\{ -1 + \frac{L\alpha_{\psi,n}}{2} \right\} \alpha_{\psi,n} \|\nabla G(\cdot, f_n)_{\psi_n}\|_{S_1}^2 + \frac{L}{2} \|\varphi_n - \Phi(\psi_n, f_n)\|_{S_2}.
\end{aligned}
\tag{60}
$$

Thus, by combining (57), (58), (59), and (60), we get

$$
\begin{aligned}
G(\psi_{n+1}, f_{n+1}) \leq\; & G(\psi_n, f_n) \\
& + \left\{ -1 + \frac{L\alpha_{\psi,n}}{2} + L(1+L)(1+L_\beta)\alpha_{\psi,n} + L_\beta \alpha_{\psi,n} \right\} \alpha_{\psi,n} \|\nabla G(\cdot, f_n)_{\psi_n}\|_{S_1}^2 \\
& + \left\{ -1 + \frac{L\alpha_{f,n}}{2} + \frac{L(1+L_\beta)\alpha_{f,n}}{2} + L_\beta \alpha_{f,n} \right\} \alpha_{f,n} \|\nabla G(\psi_n, \cdot)_{f_n}\|_{S''}^2 \\
& + \underbrace{\left\{ L + L^2 L_\beta \alpha_{\psi,n}^2 + L^3(1+L)(1+L_\beta)\alpha_{\psi,n}^2 + L^2 L_\beta \alpha_{f,n}^2 + \frac{L^2(1+L_\beta)\alpha_{f,n}^2}{2} \right\}}_{\text{Assumption 5.13 (ii)} \leq C} \\
& \times \|\Phi(\psi_n, f_n) - \varphi_n\|_{S_2}^2.
\end{aligned}
\tag{61}
$$

Third, we estimate $\|\Phi(\psi_n, f_n) - \varphi_n\|_{S_2}^2 =: \delta_n$. Using Lemma B.2 and Assumption 5.13 (i), we evaluate that

$$
\begin{aligned}
&\|\varphi_n - \Phi(f_{n-1}, \psi_{n-1})\|_{S_2}^2 \\
&\leq \|\varphi_{n-1} - \Phi(f_{n-1}, \psi_{n-1})\|_{S_2}^2 + 2\langle\varphi_{n-1} - \Phi(f_{n-1}, \psi_{n-1}), \varphi_n - \varphi_{n-1}\rangle_{S_2} + \|\varphi_n - \varphi_{n-1}\|_{S_2}^2 \\
&\leq (1 - \beta\alpha_{\varphi,n-1})\|\varphi_n - \Phi(f_{n-1}, \psi_{n-1})\|_{S_2}^2 + (-1 + L\alpha_{\varphi,n-1})\|\varphi_n - \varphi_{n-1}\|_{S_2}^2 \\
&\leq (1 - \beta\alpha_{\varphi,n-1})\delta_{n-1},
\end{aligned}
$$

which implies that by using Young's inequality, Lemma B.1, (52), and (57), we have

$$
\begin{aligned}
\delta_n &= \|\Phi(\psi_n, f_n) - \varphi_n\|_{S_2}^2 \\
&\leq (1 + \beta\alpha_{\varphi,n-1})\|\Phi(\psi_{n-1}, f_{n-1}) - \varphi_n\|_{S_2}^2 \\
&\quad + 2\left(1 + \frac{1}{\beta\alpha_{\varphi,n-1}}\right)\left(\|\Phi(\psi_{n-1}, f_n) - \Phi(\psi_n, f_n)\|_{S_2}^2 + \|\Phi(\psi_{n-1}, f_n) - \Phi(\psi_{n-1}, f_{n-1})\|_{S_2}^2\right) \\
&\leq (1 - \beta^2\alpha_{\varphi,n-1}^2)\delta_{n-1} + \frac{2L^2}{\beta^2}\left(1 + \frac{1}{\beta C_0}\right)\left(\|\psi_{n-1} - \psi_n\|_{S_2}^2 + \|f_{n-1} - f_n\|_{S''}^2\right) \\
&\leq \left\{(1 - \beta^2\alpha_{\varphi,n-1}^2) + \frac{2L^2}{\beta^2}\left(1 + \frac{1}{\beta C_0}\right)(\alpha_{\psi,n-1}^2 + \alpha_{f,n-1}^2)\right\}\delta_{n-1} \\
&\quad + \frac{2L^2}{\beta^2}\left(1 + \frac{1}{\beta C_0}\right)\left(\alpha_{\psi,n-1}^2\|\nabla G(\cdot, f_{n-1})_{\psi_{n-1}}\|_{S_1}^2 + \alpha_{f,n-1}^2\|\nabla G(\psi_{n-1}, \cdot)_{f_{n-1}}\|_{S''}^2\right) \\
&\leq \gamma\delta_{n-1} + \frac{2L^2}{\beta^2}\left(1 + \frac{1}{\beta C_0}\right)\left(\alpha_{\psi,n-1}^2\|\nabla G(\cdot, f_{n-1})_{\psi_{n-1}}\|_{S_1}^2 + \alpha_{f,n-1}^2\|\nabla G(\psi_{n-1}, \cdot)_{f_{n-1}}\|_{S''}^2\right),
\end{aligned}
\tag{62}
$$

where we have employed Assumption 5.13 (iii) for last inequality. Then, we have

$$
\delta_n \leq \gamma^n\delta_0 + \frac{2L^2}{\beta^2}\left(1 + \frac{1}{\beta C_0}\right)\sum_{i=0}^{n}\left(\alpha_{\psi,i}^2\gamma^{n-i}\|\nabla G(\cdot, f_i)_{\psi_i}\|_{S_1}^2 + \alpha_{f,i}^2\gamma^{n-i}\|\nabla G(\psi_i, \cdot)_{f_i}\|_{S''}^2\right).
$$

By this and (61), we have

$$
\begin{aligned}
&\left\{1 - \frac{L\alpha_{\psi,n}}{2} - L(1+L)(1+L_\beta)\alpha_{\psi,n} - L_\beta\alpha_{\psi,n}\right\}\alpha_{\psi,n}\|\nabla G(\cdot, f_n)_{\psi_n}\|_{S_1}^2 \\
&+ \left\{1 - \frac{L\alpha_{f,n}}{2} - \frac{L(1+L_\beta)\alpha_{f,n}}{2} - L_\beta\alpha_{f,n}\right\}\alpha_{f,n}\|\nabla G(\psi_n, \cdot)_{f_n}\|_{S''}^2 \\
&\leq G(\psi_n, f_n) - G(\psi_{n+1}, f_{n+1}) + C\gamma^n\delta_0 \\
&\quad + \frac{2L^2C}{\beta^2}\left(1 + \frac{1}{\beta C_0}\right)\sum_{i=0}^{n}\left(\alpha_{\psi,i}^2\gamma^{n-i}\|\nabla G(\cdot, f_i)_{\psi_i}\|_{S_1}^2 + \alpha_{f,i}^2\gamma^{n-i}\|\nabla G(\psi_i, \cdot)_{f_i}\|_{S''}^2\right),
\end{aligned}
$$

and taking the summation over the interval $n = 0$ to $N - 1$,

$$\sum_{n=0}^{N-1} \left\{ 1 - \frac{L\alpha_{\psi,n}}{2} - L(1+L)(1+L_\beta)\alpha_{\psi,n} - L_\beta\alpha_{\psi,n} \right\} \alpha_{\psi,n} \|\nabla G(\cdot, f_n)_{\psi_n}\|_{S_1}^2$$

$$+ \sum_{n=0}^{N-1} \left\{ 1 - \frac{L\alpha_{f,n}}{2} - \frac{L(1+L_\beta)\alpha_{f,n}}{2} - L_\beta\alpha_{f,n} \right\} \alpha_{f,n} \|\nabla G(\psi_n, \cdot)_{f_n}\|_{S''}^2$$

$$\leq G(\psi_0, f_0) - G(\psi_N, f_N) + C\delta_0 \sum_{n=0}^{N-1} \gamma^n$$

$$+ \frac{2L^2 C}{\beta^2} \left( 1 + \frac{1}{\beta C_0} \right) \sum_{n=0}^{N-1} \sum_{i=0}^{n} \left( \alpha_{\psi,i}^2 \gamma^{n-i} \|\nabla G(\cdot, f_i)_{\psi_i}\|_{S_1}^2 + \alpha_{f,i}^2 \gamma^{n-i} \|\nabla G(\psi_i, \cdot)_{f_i}\|_{S''}^2 \right),$$

$$\leq G(\psi_0, f_0) - \inf_{\psi, f} G(\psi, f) + C\delta_0 \sum_{n=0}^{\infty} \gamma^n$$

$$+ \frac{2L^2 C}{\beta^2} \left( 1 + \frac{1}{\beta C_0} \right) \left( \sum_{i=0}^{\infty} \gamma^i \right) \sum_{n=0}^{N-1} \left( \alpha_{\psi,n}^2 \|\nabla G(\cdot, f_n)_{\psi_n}\|_{S_1}^2 + \alpha_{f,n}^2 \|\nabla G(\psi_n, \cdot)_{f_n}\|_{S''}^2 \right),$$

which is equivalent to

$$\sum_{n=0}^{N-1} \underbrace{\left\{ 1 - \frac{L\alpha_{\psi,n}}{2} - L(1+L)(1+L_\beta)\alpha_{\psi,n} - L_\beta\alpha_{\psi,n} - \frac{2L^2 C\alpha_{\psi,n}}{\beta^2(1-\gamma)} \left( 1 + \frac{1}{\beta C_0} \right) \right\}}_{\text{Assumption 5.13 (iv) } \geq C_\psi > 0}$$

$$\times \alpha_{\psi,n} \|\nabla G(\cdot, f_n)_{\psi_n}\|_{S_1}^2$$

$$+ \sum_{n=0}^{N-1} \underbrace{\left\{ 1 - \frac{L\alpha_{f,n}}{2} - \frac{L(1+L_\beta)\alpha_{f,n}}{2} - L_\beta\alpha_{f,n} - \frac{2L^2 C\alpha_{f,n}}{\beta^2(1-\gamma)} \left( 1 + \frac{1}{\beta C_0} \right) \right\}}_{\text{Assumption 5.13 (v) } \geq C_f > 0}$$

$$\times \alpha_{f,n} \|\nabla G(\psi_n, \cdot)_{f_n}\|_{S''}^2$$

$$\leq G(\psi_0, f_0) - \inf_{\psi, f} G(\psi, f) + \frac{C\delta_0}{1-\gamma}.$$

Finally, we estimate that

$$\left\| \widehat{\nabla G}_{\psi,N} \right\|_{S_1} = \left\| \frac{\sum_{n=0}^{N-1} \alpha_{\psi,n} \nabla G(f_n, \cdot)_{\psi_n}}{\sum_{n=0}^{N-1} \alpha_{\psi,n}} \right\|_{S_1}$$

$$\leq \frac{\sum_{n=0}^{N-1} \alpha_{\psi,n} \|\nabla G(f_n, \cdot)_{\psi_n}\|_{S_1}}{\sum_{n=0}^{N-1} \alpha_{\psi,n}} \leq \frac{\left( \sum_{n=0}^{N-1} \alpha_{\psi,n} \|\nabla G(f_n, \cdot)_{\psi_n}\|_{S_1}^2 \right)^{1/2}}{\left( \sum_{n=0}^{N-1} \alpha_{\psi,n} \right)^{1/2}}.$$

By the same way, we estimate that

$$\left\| \widehat{\nabla G}_{f,N} \right\|_{S''} \leq \frac{\left( \sum_{n=0}^{N-1} \alpha_{f,n} \|\nabla G(\psi_n, \cdot)_{f_n}\|_{S''}^2 \right)^{1/2}}{\left( \sum_{n=0}^{N-1} \alpha_{f,n} \right)^{1/2}}.$$

Therefore, we conclude Theorem 5.14.

$\square$

# C  Proofs in Section 6.1

## C.1  Proof of Proposition 6.1

*Proof.* For $\alpha \in [0,1]$, $\psi_1, \psi_2 \in \mathcal{C}(X)$, and $\mu_1, \mu_2 \in \mathcal{M}(X)$,

$$
\begin{aligned}
&R(\alpha\psi_1 + (1-\alpha)\psi_2, \alpha\mu_1 + (1-\alpha)\mu_2) \\
&\leq \alpha^2 \int \ell(\psi_1, \psi_0) d\mu_1 + (1-\alpha)^2 \int \ell(\psi_2, \psi_0) d\mu_2 \\
&\quad + \alpha(1-\alpha)\left( \int \ell(\psi_1, \psi_0) d\mu_2 + \int \ell(\psi_2, \psi_0) d\mu_1 \right) \\
&\quad + \alpha V(\psi_1) + (1-\alpha) V(\psi_2) - \frac{\alpha(1-\alpha)\gamma}{2} \|\psi_1 - \psi_2\|^2_{\mathcal{C}(X),2} \\
&\quad + \alpha W(\mu_1) + (1-\alpha) W(\mu_2) - \frac{\alpha(1-\alpha)\gamma}{2} \|\mu_1 - \mu_2\|^{*2}_{\mathcal{C}(X),1} \\
&= \alpha \underbrace{\left( \int \ell(\psi_1, \psi_0) d\mu_1 + V(\psi_1) + W(\mu_1) \right)}_{=R(\psi_1,\mu_1)} + (1-\alpha) \underbrace{\left( \int \ell(\psi_2, \psi_0) d\mu_2 + V(\mu_2) + W(\mu_2) \right)}_{=R(\psi_2,\mu_2)} \\
&\quad + \alpha(1-\alpha) \underbrace{\left( -\int (\ell(\psi_1,\psi_0) - \ell(\psi_2,\psi_0)) d(\mu_1 - \mu_2) - \frac{\gamma}{2} \|\psi_1 - \psi_2\|^2_{\mathcal{C}(X),2} - \frac{\gamma}{2} \|\mu_1 - \mu_2\|^{*2}_{\mathcal{C}(X),1} \right)}_{=(*)},
\end{aligned}
$$

and $(*)$ is non-positive because we have

$$
\begin{aligned}
(*) &\leq \underbrace{\|\ell(\psi_1,\psi_0) - \ell(\psi_2,\psi_0)\|_{\mathcal{C}(X),1}}_{\leq \rho\|\psi_1-\psi_2\|_{\mathcal{C}(X),2} \leq \gamma\|\psi_1-\psi_2\|_{\mathcal{C}(X),2}} \|\mu_1 - \mu_2\|^{\star}_{\mathcal{C}(X),1} \\
&\quad - \frac{\gamma}{2} \|\psi_1 - \psi_2\|^2_{\mathcal{C}(X),2} - \frac{\gamma}{2} \|\mu_1 - \mu_2\|^{*2}_{\mathcal{C}(X),1} \\
&\leq -\frac{\gamma}{2} \left( \|\psi_1 - \psi_2\|_{\mathcal{C}(X),2} - \|\mu_1 - \mu_2\|^{\star}_{\mathcal{C}(X),1} \right)^2 \leq 0.
\end{aligned}
$$

$\square$

## C.2  Proof of Lemma 6.2

*Proof.* For $\psi, \varphi \in S_{\mathcal{C},a,b}$,

$$
\begin{aligned}
D_{I_{h,\mu}}(\psi|\varphi) &= I_{h,\mu}(\psi) - I_{h,\mu}(\varphi) - d(I_{h,\mu})_\varphi(\psi - \varphi) \\
&= \int \left( h(\psi) - h(\varphi) - h'(\varphi)(\psi - \varphi) \right) d\mu \\
&\leq \frac{L}{2} \int |\psi - \varphi|^2 \, d\mu = \|\psi - \varphi\|^2_{L^2(X,\mu)}.
\end{aligned}
$$

$\square$

## C.3  Proof of Lemma 6.3

*Proof.* By the definition of $f$-divergence (24), we have

$$
J_f^\star(\varphi) = \sup_{\mu \in \mathcal{M}(X)} \int \varphi d\mu - J_f(\mu) = \sup_{\mu \ll \nu_0} \int \varphi d\mu - \int f\left( \frac{d\mu}{d\nu_0} \right) d\nu_0.
$$

We solve a concave maximization problem for $\mu \mapsto \int \varphi d\mu - \int f\left( \frac{d\mu}{d\nu_0} \right) d\nu_0$. We consider

$$
\frac{d}{d\epsilon} \left( \int \varphi d(\mu + \epsilon\chi) - \int f\left( \frac{d\mu + \epsilon\chi}{d\nu_0} \right) d\nu_0 \right) \bigg|_{\epsilon=0} = 0,
$$

which is equivalent to

$$\int \varphi d\chi - \int f'\left(\frac{d\mu}{d\nu_0}\right)\frac{d\chi}{d\nu_0}d\nu_0 = \int \left(\varphi - f'\left(\frac{d\mu}{d\nu_0}\right)\right)d\chi = 0,$$

for all $\chi$. Then, the optimal $\mu$ satisfies

$$\varphi = f'\left(\frac{d\mu}{d\nu}\right).$$

By the assumption, $f'$ is invertible, and $\varphi \in S_{\mathcal{C},f}$. Substituting $\frac{d\mu}{d\nu_0} = (f')^{-1}(\varphi)$ into

$$J_f^\star(\varphi) = \sup_\mu \int \varphi \frac{d\mu}{d\nu_0}d\nu_0 - \int f\left(\frac{d\mu}{d\nu_0}\right)d\nu_0,$$

then, we obtain that

$$J_f^\star(\varphi) = \int \left\{\varphi \cdot (f')^{-1}(\varphi) - f \circ (f')^{-1}(\varphi)\right\}d\nu_0.$$

$\square$

## C.4 Proof of Lemma 6.6

*Proof.* By the definition of IPM (28), we have

$$J_{\mathrm{IPM},\nu_0}(\mu) = \sup_{\varphi \in \mathcal{C}(X)} \int \varphi d\mu - \int \varphi d\nu_0 - \chi\{\varphi \in \mathcal{F}\}.$$

By the Fenchel-Moreau theorem, we obtain that

$$J_{\mathrm{IPM},\nu_0}^\star(\varphi) = \int \varphi d\nu_0 + \chi\{\varphi \in \mathcal{F}\}.$$

$\square$

## C.5 Proof of Proposition 6.9

*Proof.* (2) holds due to the convexity of $k(\cdot)$. (4) follows from the linearity of $\mu \mapsto \mathcal{K}_1(\psi, \mu, \varphi)$ and the norm $\frac{\gamma}{2}\|\cdot\|_{\mathcal{H}_\sigma}^\star$ induced by inner products.

For (3), it holds that

$$D_{\mathcal{K}_1(\cdot,\mu,\varphi)}(\psi_1|\psi_2) = \frac{1}{2}\int (\psi_1 - \psi_2)^2 d\mu + \frac{\gamma}{2}\|\psi_1 - \psi_2\|_{\mathcal{H}_{2\sqrt{2}\sigma}}^2,$$

and by $\mu \in S'$, we have

$$\frac{1}{2}\int (\psi_1 - \psi_2)^2 d\mu \le \frac{1}{2}\int (\psi_1 - \psi_2)^2 d\mu_u = \frac{1}{2}\|\psi_1 - \psi_2\|_{L^2(X,\mu_u)}^2.$$

For (5), we estimate by using the $L_k$-smoothness of $k : (a,b) \to \mathbb{R}$ and $\nu_0 \in S'$

$$\begin{aligned}
D_{-\mathcal{K}_1(\psi,\mu,\cdot)}(\varphi_1|\varphi_2) &= D_{\int k(\cdot)d\nu_0}(\varphi_1|\varphi_2) \\
&= \int \left\{k(\varphi_1) - k(\varphi_2) - k'(\varphi_2)(\varphi_1 - \varphi_2)\right\}d\nu_0 \\
&\le \frac{L_k}{2}\int |\varphi_1 - \varphi_2|^2 d\mu_u \\
&= \frac{L_k}{2}\|\varphi_1 - \varphi_2\|_{L^2(X,\mu_u)}^2.
\end{aligned}$$

Finally, we will prove (1). To apply Proposition 6.1 as $\frac{1}{2}\ell(\cdot, \psi_0) = (\cdot - \psi_0)^2$, $V(\psi) = \frac{\gamma}{2}\|\psi\|^2_{\mathcal{H}_{2\sqrt{2}\sigma}}$, $W(\mu) = \frac{\gamma}{2}\|\mu\|^{\star 2}_{\mathcal{H}_\sigma}$, we verify assumptions (i), (ii), and (iii) in Proposition 6.1. (i) holds due to the convexity of $t \mapsto (t-s)^2$. (iii) holds because norms $\|\cdot\|_{\mathcal{H}_{2\sqrt{2}\sigma}}$ and $\|\cdot\|^\star_{\mathcal{H}_\sigma}$ are induced by inner products. We prove (ii) as followings:

By Chu et al. (2020, Proposition 14), the RKSH norm $\|f\|_{\mathcal{H}_\sigma}$ is represented as

$$\|f\|^2_{\mathcal{H}_\sigma} = \sum_{k=0}^\infty (\frac{1}{2}\sigma^2)^k \sum_{|\alpha|=k} \frac{1}{\alpha!} \|\partial_x^\alpha f\|^2_{L^2(\mathbb{R}^d)}, \tag{63}$$

for $f \in \mathcal{H}_\sigma$. Here, we employ the multi-index notation with $d$-dimensional multi-index $\alpha = (\alpha_1, ..., \alpha_d) \in \mathbb{N}_0^d$ where the sum of its components denotes the $|\alpha| = \alpha_1 + \cdots + \alpha_d$. Additionally, we define the factorial of the multi-index as $\alpha! = \alpha_1! \cdots \alpha_d!$, and the partial derivative as $\partial_x^\alpha = \partial_{x_1}^{\alpha_1} \cdots \partial_{x_d}^{\alpha_d}$.

We estimate that

$$\begin{aligned}
&\|\partial_x^\alpha ((\psi_1 + \psi_2 - 2\psi_0)(\psi_1 - \psi_2))\|^2_{L^2(\mathbb{R}^d)} \\
&= \left\|\sum_{\beta \leq \alpha} \binom{\alpha}{\beta} \partial_x^{\alpha-\beta}(\psi_1 + \psi_2 - 2\psi_0)\partial_x^\beta(\psi_1 - \psi_2)\right\|^2_{L^2(\mathbb{R}^d)} \\
&\leq \left(\sum_{\beta \leq \alpha} \binom{\alpha}{\beta} \underbrace{\left\|\partial_x^{\alpha-\beta}(\psi_1 + \psi_2 - 2\psi_0)\right\|_{L^\infty(\mathbb{R}^d)}}_{\leq 4C_b} \left\|\partial_x^\beta(\psi_1 - \psi_2)\right\|_{L^2(\mathbb{R}^d)}\right)^2 \\
&\leq (4C_b)^2 \underbrace{\left(\sum_{\beta \leq \alpha} \binom{\alpha}{\beta}\right)^2}_{=(2^k)^2} \times \underbrace{\left(\sum_{\beta \leq \alpha} \left\|\partial_x^\beta(\psi_1 - \psi_2)\right\|_{L^2(\mathbb{R}^d)}\right)^2}_{\leq 2^k \sum_{\beta \leq \alpha}\left\|\partial_x^\beta(\psi_1-\psi_2)\right\|^2_{L^2(\mathbb{R}^d)}} \\
&\leq 16C_b^2 8^k \sum_{\beta \leq \alpha} \left\|\partial_x^\beta(\psi_1 - \psi_2)\right\|^2_{L^2(\mathbb{R}^d)},
\end{aligned} \tag{64}$$

where the first equality employs the Leibniz formula, the second inequality utilizes the Cauchy–Schwarz inequality and the result of $\psi_1, \psi_2, \psi_0 \in S_1$, and the third inequality makes use of the Cauchy–Schwarz inequality and multi-binomial theorem.

By using (63) and (64), we further estimate that

$$\left\| \frac{1}{2}(\psi_1 - \psi_0)^2 - \frac{1}{2}(\psi_2 - \psi_0)^2 \right\|^2_{\mathcal{H}_\sigma}$$

$$= \frac{1}{4} \left\| (\psi_1 + \psi_2 - 2\psi_0)(\psi_1 - \psi_2) \right\|^2_{\mathcal{H}_\sigma}$$

$$= 4C_b^2 \sum_{k=0}^\infty (4\sigma^2)^k \sum_{|\alpha|=k} \frac{1}{\alpha!} \sum_{\beta \leq \alpha} \left\| \partial_x^\beta (\psi_1 - \psi_2) \right\|^2_{L^2(\mathbb{R}^d)}$$

$$= 4C_b^2 \sum_{k=0}^\infty \sum_{|\alpha|=k} \left[ \sum_{\beta \geq \alpha} (4\sigma^2)^{|\beta|} \frac{1}{\beta!} \right] \left\| \partial_x^\alpha (\psi_1 - \psi_2) \right\|^2_{L^2(\mathbb{R}^d)}$$

$$\leq 4C_b^2 \sum_{k=0}^\infty (4\sigma^2)^k \sum_{|\alpha|=k} \frac{1}{\alpha!} \left[ \sum_{\beta \geq \alpha} (4\sigma^2)^{|\beta|-k} \right] \left\| \partial_x^\alpha (\psi_1 - \psi_2) \right\|^2_{L^2(\mathbb{R}^d)}$$

$$\leq 4C_b^2 \sum_{k=0}^\infty (\frac{1}{2}(2\sqrt{2}\sigma)^2)^k \sum_{|\alpha|=k} \frac{1}{\alpha!} \underbrace{\left[ \sum_{\beta_d \geq \alpha_d} \cdots \sum_{\beta_1 \geq \alpha_1} (4\sigma^2)^{\beta_d - \alpha_d} \cdots (4\sigma^2)^{\beta_1 - \alpha_1} \right]}_{=\left[\sum_{j \geq k}(4\sigma^2)^{j-k}\right]^d = C_\sigma^d} \left\| \partial_x^\alpha (\psi_1 - \psi_2) \right\|^2_{L^2(\mathbb{R}^d)}$$

$$\leq 4C_b^2 C_\sigma^d \sum_{k=0}^\infty (\frac{1}{2}(2\sqrt{2}\sigma)^2)^k \sum_{|\alpha|=k} \frac{1}{\alpha!} \left\| \partial_x^\alpha (\psi_1 - \psi_2) \right\|^2_{L^2(\mathbb{R}^d)}$$

$$= 4C_b^2 C_\sigma^d \left\| (\psi_1 - \psi_0) \right\|^2_{\mathcal{H}_{2\sqrt{2}\sigma}},$$

where $C_\sigma = \sum_{j \in \mathbb{N}_0} (4\sigma^2)^j < \infty$, which implies that $\psi \mapsto \frac{1}{2}(\psi - \psi_0)^2$ is $4C_b^2 C_\sigma^d$-Lipschitz with respect to $\|\cdot\|_{\mathcal{H}_\sigma}$ and $\|\cdot\|_{\mathcal{H}_{2\sqrt{2}\sigma}}$. Thus, by the assumption of $\gamma \geq 4C_b^2 C_\sigma^d$ and applying Proposition 6.1 to our setting, we conclude that $(\psi, \mu) \mapsto \mathcal{K}_1(\psi, \mu, \varphi)$ is convex. $\square$

## D Proofs in Section 6.2

### D.1 Proof of Lemma 6.10

*Proof.* Since $\mathcal{I} : \mathcal{M}(X) \to \mathbb{R}$ is $(1/\beta)$-smooth with respect to $\|\cdot\|_{\mathcal{M}(X)}$, the convex conjugate $\mathcal{I}^\star : \mathcal{C}(X) \to \mathbb{R}$ is $\beta$-strongly convex with respect to $\|\cdot\|^\star_{\mathcal{M}(X)}$. By this and

$$(J_{\nu_0} \oplus \mathcal{I})^\star = J_{\nu_0}^\star + \mathcal{I}^\star,$$

then $(J_{\nu_0} \oplus \mathcal{I})^\star$ is $\beta$-strongly convex with respect to $\|\cdot\|^\star_{\mathcal{M}(X)}$ due to the fact that strong convexity is preserved by adding convex functions. $\square$

### D.2 Proof of Lemma 6.11

*Proof.* As $h$ is Lipschitz continuous, $h$ is absolutely continuous. Thus, the derivative $\nabla h$ of $h$ is defined a.e. in $X$ with respect to the Lebesgue measure $m$. By the assumption $\mu << m$, the derivative $\nabla h$ is also defined a.e. in $X$ with respect to probability measure $\xi$, which implies that

$$\frac{d}{d\epsilon} \mathcal{J}_{h,\xi}(f + \epsilon g) \Big|_{\epsilon=0} = \int_Z \nabla h(f(z)) \cdot g(z) \mu(dz).$$

$\square$

### D.3 Proof of Proposition 6.13

*Proof.* (1) is given by Lemma 6.10. (2) holds from Lemma 6.11, and Gâteaux differentials are given by

$$dG_1(\cdot, f, \varphi)_\psi(\eta) = 2\int (\psi \circ f - \psi_0 \circ f)\eta \circ f d\xi_0,$$

$$dG_1(\psi, \cdot, \varphi)_f(g) = 2\int (\psi \circ f - \psi_0 \circ f)\{(\nabla\psi - \nabla\psi_0) \circ f\} \cdot g d\xi_0 + \int \{\nabla\varphi \circ f\} \cdot g d\xi_0,$$

$$dG_1(\psi, f, \cdot)_\varphi(\phi) = \int \phi \circ f d\xi_0 - \int k'(\varphi) \cdot \phi d\nu_0 - \beta \langle \phi, \varphi \rangle_{\mathcal{H}_\sigma}.$$

We will confirm Assumption 5.11 as follows:

(a): for $\psi_1, \psi_2 \in S_{1,c}$, $f \in S_{1,c}$, $\varphi \in S_{2,c}$, and $\eta \in S_1$ with $\|\eta\|_{H^1(X)} \leq 1$,

$$dG_1(\cdot, f, \varphi)_{\psi_1}(\eta) - dG_1(\cdot, f, \varphi)_{\psi_2}(\eta) = 2\int (\psi_1 \circ f - \psi_2 \circ f)\eta \circ f d\xi_0$$

$$= 2\int (\psi_1 - \psi_2)\eta d(f_\sharp \xi_0)$$

$$\leq 2C_3\|\psi_1 - \psi_2\|_{L^2(X)}\|\eta\|_{L^2(X)}$$

$$\leq 2C_3\|\psi_1 - \psi_2\|_{H^1(X)},$$

$$\Rightarrow \|dG_1(\cdot, f, \varphi)_{\psi_1} - dG_1(\cdot, f, \varphi)_\psi\|^\star_{H^1(X)} \leq 2C_3\|\psi_1 - \psi_2\|_{H^1(X)}.$$

(b): for $\psi \in S_{1,c}$, $f_1, f_2 \in S''_c$, $\varphi \in S_{2,c}$, and $\eta \in S_1$ with $\|\eta\|_{H^1(X)} \leq 1$,

$$dG_1(\cdot, f_1, \varphi)_\psi(\eta) - dG_1(\cdot, f_2, \varphi)_\psi(\eta)$$

$$= 2\int (\psi \circ f_1 - \psi_0 \circ f_1)\eta \circ f_1 d\xi_0 - 2\int (\psi \circ f_2 - \psi_0 \circ f_2)\eta \circ f_2 d\xi_0$$

$$\leq 2\int |\psi \circ f_1 - \psi_0 \circ f_1||\eta \circ f_1 - \eta \circ f_2| d\xi_0$$

$$+ 2\int |\psi \circ f_1 - \psi \circ f_2||\eta \circ f_2| d\xi_0 + 2\int |\psi_0 \circ f_1 - \psi_0 \circ f_2||\eta \circ f_2| d\xi_0$$

$$\leq 4C_2 \mathrm{Lip}(\eta) \int |f_1 - f_2| d\xi_0 + 4C_1 \int |f_1 - f_2||\eta \circ f_2| d\xi_0$$

$$\leq 4C_2\|\nabla\eta\|_{L^\infty(X)}|Z|^{1/2}\|f_1 - f_2\|_{L^2(Z;X,\xi_0)} + 4C_1 C_3^{1/2}\|\eta\|_{L^2(X)}\|f_1 - f_2\|_{L^2(Z;X,\xi_0)}$$

$$\leq \left(4C_2\widetilde{C}_{d,X}|Z|^{1/2} + 4C_1 C_3^{1/2}\right)\|f_1 - f_2\|_{L^2(Z;X,\xi_0)},$$

$$\Rightarrow \|dG_1(\cdot, f_1, \varphi)_\psi - dG_1(\cdot, f_2, \varphi)_\psi\|^\star_{H^1(X)} \leq \left(4C_2\widetilde{C}_{d,X}|Z|^{1/2} + 4C_1 C_3^{1/2}\right)\|f_1 - f_2\|_{L^2(Z;X,\xi_0)},$$

where the last inequality results from Lemma D.1 where $\widetilde{C}_{d,X} > 0$ is some constant depending on $d$ and $X$.

(c): for $\psi \in S_{1,c}$, $f \in S_{1,c}$, $\varphi_1, \varphi_2 \in S_{2,c}$, and $\eta \in S_1$ with $\|\eta\|_{H^1(X)} \leq 1$,

$$dG_1(\cdot, f, \varphi_1)_\psi(\eta) - dG_1(\cdot, f, \varphi_2)_\psi(\eta) = 0$$

$$\Rightarrow \|dG_1(\cdot, f, \varphi_1)_\psi - dG_1(\cdot, f, \varphi_2)_\psi\|^\star_{H^1(X)} \leq c\|\varphi_1 - \varphi_2\|_{\mathcal{H}_\sigma},$$

for any $c > 0$.

(d): for $\psi \in S_{1,c}$, $f_1, f_2 \in S''_c$, $\varphi \in S_{2,c}$, and $g \in S''$ with $\|g\|_{L^2(Z;X,\xi_0)} \leq 1$,

$$
d\mathcal{G}_1(\psi, \cdot, \varphi)_{f_1}(g) - d\mathcal{G}_1(\psi, \cdot, \varphi)_{f_2}(g)
$$

$$
= 2 \int (\psi \circ f_1 - \psi_0 \circ f_1)\{(\nabla\psi - \nabla\psi_0) \circ f_1\} \cdot g d\xi_0 + \int \{\nabla\varphi \circ f_1\} \cdot g d\xi_0
$$

$$
- 2 \int (\psi \circ f_2 - \psi_0 \circ f_2)\{(\nabla\psi - \nabla\psi_0) \circ f_2\} \cdot g d\xi_0 - \int \{\nabla\varphi \circ f_2\} \cdot g d\xi_0
$$

$$
\leq 4C_2 \int |(\nabla\psi - \nabla\psi_0) \circ f_1 - (\nabla\psi - \nabla\psi_0) \circ f_2| |g| d\xi_0
$$

$$
+ 4C_2 \int \{|\psi \circ f_1 - \psi \circ f_2| + |\psi_0 \circ f_1 - \psi_0 \circ f_2|\} |g| d\xi_0 + \int |\nabla\varphi \circ f_1 - \nabla\varphi \circ f_2| |g| d\xi_0,
$$

$$
\leq (16C_1 C_2 + C_4) \int |f_1 - f_2| |g| d\xi_0
$$

$$
\leq (16C_1 C_2 + C_4)\|f_1 - f_2\|_{L^2(Z;X,\xi_0)},
$$

$$
\Rightarrow \|d\mathcal{G}_1(\psi, \cdot, \varphi)_{f_1} - d\mathcal{G}_1(\psi, \cdot, \varphi)_{f_2}\|^\star_{L^2(Z;X,\xi_0)} \leq (16C_1 C_2 + C_4)\|f_1 - f_2\|_{L^2(Z;X,\xi_0)}.
$$

(e): for $\psi_1, \psi_2 \in S_{1,c}$, $f \in S''_c$, $\varphi \in S_{2,c}$, and $g \in S''$ with $\|g\|_{L^2(Z;X,\xi_0)} \leq 1$,

$$
d\mathcal{G}_1(\psi_1, \cdot, \varphi)_f(g) - d\mathcal{G}_1(\psi_2, \cdot, \varphi)_f(g)
$$

$$
= 2 \int (\psi_1 \circ f - \psi_0 \circ f)\{(\nabla\psi_1 - \nabla\psi_0) \circ f\} \cdot g d\xi_0
$$

$$
- 2 \int (\psi_2 \circ f - \psi_0 \circ f)\{(\nabla\psi_2 - \nabla\psi_0) \circ f\} \cdot g d\xi_0
$$

$$
\leq 2 \int |\psi_1 \circ f - \psi_0 \circ f| |\nabla\psi_1 \circ f - \nabla\psi_2 \circ f| |g| d\xi_0
$$

$$
+ 2 \int |\psi_1 \circ f - \psi_2 \circ f| |\nabla\psi_2 \circ f - \nabla\psi_0 \circ f| |g| d\xi_0
$$

$$
\leq 4C_2 \int |\nabla\psi_1 \circ f - \nabla\psi_2 \circ f| |g| d\xi_0 + 4C_2 \int |\psi_1 \circ f - \psi_2 \circ f| |g| d\xi_0
$$

$$
\leq 4C_2 C_3^{1/2}\|\psi_1 - \psi_2\|_{H^1(X)},
$$

$$
\Rightarrow \|d\mathcal{G}_1(\psi_1, \cdot, \varphi)_f(g) - d\mathcal{G}_1(\psi_2, \cdot, \varphi)_f(g)\|^\star_{L^2(Z;X,\xi_0)} \leq 4C_2 C_3^{1/2}\|\psi_1 - \psi_2\|_{H^1(X)}.
$$

(f): for $\psi \in S_{1,c}$, $f \in S''_c$, $\varphi_1, \varphi_2 \in S_{2,c}$, and $g \in S''$ with $\|g\|_{L^2(Z;X,\xi_0)} \leq 1$,

$$
d\mathcal{G}_1(\psi, \cdot, \varphi_1)_f(g) - d\mathcal{G}_1(\psi, \cdot, \varphi_2)_f(g)
$$

$$
= \int (\nabla\varphi_1 \circ f) \cdot g - (\nabla\varphi_2 \circ f) \cdot g d\xi_0
$$

$$
\leq C_3^{1/2}\|\nabla\varphi_1 - \nabla\varphi_2\|_{L^2(X)}\|g\|_{L^2(Z;X,\xi_0)}
$$

$$
\leq C_3^{1/2}\widetilde{C}_\sigma\|\varphi_1 - \varphi_2\|_{\mathcal{H}_\sigma}
$$

$$
\Rightarrow \|d\mathcal{G}_1(\psi, \cdot, \varphi_1)_f - d\mathcal{G}_1(\psi, \cdot, \varphi_2)_f\|^\star_{L^2(Z;X,\xi_0)} \leq C_3^{1/2}\widetilde{C}_\sigma\|\varphi_1 - \varphi_2\|_{\mathcal{H}_\sigma},
$$

where the last inequality follows from (63) where $\widetilde{C}_\sigma > 0$ is some constant depending on $\sigma$.

(g): for $\psi \in S_{1,c}$, $f \in S_{1,c}$, $\varphi_1, \varphi_2 \in S_{2,c}$, and $\phi \in S_2$ with $\|\phi\|_{\mathcal{H}_\sigma} \leq 1$,

$$d\mathcal{G}_1(\psi, f, \cdot)_{\varphi_1}(\phi) - d\mathcal{G}_1(\psi, f, \cdot)_{\varphi_2}(\phi)$$

$$= \int \{k'(\varphi_2) - k'(\varphi_1)\} \cdot \phi d\nu_0 + \beta \langle \phi, \varphi_2 - \varphi_1 \rangle_{\mathcal{H}_\sigma}$$

$$\leq L_k \sup_{x \in X} \left| \frac{d\nu_0}{dm}(x) \right|^{1/2} \|\varphi_1 - \varphi_2\|_{L^2(X)} + \beta \|\varphi_1 - \varphi_2\|_{\mathcal{H}_\sigma}$$

$$\leq \left( L_k \sup_{x \in X} \left| \frac{d\nu_0}{dm}(x) \right|^{1/2} + \beta \right) \|\varphi_1 - \varphi_2\|_{\mathcal{H}_\sigma},$$

$$\Rightarrow \|d\mathcal{G}_1(\psi, f, \cdot)_{\varphi_1} - d\mathcal{G}_1(\psi, f, \cdot)_{\varphi_2}\|_{\mathcal{H}_\sigma}^\star \leq \left( L_k \sup_{x \in X} \left| \frac{d\nu_0}{dm} \right|^{1/2} + \beta \right) \|\varphi_1 - \varphi_2\|_{\mathcal{H}_\sigma}.$$

(h): for $\psi_1, \psi_2 \in S_{1,c}$, $f \in S_{1,c}$, $\varphi \in S_{2,c}$, and $\phi \in S_2$ with $\|\phi\|_{\mathcal{H}_\sigma} \leq 1$,

$$d\mathcal{G}_1(\psi_1, f, \cdot)_\varphi(\phi) - d\mathcal{G}_1(\psi_2, f, \cdot)_\varphi(\phi) = 0,$$

$$\Rightarrow \|d\mathcal{G}_1(\psi_1, f, \cdot)_\varphi - d\mathcal{G}_1(\psi_2, f, \cdot)_\varphi\|_{H^1(X)}^\star \leq c\|\psi_1 - \psi_2\|_{H^1(X)},$$

for any $c > 0$.

(i): for $\psi \in S_{1,c}$, $f_1, f_2 \in S_{1,c}$, $\varphi \in S_{2,c}$, and $\phi \in S_2$ with $\|\phi\|_{\mathcal{H}_\sigma} \leq 1$,

$$d\mathcal{G}_1(\psi, f_1, \cdot)_\varphi(\phi) - d\mathcal{G}_1(\psi, f_2, \cdot)_\varphi(\phi) = \int \phi \circ f_1 - \phi \circ f_2 d\xi_0$$

$$\leq \mathrm{Lip}(\phi) \int |f_1 - f_2| d\xi_0$$

$$\leq \widetilde{C}_{\sigma, d, X} \xi_0(Z)^{1/2} \|f_1 - f_2\|_{L^2(Z; X, \xi_0)}$$

$$\Rightarrow \|d\mathcal{G}_1(\psi, f_1, \cdot)_\varphi - d\mathcal{G}_1(\psi, f_2, \cdot)_\varphi\|_{\mathcal{H}_\sigma}^\star \leq \widetilde{C}_{\sigma, d, X} \xi_0(Z)^{1/2} \|f_1 - f_2\|_{L^2(Z; X, \xi_0)},$$

where the last inequality results from Lemma D.1 where $\widetilde{C}_{\sigma, d, X} > 0$ is some constant depending on $\sigma$, $d$, and $X$. $\qquad \square$

We have employed the following fundamental Lemma in the proof of Proposition 6.13.

**Lemma D.1.** *Let $\Omega \subset \mathbb{R}^d$ be a compact set, and let $f : \Omega \to \mathbb{R}$ be Lipschitz continuous. Then, we have*

$$\|f\|_{L^\infty(\Omega)} \leq \max \left\{ \frac{(d+1)|\Omega|^{1/2}}{2^d} \|f\|_{L^2(\Omega)}, \left( \frac{(d+1)d^{d/2}|\Omega|^{1/2}}{2^d} \right)^{\frac{1}{d+1}} \|f\|_{L^2(\Omega)}^{\frac{1}{d+1}} \right\}.$$

*Proof.* Assume that $f : \Omega \to \mathbb{R}$ be $L$-Lipschitz with $L \geq 1$. We denote by

$$a := \mathrm{argmax}_{x \in \Omega} f(x),$$

and

$$M := \|f\|_{L^\infty(\Omega)} = \max_{x \in \Omega} f(x).$$

Without loss of generality, we can assume that $M > 0$.

As $f : \Omega \to \mathbb{R}$ is $L$-Lipschitz, that is,

$$|f(x) - f(a)| \leq L|x - a|, \ x \in \Omega,$$

we estimate that

$$L \int_0^{M/L} 1_{\{|x-a| \le v\}} dv \le L\left(\frac{M}{L} - |x-a|\right)$$
$$\le -L|x-a| + M \le f(x),$$

which implies that

$$\int_\Omega |f(x)| dx \ge L \int_\Omega \int_0^{M/L} 1_{\{|x-a| \le v\}} dv dx$$
$$= L \int_0^{M/L} \int_\Omega 1_{\{|x-a| \le v\}} dx dv$$
$$\ge L \int_0^{M/L} \prod_{i=1}^d \int_{-R+a_i}^{R+a_i} 1_{\{|x_i - a_i| \le v/\sqrt{d}\}} dx dv$$
$$= L \int_0^{M/L} Q_{R,d}(v) dv,$$

where

$$Q_{R,d}(v) := \begin{cases} \left(\frac{2v}{\sqrt{d}}\right)^d & \frac{v}{\sqrt{d}} < R \\ (2R)^d & \frac{v}{\sqrt{d}} \ge R. \end{cases}$$

Here, $R > 1$ is chosen large enough such that

$$\Omega \subset \prod_{i=1}^d [-R+a_i, R+a_i].$$

By direct computation, we can show that

$$\int_\Omega |f(x)| dx$$
$$\ge \begin{cases} 2^d R^d \left(M - RL\sqrt{d}\frac{d}{d+1}\right) & \frac{M}{L} > \sqrt{d}R \\ 2^d R^d \frac{\sqrt{d}}{d+1} LR^{d+1} & \frac{M}{L} \le \sqrt{d}R \end{cases}$$
$$\ge \begin{cases} \frac{2^d}{d+1} \|f\|_{L^\infty(\Omega)} & \frac{M}{L} > \sqrt{d}R \\ \frac{2^d}{(d+1)d^{d/2}} \|f\|_{L^\infty(\Omega)}^{d+1} & \frac{M}{L} \le \sqrt{d}R. \end{cases}$$

Therefore, we conclude that

$$\|f\|_{L^\infty(\Omega)}$$
$$\le \max\left\{ \frac{d+1}{2^d} \|f\|_{L^1(\Omega)}, \left(\frac{(d+1)d^{d/2}}{2^d}\right)^{\frac{1}{d+1}} \|f\|_{L^1(\Omega)}^{\frac{1}{d+1}} \right\}$$
$$\le \max\left\{ \frac{(d+1)|\Omega|^{1/2}}{2^d} \|f\|_{L^2(\Omega)}, \left(\frac{(d+1)d^{d/2}|\Omega|^{1/2}}{2^d}\right)^{\frac{1}{d+1}} \|f\|_{L^2(\Omega)}^{\frac{1}{d+1}} \right\}.$$

$\square$

