# OpenReview forum: "Convergences for Minimax Optimization Problems over Infinite-Dimensional Spaces Towards Stability in Adversarial Training"
_TMLR — Accepted by TMLR_

### Review · Reviewer_qvrN · 2024-04-24

**Summary Of Contributions:**

This paper uses functional analysis tools to study the convergence behavior of minimax optimization problems, focusing specifically on GD applied in infinite-dimensional spaces. Under appropriate assumptions for their minimax problem, the authors demonstrate that GD converges to the minimax solution in convex-concave settings, while in nonconvex-concave scenarios, GD has a vanishing gradient norm and converge to a stationary point. Furthermore, they illustrate how these assumptions can be met in certain GANs and UDA algorithms, and they justify the effectiveness of techniques like spectral normalization and gradient penalty in stabilizing adversarial training.

**Audience:**

Yes

**Broader Impact Concerns:**

I do not see any ethical concerns with this paper, as it is purely theoretical in nature.

**Claims And Evidence:**

Yes

**Requested Changes:**

Below are some suggestions for adjustments, with the first two points being my primary concerns, while the remaining points can be used to strengthen the work.

1. *Improve the clarity of the motivation behind analyzing the infinite-dimensional space.* Given that the effectiveness of spectral normalization and gradient penalty has already been justified in finite-dimensional spaces by Chu et al. (2020), what advantages does extending the analysis to infinite-dimensional spaces provide? Is the motivation to better characterize the case of UDA by analyzing the minimax problem over infinite-dimensional space? It seems that with additional assumptions for the source risk term (e.g., convexity and smoothness), the framework of Chu et al. (2019, 2020) could also be applied to study UDA.

2. *Provide more details about the adaptations in the analysis compared to previous works.* If studying the minimax problem over infinite-dimensional space has advantages, could the authors highlight the main differences between their analysis and previous works, such as Lin et al. (2020) and Chu et al. (2019, 2020) in finite-dimensional settings? Although the related work section and other parts of this work briefly discuss similarities and differences, readers may not grasp the key modifications without digging into all of these works.

3. *Verify whether the $\mathcal{H}\Delta\mathcal{H}$-divergence satisfies the assumptions.* In Section 6.13, concrete examples of $f$-divergence and IPMs stratify the smoothness conditions. Can you also elaborate on the $\mathcal{H}\Delta\mathcal{H}$-divergence in [R1]? Note that $\mathcal{H}$-divergence and $\mathcal{H}\Delta\mathcal{H}$-divergence are studied in seminal works [R1,R2] of UDA theory, thus taking $\mathcal{H}\Delta\mathcal{H}$-divergence into discussion is meaningful.

[R1] Shai Ben-David, et al. "A theory of learning from different domains." Machine learning 79 (2010): 151-175.

[R2] Shai Ben-David, et al. "Analysis of representations for domain adaptation." Advances in neural information processing systems 19 (2006).

4. *Comment on Assumption 5.13.* Assumption 5.13 is hard to understand and may not be clear enough for why it can be satisfied immediately in Section 6.2.3. This assumption, which seems to suggest that step sizes should be sufficiently small for convergence, also implies that $L$ and $\beta$ should not be too large. For instance, if $L\gg \beta$, and $C_0$ is very small such that condition (i) is satisfied, then it seems possible that the LHS in (iii) can be very large such that $\gamma\in(0,1)$ does not exist. Can you clarify why Assumption 5.13 can be immediately satisfied in Section 6.2.3?

5. *Discuss the empirical distribution case and SGD.* In practical adversarial training, since the true distribution $\mu$ and $\nu_0$ are unknown, we usually use the training sample as a "proxy". If $\mu$ is replaced by its empirical version $\hat{\mu}$, are any modifications required in the analysis? Additionally, SGD is often used in practical algorithms. Is it possible to formulate SGD rather than GD in your framework? If such a formulation is feasible, considering that SGD introduces some randomness or noise, will it affect convergence to a stationary point in the nonconvex-concave setting?

6. *Minor point: Add a table summarizing the notations used in this paper for the convenience of readers.*

**Strengths And Weaknesses:**

**Strengths**
1. This paper extends previous analyses of minimax optimization from finite-dimensional to infinite-dimensional spaces, focusing on both convex-concave and nonconvex-concave settings.

2. In addition, the analysis framework presented here applies to both GANs and UDA. While GANs have received attention in previous literature, this work supplements the understanding of convergence properties in adversarial training for UDA, an area yet to be extensively explored.


**Weaknesses**
1. From a technical standpoint, it is unclear how much new contribution is made by extending previous theoretical analyses in Chu et al.(2019, 2020) and Lin et al.(2020) from finite-dimensional to infinite-dimensional cases. Specifically, it seems Chu et al. (2020) are already aware that certain smoothness conditions can be extended to the infinite-dimensional case, as indicated in the first paragraph in their Page 9 (following Proposition 13), where the last sentence states that "this regularization results in a function with Lipschitz gradients, so it is unsurprising that this property carries over to the infinite-dimensional case."

2. In terms of conclusions, the effectiveness of spectral normalization and gradient penalty in adversarial training for GANs has been previously justified by Chu et al. (2020). Therefore, this work may not provide additional insights into adversarial training, limiting its significance.

3. Two additional minor weaknesses include the limitation of the current analysis to convex-Lipschitz continuous loss functions in UDA (e.g., squared loss), and the difficulty in empirically verifying the theoretical results, a challenge acknowledged by the authors.

---

> ### Author Response · Authors · 2024-05-07
>
> We appreciate your valuable comments. We address your feedback as follows:
>
> > 1. Improve the clarity of the motivation ... 2. Provide more details about the adaptations  ...
>
> Our motivation is to analyze the minimax optimization problem rather than minimization as the actual setting in adversarial training like GANs and UDAs employs the convex duality of objective function to be minimized.
> However, the finite dimensional minimax problem in the adversarial training becomes nonconvex-nonconcave, making it challenging to show convergence.
> Thus, we propose to analyze the ``ideal setting" of the infinite dimensional space (the probability space and continuous function space), where this minimax optimization problem is either convex-concave or nonconvex-concave (see Section 4).
> This convex-concave and nonconvex-concave cases can be addressed by generalizing finite dimensional minimax convergences as discussed in prior works [Nedić and Ozdaglar 2009] and [Lin et al 2020], respectively.
> Please note that the our analysis differs from prior works not only in generalizing to infinite dimensional spaces but also in considering miniminmax problems interacting the effect of predictor, classifier, and generator.
>
> Firstly, we provide insights similar to [Chu et al 2020], which interpret stability techniques often used in GANs, such as spectral normalization and gradient penalty, via the sufficient conditions for minimax convergence, even though [Chu et al 2020] discussed the minimization problem.
> Secondly, we provide the insight for UDAs, emphasizing that joint convexity is a important factor for stabilization and smoothness, which could be achieved by  regularization for predictor and generator (Section 6.1.1).
> This is a new insight for UDAs, which [Chu et al 2020] never included, as they only considered minimization of divergence $\mathcal{J}$, not including source term $R$.
>
> We would like to revise our introduction and related work based on the above discussions.
>
> > 3. Verify whether the $H$-divergence  ... into discussion is meaningful.
>
> It would be interesting to verify whether the $H$-divergence satisfies the assumption.
> In our current formulation, we utilize the convex conjugates of the divergence, implying that the classifier $\varphi :X \to \mathbb{R}$ belongs to continuous function space $\mathcal{C}(X)$.
> However, the formulation of the $H$-divergence (e.g., [R1] and [R2]) employs the hypothesis spaces, where the corresponding classifier $\varphi : X \to \{0,1\}$ is a labeling function.
> Therefore, integrating the $H$-divergence would need a different formulation, which is beyond the scope of our paper.
> We would like to discuss it as our future work in Section 7.
>
>
> > 4. Comment on Assumption 5.13. Assumption 5.13 ...  immediately satisfied in Section 6.2.3?
>
> Section 6 primarily verifies the assumptions regarding the properties of the objective function (e.g., smoothness and convexity) through concrete examples of the divergence $\mathcal{J}$ and the source risk $R$.
> On the other hand, Assumption 5.13 corresponds to step sizes of GD, independently of the properties of the objective function.
> Roughly speaking, Assumption 5.13 requires that step sizes be chosen small enough depending on $\beta$ and $L$. The following details are what we would like to add to our paper:
> First, we need to choose small $\alpha_{\varphi, n}$ satisfying (i).
> However, $\alpha_{\varphi, n}$ should not converge to zero as $n \to \infty$. Let's denote its lower bound as $C_0>0$.
> Next, we need to choose small $\alpha_{\psi, n}$ and $\alpha_{f, n}$ satisfying (ii)~(v). The smallness of $\alpha_{\psi, n}$ and $\alpha_{f, n}$ depends on lower bound $C_0>0$ (see (iii)--(v)). In other words, $\alpha_{\psi, n}$ and $\alpha_{f, n}$ should be chosen to be smaller than $\alpha_{\varphi, n}$.
>
>
> > 5. Discuss the empirical distribution ...  point in the nonconvex-concave setting?
>
> Since our theory deals with any probability measure, including empirical probability, no modifications are required in the case where $\mu$ and $\nu_0$ are empirical probability measures.
>
> In our paper, we focus on discussing the stability and smoothness of GANs and UDAs, primarily examining the properties of the objective function such as convexity and smoothness.
> Thus, employing GD, which is the simplest algorithm, suffices to achieve our aim.
> However, formulating SGD within the framework of our work would be an interesting future direction.
> Notably, [Lin et al. 2020] studied the convergence in finite-dimensional minimax problems using both GD and SGD.
> Our work in Section 5.2 extends the results from GD presented in [Lin et al. 2020] to infinite dimensional spaces. Similarly, we believe that we could generalize the results for SGD as well.
> We would like to comment on these discussions in Section 7 as future work.
>
> > Minor point: ... for the convenience of readers.
>
> We will make a table to summarize all notations for the convenience of readers.

---

### Review · Reviewer_cgJA · 2024-04-26

**Summary Of Contributions:**

This paper focuses on the convergence of the minimax optimization problems over the infinite-dimensional spaces towards stability in adversarial training. Specifically, this work tackle the instability problem of adversarial optimization through a functional analysis, where the results show that the stabilization techniques (e.g., spectral normalization and gradient penalty) can serve as the necessary condition for a good convergence.

To be more concise, the paper's contributions are,
- present an analysis of the convergence to the minimax solution over the infinite-dimensional spaces;
- verify the fulfillment of sufficient conditions for the convergence properties in certain GANs and UDA settings, and provide the theoretical interpretation of existing techniques like spectral normalization and gradient penalty.

**Audience:**

Yes

**Claims And Evidence:**

Yes

**Requested Changes:**

Overall, the reviewer thinks this work presents a rigorous framework for analyzing the convergence for minimax optimization problems over infinite-dimensional spaces towards the stability in adversarial training. The following lists some specific comments,

1. Please summarize all the notations in a table to allow a easy reading.
2. Beyond the problem setting of the GAN and UDA, please also introduce and discuss further on the limitation of previous analysis for the finite-dimensional spaces of probability measures or continuous functions.
3. Could the authors further discuss the future problems of the existing theoretical frameworks except for the empirical verification, like the validity of the assumptions in more practical scenarios?

**Strengths And Weaknesses:**

Strengths:
1. This work presents a thorough proof and analysis of the convergence of the minimax optimization problem in the scheme of gradient descent. Specifically, under appropriate assumptions, the authors analyze the convex-concave problem of continuous functions and the probability measures and also consider the nonconvex-concave problem over spaces of continuous functions.
2. Most of the assumptions and definitions are clear and well-presented.
3. In the reviewer's view, there are no fatal issues in the analysis and the corresponding proof. The overall framework is rigorous for the convergence analysis of the minimax problem in the infinite-dimensional spaces of continuous functions and the probability measures.
4. The analysis framework draws theoretical insights into understanding the conditions for the convergence properties in certain GANs and UDAs settings.

Weaknesses:
1. The organization and the presentation of the paper can be further improved by better structuring the analysis framework.
2. It would be better if the authors could explicitly point out the distinguishable differences on the analysis framework compared with the previous related work (e.g.,  the unique assumption this work has made and the derivation conducted different than previous ones), or further discuss the major technical contribution for the optimization problem under the infinite-dimensional spaces.

---

> ### Author Response · Authors · 2024-05-07
>
> We appreciate your valuable comments. We address your feedback as follows:
>
> >1. Please summarize all the notations in a table to allow a easy reading.
>
> We will make a table to summarize all notations for the convenience of readers.
>
> > 2. Beyond the problem setting of the GAN and UDA, please also introduce ... probability measures or continuous functions.
>
> Our motivation is to analyze the minimax optimization problem rather than minimization as the actual setting in adversarial training like GANs and UDAs employs the convex duality of objective function to be minimized.
> However, the finite dimensional minimax problem in the adversarial training becomes nonconvex-nonconcave, making it challenging to show convergence.
> Thus, we propose to analyze the ``ideal setting" of the infinite dimensional space (the probability space and continuous function space), where this minimax optimization problem is either convex-concave or nonconvex-concave (see Section 4).
> This convex-concave and nonconvex-concave cases can be addressed by generalizing finite dimensional minimax convergences as discussed in prior works [Nedić and Ozdaglar 2009] and [Lin et al 2020], respectively.
> Please note that the our analysis differs from prior works not only in generalizing to infinite dimensional spaces but also in considering miniminmax problems interacting the effect of predictor, classifier, and generator.
>
> Firstly, we provide insights similar to [Chu et al 2020], which interpret stability techniques often used in GANs, such as spectral normalization and gradient penalty, via the sufficient conditions for minimax convergence, even though [Chu et al 2020] discussed the minimization problem.
> Secondly, we provide the insight for UDAs, emphasizing that joint convexity is a important factor for stabilization and smoothness, which could be achieved by  regularization for predictor and generator (Section 6.1.1).
> This is a new insight for UDAs, which [Chu et al 2020] never included, as they only considered minimization of divergence $\mathcal{J}$, not including source term $R$.
>
> We would like to revise our introduction and related work based on the above discussions.
>
> > 3. Could the authors further discuss the future problems ... assumptions in more practical scenarios?
>
> While in our current work, we fix $\nu_0$ as the true distribution,
> in the context of UDAs, $\nu_0$ represents the target distribution which is indeed optimized.
> Extending our current work to include optimization of $\nu_0$ would be an interesting future direction.
> Additionally, verifying $H$-divergence, a different divergence from $f$-divergence and IPMs discussed in our paper, would be meaningful for UDAs.
> We will include these discussions in our future work in Section 7.

---

### Review · Reviewer_tDMF · 2024-05-04

**Summary Of Contributions:**

This paper formulates GAN and UDA as the minimax problem in convex-concave and nonconvex-concave settings. Specifically, the study considers infinite-dimensional functional spaces and mainly focuses on the convergence under small optimization errors.
1. Convex-concave:
$$\min_{\psi, \mu \in S_1\times S^\prime}\max_{\phi\in S_2}  \mathcal{K}(\psi, \mu, \phi),$$
where $S_1, S_2 \subset \mathcal{C}(X)$ are compact convex subset of some functional space and $S^\prime \subset \mathcal{P}(X)$ is a compact convex subset of all the measures over $X$.  This is the general form of eq(8).

2. Nonconvex-concave:
$$\min_{\psi, f \in S\_{1,c}\times S_{c}^{\prime\prime} } \max_{\phi\in S\_{2,c}}  \mathcal{G}(\psi, f, \phi),$$
where $S_{1,c}\subset S_1, S_{2,c} \subset S_2$ and $S^\prime \subset \mathcal{C}(Z, X)$ are compact convex subset of some functional spaces. This is the general form of eq(10).

They further proved that,  under proper assumptions, the convex-concave setting exists convergence to the true minimax solution, and the nonconvex-concave setting shows convergence to the stationary point.

**Audience:**

Yes

**Claims And Evidence:**

Yes

**Requested Changes:**

Please fix the problems mentioned in the above section.

**Strengths And Weaknesses:**

### Strengths

1. The paper is well-written and easy to follow.
2. The technical part is solid.
3. Different conditions have been verified for certain GAN and UDA settings.

### Weaknesses
1. Unclear problem formulation
    * The optimization problem for UDA was defined as $\min_{(\psi, \mu) \in \mathcal{C}(X)\times \mathcal{P}(X)} R(\psi, \mu) + J_{\nu_0}(\mu)$, where $\psi$ is the predictor, $\mu$  and $\nu_0$ are respectively the source and target distribution on $X$.
    * However, in UDA, $\mu$ and $\nu_0$ are both fixed distributions, which cannot be optimized. In particular, $\mu$ is the source marginal data distribution on $X$. The distribution matching in UDA was realized by either conducting matching on the marginal representation space [1,2,3,4] or on the joint probability space [5,6].
    * So, I am not convinced by this part. Later on, the authors have defined $\mu=f\_{\\#}\xi_0$ as a push-forward measure in eq(10).
    * However, in the context of UDA, it should be claimed that $\xi_0$ is the source distribution and $\mu$ is some induced measure defined on the representation space.
    * A *very close formulation in [6]* using Wasserstein distance lacks a discussion, which is a minimax problem formulated to maximize some discriminator and minimize w.r.t both the predictor and feature extractor.

2. Related works

The related works in UDA  have not been sufficiently discussed, as mentioned above. In addition, some work has studied when the equilibrium exists for GAN, e.g., [7], which lacks a discussion.

3. Novelty

The paper is an extension of [8] to the infinite functional space with the same order of $N$ for convergence. Hence, the novelty is not that impressive to me. In addition, an improved minimax algorithm has been studied in [9], so it's better to involve a discussion.

[1] Ben-David, Shai, John Blitzer, Koby Crammer, Alex Kulesza, Fernando Pereira, and Jennifer Wortman Vaughan. “A theory of learning from different domains.”  In: Machine learning 79.1 (2010).

[2] Ben-David, Shai, John Blitzer, Koby Crammer, and Fernando Pereira. “Analysis of representations for domain adaptation”. In: Advances in neural information processing systems 19(2006).

[3]Ganin, Yaroslav, Evgeniya Ustinova, Hana Ajakan, Pascal Germain, Hugo Larochelle, François Laviolette, Mario Marchand, and Victor Lempitsky. “Domain-adversarial training of neural networks”. In: The journal of machine learning research 17.1 (2016).

[4]Zhao, Han, Remi Tachet Des Combes, Kun Zhang, and Geoffrey Gordon. “On learning invariant representations for domain adaptation”. In: International Conference on Machine Learning. PMLR. 2019.

[5] Acuna, David, Guojun Zhang, Marc T Law, and Sanja Fidler. “f-domain adversarial learning: Theory and algorithms”. In: International Conference on Machine Learning. PMLR. 2021.

[6]Chen, Qi and Mario Marchand. “Algorithm-Dependent Bounds for Representation Learning of Multi-Source Domain Adaptation”. In: arXiv preprint arXiv:2304.02064 (2023)

[7] Arora, Sanjeev, et al. "Generalization and equilibrium in generative adversarial nets (gans)." International conference on machine learning. PMLR, 2017.

[8] Tianyi Lin, Chi Jin, and Michael Jordan. On gradient descent ascent for nonconvex-concave minimax problems. In International Conference on Machine Learning, pages 6083–6093. PMLR, 2020.

[9] Lin, Tianyi, Chi Jin, and Michael I. Jordan. "Near-optimal algorithms for minimax optimization." Conference on Learning Theory. PMLR, 2020.

---

> ### Author Response · Authors · 2024-05-07
>
> We appreciate your valuable comments. We address your feedback as follows:
>
> > The optimization problem for UDA was defined as ... is some induced measure defined on the representation space.
>
> Our aim is to interpret stability techniques used in adversarial training via the minimax optimization problem.
> In our "ideal setting", defined in equation (7), we address either convex-concave or nonconvex-concave problems over infinite dimensional spaces, enabling us to show their convergences.
> By verifying sufficient conditions of the convergence, we interpret stability techniques commonly employed in GANs, such as spectral normalization and gradient penalty.
> Additionally, we provide new insight into UDAs from the aspect of joint convexity (see Section 6.1.1).
> Therefore, the "ideal setting" suffices for our aim in interpreting stability techniques.
> As you pointed out, we optimize the distribution $\mu$ or the transport map $f$ as variables, which deviates from the actual setting of UDAs.
> In a more realistic scenario for UDAs, $\mu$ and $\nu_0$  would be formulated as $\mu=f_{\sharp}\xi_{s}$ and $\nu_0 =g_{\sharp} \xi_{t}$, where $f$ and $g$ are, for instance, neural networks optimizing their weight parameters, and $\xi_{s}$ and $\xi_t$ are fixed source and target distributions.
> However, this scenario is highly nonconvex-nonconcave,  and this setting is more challenging to show convergence.
> We would like to incorporate these discussions in Section 7.
>
> >The paper is an extension of [8] to ... Hence, the novelty is not that impressive to me.
>
> Our novelty in terms of analysis does not improve the order but rather generalizes [Nedić and Ozdaglar 2009] and [Lin et al 2020] to infinite-dimensional spaces of probability and continuous function spaces.
> Moreover, we generalize the analysis from minimax to miniminimax (minimizing with respect to the predictor and generator and maximizing with respect to the classifier).
>
>
> > A very close formulation in [6] using Wasserstein distance... [7], which lacks a discussion... has been studied in [9], so it's better to involve a discussion.
>
> We would like to add additional discussions and comparisons with relevant references, such as [6,7,9].

---

### Author Response · Authors · 2024-05-13
**upload the revied version**

Dear editor and reviewer,

Thank you very much for handling our paper.
We have uploaded the revied version, which incorporates the feedback provided by the reviewer.
The following is the list of changes :

- In Section 2(Related work), we have expanded our discussions to include relevant papers.

- Following Assumption 5.13, we have included additional details regarding the interpretation of the assumption.

- In Section 6.3 (Interpretations of our analysis), we have provided an interpretation of joint convexity as a new contribution not previously discussed in the work of [Chu et al. (2020)].

- In Section 7 (Conclusion and future work), we have added a list of the future work.

---

### Decision · Action_Editor_VCKi · 2024-06-03

**Recommendation:** Accept as is

**Comment:**

**Paper Summary**

This paper examines the convergence of minimax optimization problems in infinite-dimensional spaces within the context of adversarial training, including GANs and unsupervised domain adaptation.

**Claims and Evidence**

This is a theoretical paper. In the initial review, reviewers tDMF and cgJA expressed concerns about the clarity of the problem formulation. After the rebuttal, none of the reviewers raised further significant concerns regarding the theoretical results. Reviewers also noted the omission of several related works in the domain adaptation field, which the authors addressed in the revised version. Therefore, I find this part satisfactory.

**Audience**

In the initial review, some reviewers questioned the significance of the theoretical results. I acknowledge the relatively modest significance of these results. However, the acceptance criteria for TMLR emphasize interesting insights rather than the significance of the results (i.e., this part should be considered for the featured certificate rather than acceptance). Based on the official recommendations from the reviewers, all of them agree that this paper could offer certain valuable insights for the sub-community. Thus, I find this part satisfactory as well.


**Decision**

Based on the aforementioned discussions, I think this paper meets the criteria for acceptance. The reviewers did not oppose the acceptance decision in their official recommendation. Therefore, I recommend acceptance.

**Audience:**

Yes

**Claims And Evidence:**

Yes.